# TabSTAR: A Tabular Foundation Model for Tabular Data with Text Fields

**Alan Arazi     Eilam Shapira     Roi Reichart**
{alanarazi7, eilam.shapira, roireichart}@gmail.com
Technion - IIT

## Abstract

While deep learning has achieved remarkable success across many domains, it has historically underperformed on tabular learning tasks, which remain dominated by gradient boosting decision trees. However, recent advancements are paving the way for Tabular Foundation Models, which can leverage real-world knowledge and generalize across diverse datasets, particularly when the data contains free-text. Although incorporating language model capabilities into tabular tasks has been explored, most existing methods utilize static, target-agnostic textual representations, limiting their effectiveness. We introduce TabSTAR: a Tabular Foundation Model with Semantically Target-Aware Representations. TabSTAR is designed to enable transfer learning on tabular data with textual features, with an architecture free of dataset-specific parameters. It unfreezes a pretrained text encoder and takes as input target tokens, which provide the model with the context needed to learn task-specific embeddings. TabSTAR achieves state-of-the-art performance for both medium- and large-sized datasets across known benchmarks of classification tasks with text features, and its pretraining phase exhibits scaling laws in the number of datasets, offering a pathway for further performance improvements.[1]

## 1   Introduction

In recent years, deep learning has profoundly reshaped research and practice in computer vision [50, 66, 33, 17] and natural language processing [56, 5, 79, 16, 13]. This transformation was notably accelerated by the rise of foundation models [8, 88, 4], capable of cross-modal understanding and generalization from massive pretraining across heterogeneous data sources. Importantly, they enabled an end-to-end approach that outperformed previous modular alternatives [71, 1]. Moreover, deep learning models excel at transfer learning [91], generalizing from their pretraining data to new tasks. Their strength, combined with techniques like In-Context Learning (ICL) [13] and Parameter-Efficient Fine-Tuning (PEFT) [41], has enabled rapid adaptation to new tasks with only limited labeled data.

Despite this progress, deep learning has historically lagged behind gradient-boosted decision trees (GBDTs) on tabular data [12, 15, 60], in both classification and regression tasks [65, 11, 31, 55]. The heterogeneity of tabular data, which lacks the spatial locality of images or the sequential order of text, makes it more challenging for deep models to learn. Consequently, GBDTs have remained the de facto standard for tabular learning, offering strong out-of-the-box performance, computational efficiency, and built-in inductive biases (e.g., robustness to skewed feature distributions and automatic feature selection) that make them especially well-suited to heterogeneous datasets [31]. Nonetheless, GBDTs cannot be pretrained to reuse strong representations for downstream tasks. This limitation becomes critical in low-data settings like those often found in healthcare applications [52]. Crucially, they must rely on external embedding models to process unstructured data types like text and images, yielding fixed feature representations that cannot be finetuned for a specific prediction task.

---

[1]Code is available at `https://github.com/alanarazi7/TabSTAR`.

39th Conference on Neural Information Processing Systems (NeurIPS 2025).

Table 1: A binary classification toy dataset for hospital patient release outcomes. *Decision* is the target variable. *Age* (numerical), *Department* (high-cardinality), and *Report* (textual) are the features.

| Age | Department | Report | Decision |
|-----|-----------|--------|----------|
| 45 | Cardiology | Mild chest discomfort. | Released |
| 62 | Neurology | Complaints of headache and occasional dizziness. | Hospitalized |
| 38 | Oncology | Completed treatment cycle without adverse reactions. | Released |
| 55 | Neurology | Reports episodes of vertigo and memory lapses. | Hospitalized |

The emerging field of Tabular Foundation Models (TFMs) has begun addressing these shortcomings, introducing powerful cross-dataset learning strategies [84, 47, 38]. However, the flagship model TabPFN-v2 [38] still handles text inputs no more flexibly than conventional GBDTs. This design choice is not incidental; historically, tabular benchmarks have prioritized numerical datasets without free-text features, largely for ease of modeling and evaluation. A recent study [48] of mainstream tabular datasets benchmarks [25, 23, 87, 55] found that half of these datasets are more than 20 years old, being a poor representation of modern real-world data.

Real-world tabular datasets often include high-cardinality[2] and free-text features [14], illustrated by a toy example in Table 1. In such datasets, free-text features (e.g., *Report*) carry rich semantic information critical for tasks like predicting whether a patient will be discharged from the hospital or require continued care. Yet, most models encode them in a target-agnostic manner, delegating to a generic embedding that fails to capture task-specific nuances for predicting *Decision*. Crucially, that same embedding would have been used for a different target variable (e.g., *Treatment Cost*). Similarly, categorical features with dozens of unique values (e.g., *Department*) are difficult to encode efficiently without external knowledge, making naive approaches brittle and limiting generalization. Importantly, the column names, which could guide the model toward more effective representations, are typically ignored. Addressing these limitations is crucial for developing tabular models that leverage semantic information, transfer knowledge from many datasets, and generalize across domains.

In this paper, we introduce **TabSTAR**: a novel **Tab**ular Foundation Model with **S**emantically **T**arget-**A**ware **R**epresentations, designed explicitly for end-to-end handling of purely textual features. By integrating an unfrozen text encoder at its core, TabSTAR can optimize free-text feature representations, demonstrating their clear superiority over alternative frozen embedding approaches. Additionally, it introduces a novel approach of *target-aware tokens*, which inject semantic information about the target variable as part of the input, allowing for efficient parameter sharing and resulting in an architecture with no dataset-specific parameters (see Figure 1). TabSTAR's training is highly efficient[3] and its performance steadily improves with more pretraining data. Empirically, TabSTAR achieves state-of-the-art (SOTA) performance on classification datasets containing textual features, surpassing leading TFMs as well as GBDTs tuned for 4 hours.

## 2 Related Work

This section reviews prior work in supervised tabular learning. We begin with deep learning methods tailored for tabular data, which were applied to a single dataset. We then discuss cross-dataset transfer learning techniques that improve generalization by leveraging related datasets. Next, we cover the field of TFMs, which aim to generalize across diverse tasks and datasets through large-scale pretraining. Finally, we review recent work on applying large language models (LLMs) to tabular data and elaborate on existing AutoML [34] multimodal solutions. As we focus on supervised learning, we do not cover self-supervised methods [82, 6] and their downstream applications.

**Deep Learning on a Single Tabular Dataset**   Several architectures have been proposed to enhance deep learning for tabular data [70, 45, 81, 85]. *TabNet* [3] and *TabTransformer* [43] introduced attention mechanisms into tabular deep learning, while *FT-Transformer* [26] and its improvement [27] jointly integrated numerical and categorical features into a transformer [79]. Other novel approaches leveraged inter-example information at inference time, with *SAINT* [69] proposing row-

---

[2]High-cardinality features are categorical columns with a large number of unique values.

[3]Pretraining within 48 hours on a single A40 GPU. Finetuning with PEFT for a low memory footprint.

level attention between examples, *Non-Parametric Transformers* [49] processing the entire dataset, including labels, in a single forward pass, and *TabR* [28] combining a k-nearest-neighbor mechanism with a traditional Multi-Layer Perceptron (MLP) architecture. Recent works such as *TabM* [29] and *RealMLP* [39] focused on refining MLPs without an attention component. Despite these innovations, single-dataset deep learning models have not yet convincingly outperformed GBDTs [65, 31, 64]. Furthermore, none of them addressed the challenge of modeling tabular data with rich textual features.

**Cross-Dataset Transfer Learning**   Deep learning has been proven to shine when performing transfer learning in many machine learning domains [91]. Motivated by this success, [52, 89] proved that cross-dataset learning can boost single-dataset performance, but were limited to strict requirements such as partial overlap of feature names. To address this limitation, *TransTab* [83] integrated semantic understanding into feature tokenization, and *XTab* [90] pretrained a transformer backbone with dataset-specific parameters, proving that pretraining contributes to a stronger initialization for a downstream task. Despite their small scale, these studies demonstrated cross-dataset transfer learning's potential, laying essential groundwork for the rise of TFMs.

**Tabular Foundation Models**   TFMs represent an emerging paradigm in tabular learning. While the definition is still evolving, we adopt the framing proposed by [77], which identifies key desired characteristics of TFMs: large-scale pretraining with adaptability to downstream tasks, mixed-type column support, cross-domain generalization, use of textual metadata,[4] and column-order invariance.

*TabPFN* [36] is recognized as the first TFM, and its successor *TabPFN-v2* [38] currently sets the SOTA in tabular learning, becoming a popular approach for TFMs [61, 21, 54]. TabPFN-v2 was the first model to consistently outperform GBDTs on medium-sized datasets, by pretraining Bayesian Prior-Data Fitted Networks (PFNs) [58] on 130 million synthetic datasets. Using ICL at inference time, it accepts up to 10,000 examples as input and predicts without updating its weights. TabPFN-v2 inspired *TabICL* [61] which improved scalability, and *TabDPT* [54], trained on real data. However, they all rely on off-the-shelf embeddings for text features like GBDTs, limiting their performance.

*CM2* [86], *CARTE* [47], and *TP-BERTa* [84] represent a shift toward semantic tabular modeling, leveraging textual signals and external knowledge at a greater scale. Unlike prior methods, these models transfer knowledge via language representations. *CM2* was pretrained on over 2,000 datasets, but did not focus on free-text features and used static word embeddings without further finetuning them. *CARTE* encodes tables as star-shaped graphs, jointly representing features by their names and values, and applies attention over the graph to capture contextual relations. While effective for high-cardinality features, it lacks exposure to longer free-text fields during pretraining and was proven useful mainly for small datasets. *TP-BERTa* adapts *RoBERTa* [53] with intra-feature attention and a tokenization scheme that maps numerical values into discrete relative-magnitude bins, to address the weakness of language models when tokenizing numbers [75]. Although it performs well, its use of dataset-specific output layers limits scalability and complicates multi-task learning. Consequently, they trained two separate models,[5] wasting potential for better cross-dataset learning. Notably, none of these approaches finetune semantic representations during downstream task training. In our work, we demonstrate that this is critical to align textual and tabular features.

**Large Language Models for Tabular Data**   The remarkable success of LLMs is unprecedented [13, 59]. During the past years, several research attempts have tried to combine LLMs and tabular data. One line of work focuses on using LLMs directly for tabular prediction by converting tabular data into serialized text. *TabLLM* [35] assessed LLMs under few-shot scenarios, while *Tabula-8b* [24] finetuned the Llama 3-8B model extensively on tabular data. Although useful for few-shot learning, these models are computationally expensive,[6] suboptimal for numerical features [75, 77], and potentially compromised on widely-used benchmarks due to prior exposure during training [9]. While current generations of LLMs weren't adopted for tabular learning, their emergent knowledge from their pretraining could be crucial when textual features are present [77, 20]. Additionally, LLMs can be used in multiple aspects of tabular learning, as they seem to be promising synthetic data generators [10, 68], useful data cleaners [7], and clever feature engineers [37].

---

[4]Contextual information such as the dataset description, column names, and category names.

[5]One for classification and one for regression. A joint model for both tasks performed significantly worse.

[6]Llama 3-8b has orders of magnitude more parameters than TP-BERTa, which has roughly 110M parameters.

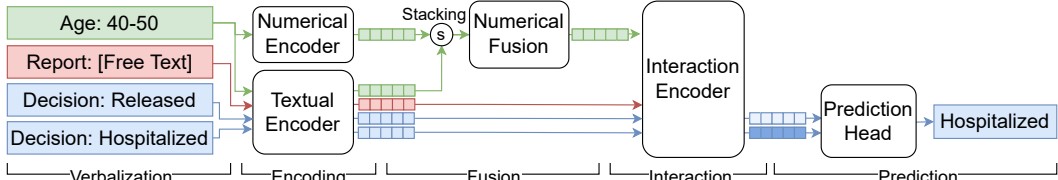

Figure 1: The TabSTAR architecture illustrated with our toy dataset. The model processes numerical features, textual features, and all possible target values for classification.

**Multimodal AutoML** Historically, textual tabular datasets have been largely overlooked in classical tabular benchmarks. However, the AutoML [34] community has made significant progress in developing multimodal solutions. In particular, *AutoGluon* [19] introduced the *AutoML Multimodal Benchmark* [63], initially focusing on text features and later evolving into *AutoGluon-Multimodal* [73, 72], which incorporates images as well. This powerful AutoML framework can fuse text and image foundation models with tabular models and ensemble multiple models through a meta-learning approach [22], making it one of the few systems able to refine static textual representations via joint learning. Nevertheless, this line of work should not be seen as a single model but rather as a highly optimized, production-ready system. According to the authors, it is "a collection of tricks that significantly enhance performance" [73], establishing itself as a robust approach for multimodal, multi-model tabular learning. However, this line of work remains somewhat orthogonal to the development of novel TFMs.

## 3 TabSTAR

In this section, we introduce TabSTAR: a Tabular Foundation Model with Semantically Target-Aware Representations. Our training framework consists of two stages: (1) **Pretraining**, where the model is pretrained over a corpus of tabular datasets[7] in a multi-task regime, mixing classification with regression tasks, then (2) **Finetuning**, where the pretrained model is further trained with *LoRA* [42] on a single downstream task. TabSTAR is designed to enable effective cross-dataset learning by applying supervised learning on the target variable in both stages. At its core, it uses an unfrozen encoder-only language model, which can potentially invoke world knowledge acquired during the language model pretraining.[8] The encoder is combined with a tabular-specific architecture tailored to structured data, mitigating the known limitations of language models in tabular settings [75, 77].

TabSTAR's architecture comprises five core modules: (1) **Verbalization**, mapping every feature into a textual representation composed of both the column name and value, with a special treatment to numerical features for full numerical precision; (2) **Encoding**, transforming semantic and numerical inputs into meaningful embeddings of the same dimension; (3) **Fusion**, integrating textual and numerical representations on each feature independently; (4) **Interaction**, modeling dependencies and relationships between different features through cross-feature self-attention; and (5) **Prediction**, where outputs are projected into a real value for regression or a probability distribution for classification. Figure 1 illustrates the architecture, Appendix A elaborates it, and Appendix B discusses the training.

A key innovation of TabSTAR is the introduction of *target-aware tokens*, a novel approach that integrates the target variable's identity as an input to the model. Unlike existing TFMs [38, 47, 84, 86, 90, 83], which treat the target value as a mere label, TabSTAR fuses target-awareness from the very beginning. For classification tasks, each target value is verbalized and encoded like any other feature. Then, features and target tokens interact with each other, building representations that are then used for prediction. Crucially, this target-awareness allows parameter sharing between all target tokens, which can later use a shared prediction head that maps tokens to probabilities regardless of the number of classes and their identity. By doing so, TabSTAR eliminates the need for dataset-specific components commonly found in prior work [26, 90, 84]. TabSTAR's flexible architecture effortlessly scales[9] to any dataset size, and handles any number of classes in multiclass classification tasks.

---

[7]Ranging from metadata-rich, text-heavy datasets to numeric-only tables lacking column names.

[8]Note that the language-model pretraining occurs before TabSTAR's pretraining. Unless specified differently, the term pretraining refers to TabSTAR's pretraining, which assumes the use of a pretrained language model.

[9]Except when the number of features becomes very large, where memory limitations may arise.

Table 2: An illustrative verbalization of the first patient of Table 1. Each semantic feature is verbalized with its name and value. The numerical *Age* value 45 is standardized (mapped into z-scores, e.g., 0.27) and binned (providing a range to the verbalization, e.g., 40-50, and its quantile). The target variable *Decision* is mapped into its two possible elements, regardless of its original true value.

| Name | Value | Semantic | Numerical |
|------|-------|----------|-----------|
| Age | 45 | "Age: 40–50 (Quantile 50–60%)" | 0.27 |
| Department | Cardiology | "Department: Cardiology" | – |
| Report | Mild chest discomfort. | "Report: Mild chest discomfort." | – |
| Decision | Hospitalized | "Target. Decision: Hospitalized" | – |
| Decision | Released | "Target. Decision: Released" | – |

**Verbalization**   All the features and each of the target values are processed into a sequence of *elements*. Numerical features are processed into two inputs: a numerical one and a semantic one. The numerical input is standardized using z-scores, with outlier clipping at ±3 standard deviations. In addition, they are verbalized using quantile discretization into 10 bins, a novel approach to mitigate the precision loss inherent in language models [75]. Appendix A.1 shows a precise example and §6 discusses different verbalization strategies. In contrast, semantic features are directly verbalized by concatenating the feature name and textual value, without any numerical representation. The target variable is also included as part of the input: In classification tasks, each of the $C$ possible values is represented by an element, constant for every example, while the true value remains hidden. For regression tasks, a single element is verbalized, carrying only the target name. Table 1 shows a toy dataset of patient records and outcomes and Table 2 shows the verbalization for the first patient.

**Encoding**   We employ a pretrained *e5-small-v2* [80] embedding model for semantic encoding, chosen for its strong performance on the *MTEB* benchmark [57] with a relatively modest parameter count. By unfreezing the upper half of its layers, the representations are optimized for predicting the target variable, which leads to a significant impact on TabSTAR performance (see §6). Each verbalization element is encoded independently into a semantic representation, with attention applied between tokens within each sequence element. In parallel, we encode standardized numerical values by projecting them into the same dimension using a small MLP. For the patient in Table 2, this results in a numerical embedding for *Age* alongside semantic representations for each of the 5 verbalizations.

**Fusion**   To obtain a unified representation for each sequence element, we apply a fusion block consisting of a single encoder-only transformer layer. Crucially, each numerical feature is fused independently, as the block attends only to its numerical and semantic embeddings. In our running example, the representation of *Age* now jointly captures both its semantic context (the fact that the value represents age) as well as its numerical value (the patient's age, 45, or 0.27 after standardization).

**Interaction**   The fused, semantically-rich and numerically-grounded representations of all elements interact via a 6-layer Transformer encoder [79]. Each input element is now a token, with feature tokens and target tokens all attending to each other. Unlike standard language models, which integrate positional encoding, the Interaction module's inputs are order-invariant, a desideratum for TFMs, as defined by [77]. The encoder produces contextualized representations for each target value. In our example, this yields dedicated embeddings for the *Release* and *Hospitalization* target values. The role of these representations is to carry information about how likely each value is to be the true value.

**Prediction**   TabSTAR is designed for cross-dataset learning, with shared regression and classification heads used during both pretraining and finetuning. For classification, each of the $C$ target tokens is processed independently through the same classification head, which projects them to scores. We then apply a softmax over all the possible values to yield a probability distribution. Crucially, the fact that target tokens for every class in every dataset share the same classification head allows efficient parameter sharing, flexibly supports any number of output classes, and removes any need for dataset-specific parameters. This is not only efficient during pretraining, but also provides a better initialization for finetuning. In our example, both the *Released* and *Hospitalized* tokens go through the same classification head, which maps them from representations to logits. Applying softmax yields predicted probabilities. For regression tasks, a single target token is projected into a real value.

# 4 Experiments

**The TabSTAR Pretraining Corpus**   While TabSTAR could be pretrained on a massive scale, for this work we limit ourselves to a modest pretraining corpus focusing on classification, as we believe that TabSTAR's inductive biases are best suited to shine in this task. We manually curate a pretraining corpus of 350 high-quality tabular datasets (253 classification, 97 regression), in a tedious process in which we uncover numerous duplications in the most popular tabular repositories, OpenML [78] and Kaggle,[10] as elaborated by [76]. We begin by sourcing datasets from popular benchmarks [25, 47, 23, 22, 31, 63, 55, 32], but observe that the presence of textual tabular datasets in them is very rare. Thus, we furthermore expand our corpus, focusing on classification datasets with rich semantic content. See Appendix C for more details.

**Benchmark**   Tabular datasets with free-text have seen little prior research, and accordingly, benchmarks are rare. To address this, we compile all available datasets from three sources: (1) the *AutoML Multimodal Benchmark* [63], (2) the CARTE paper [47], and (3) the analysis of free-text and high-cardinality features by [32]. After deduplication, our final benchmark includes 50 datasets. Despite its breadth, this collection has two key limitations: First, the benchmark is heavily skewed towards regression tasks, with 36 datasets. Secondly, 29 out of these 36 datasets were solely contributed by the CARTE benchmark, which focuses more heavily on high-cardinality features, rather than longer texts, as it was pretrained over knowledge graphs. While our main motivation is classification tasks with textual features, we decide nevertheless to evaluate on the full set of 50 datasets, although it is biased toward regression problems and high-cardinality features (see Appendix D).

**Baselines**   We compare TabSTAR against **GBDTs**: *Random Forest* [12], *LightGBM* [46], *XGBoost* [15], and *CatBoost* [60]; **MLPs**: *RealMLP* [39] and *TabM*; and **TFMs**: [29], *CARTE* [47], *TabDPT* [54], *TabPFN-v2* [38], and *TabICL* [61] which only supports classification tasks. For several models[11] we consider both default variants as well as tuned ones, where hyperparameters are optimized separately for each task using random search with 5-fold cross-validation under a 4-hour budget on 8 CPU cores. Since the public TabPFN-v2 model does not support text, we use their closed-sourced API client.[12] For models lacking native support for textual features, we embed text using *e5-small-v2* [80], allowing a fair comparison. For more details about the hyperparameters for each baseline as well as exclusion of baselines due to potential leakage concerns, see Appendix E.

**Experimental Setup**   Each of the 50 datasets in the benchmark is evaluated with 10 random train-test splits (90% training, 10% testing), resulting in 500 runs per model. While 30 of the datasets have more than 10,000 examples, most TFMs can't effectively scale beyond it: TabPFN-v2, for example, employs ICL and thus receives as input at most 10,000 examples. While CARTE imposes no strict size cap, its tuning is slow, inefficient, and impractical for larger datasets.[13] Because of these important limitations, we consider two experiment conditions: (1) **10K**: Each model is trained[14] over at most 10,000 training examples, and (2) **Unlimited**: We evaluate TabSTAR and the most competitive, scalable baselines on the full version of the 30 datasets.[15]

**The TabSTAR Training**   To maximize the value of cross-dataset learning, instead of pretraining TabSTAR once, we create five dataset splits. Each variant is pretrained on the 350 pretraining datasets and 40 of the benchmark datasets, while the other 10 serve exclusively as its test set. Crucially, the whole collection was carefully curated to prevent any data leakage from duplicate or overly similar datasets. As a result, each dataset is evaluated by finetuning the single pretrained variant, which excludes it from its pretraining. For finetuning, while dataset-specific hyperparameter tuning can boost performance, we believe that robust out-of-the-box defaults are essential for TFMs and their evaluation. Therefore, we use a default hyperparameter configuration that was found robust over a disjoint set of tabular datasets, as detailed in Appendix B.2.

---

[10]https://www.kaggle.com/datasets

[11]All MLPs and GBDTs, except the naive Random Forest baseline, used as a weak baseline.

[12]https://github.com/PriorLabs/tabpfn-client

[13]In their own paper, CARTE was evaluated only over up to 2,048 examples, without scaling guarantees.

[14]While ICL-based TFMs aren't technically trained, we adopt this term for conciseness.

[15]We technically cap the amount of examples to 100,000 for computational efficiency.

# 5 Results

We evaluate each model using AUROC (classification) and $R^2$ (regression) as metrics. Following [38], we normalize scores per dataset split to the $[0, 1]$ range, using the best and worst model performance as anchors.[16] The normalized scores are averaged across all runs, with 95% CIs. Performance for all models on both conditions is shown in Figure 2 (classification) and Figure 3 (regression). For certain datasets, we are unable to execute TabPFN-v2, CARTE, and TabDPT due to model-specific implementation issues or scalability constraints. Reported averages for these models are computed only over the datasets where evaluation is feasible. Appendix F.1 elaborates on technical limitations, Appendix F.2 on dataset-level performance, and Appendix F.3 on head-to-head comparisons.

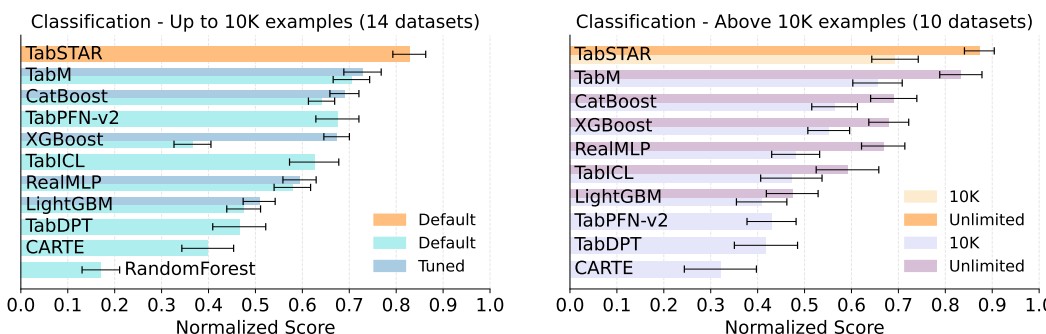

Figure 2: Comparison of normalized scores with 95% CIs between TabSTAR and baseline models in classification tasks, evaluated on up to 10,000 examples (left) and above 10,000 (right).

**In classification problems, TabSTAR consistently achieves SOTA performance.** This is evident both when restricting the dataset size to 10,000 examples and when using larger datasets in the unlimited condition. For the 10K condition, TabSTAR achieves a 0.83 score, followed by TabM-Tuned (0.73), CatBoost-Tuned (0.69), and TabPFN-v2 (0.67). When analyzing head-to-head comparisons (see Appendix F.3), TabSTAR outperforms TabPFN-v2 (8/11 datasets), TabM-Tuned (10/14), and CatBoost-Tuned (12/14). For the unlimited condition, TabSTAR-Unlimited achieves a 0.84 score, followed by TabM-Tuned-Unlimited (0.79), and much above the rest. Importantly, unlimited variants significantly surpass the 10K ones, emphasizing the importance of models with no scaling restrictions.

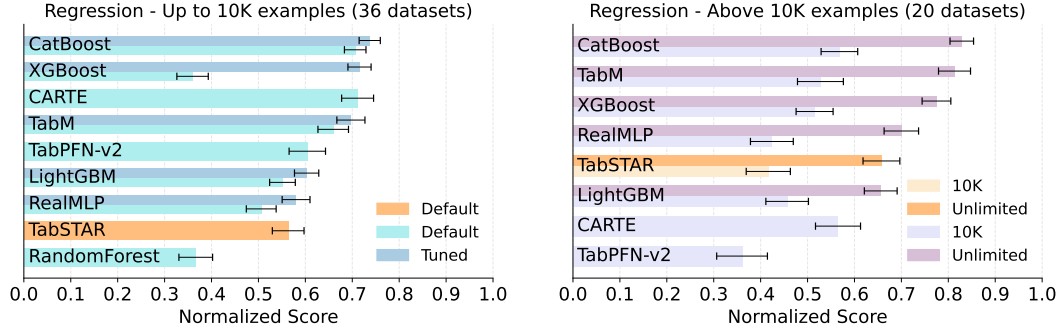

Figure 3: Comparison of normalized scores with 95% CIs between TabSTAR and baseline models in regression tasks, evaluated on up to 10,000 examples (left) and above 10,000 (right).

Although regression is not our main focus, TabSTAR achieves competitive results in the 10K condition, but clearly does not set the SOTA. Surprisingly, while TabPFN-v2 is superior, it significantly underperforms compared to GBDTs which dominate this category. This emphasizes the need for better modeling of textual tabular learning, especially since TabPFN-v2 has shown remarkable performance in non-textual tabular datasets, and CARTE set the SOTA for small datasets. In the unlimited setting, TabSTAR scales well and surpasses other TFMs which cannot scale, but the gap from GBDTs remains significant. §7 discusses this limitation and suggests promising directions for future generations of TabSTAR to achieve SOTA in regression as well.

---

[16]For a single run, the best model gets 1, the worst gets 0, and the rest are linearly scaled accordingly.

Table 3: Cost Analysis. Median training and inference times and peak memory usage on GPU and CPU, aggregated over the 50 datasets of the benchmark with up to 10,000 examples.

| Model | Train | | | Inference | | |
| --- | --- | --- | --- | --- | --- | --- |
| | Time (s) | GPU (GB) | CPU (GB) | Time (s) | GPU (GB) | CPU (GB) |
| CatBoost-CPU | 360.8 | – | 2.2 | 34.0 | – | 2.1 |
| CatBoost | 68.5 | 1.3 | 1.5 | 2.0 | 1.3 | 1.5 |
| LightGBM | 39.1 | 1.3 | 1.5 | 2.0 | 1.3 | 1.5 |
| RandomForest | 86.1 | 1.3 | 1.5 | 2.5 | 1.3 | 1.5 |
| RealMLP | 136.4 | 1.4 | 1.7 | 2.5 | 1.3 | 1.7 |
| TabDPT | 10.4 | 2.2 | 1.8 | 161.4 | 33.8 | 6.0 |
| TabICL | 42.6 | 1.2 | 1.7 | 13.4 | 25.4 | 2.1 |
| TabM | 19.1 | 1.8 | 4.4 | 1.5 | 1.8 | 4.3 |
| TabSTAR | 493.2 | 4.7 | 1.8 | 3.0 | 1.3 | 1.8 |
| XGBoost | 38.4 | 1.3 | 1.5 | 2.5 | 1.3 | 1.5 |

**Cost Analysis**    Table 3 reports the computational requirements of TabSTAR and competing base-lines, measured by runtime and peak memory usage, for both training and inference. To highlight the importance of GPU acceleration when text embeddings are part of the preprocessing, we include a variant of CatBoost restricted to CPU only. We exclude TabPFN-v2, whose performance is probably comparable to TabICL, since it is only accessible via an API, and CARTE, whose extensive tuning requirements and prohibitive training costs forced evaluation on less capable hardware.

We find that TabSTAR incurs modest inference costs, comparable to GPU-accelerated GBDTs, since GPU processing of text embeddings dominates runtime. This is evident from the CatBoost-CPU variant, whose inference is nearly ten times slower than TabSTAR. Moreover, TabICL and TabDPT are orders of magnitude slower and consume far more memory, emphasizing the scalability limitations for ICL-based models. During training, TabSTAR is considerably slower than the baselines. However, while fitting a single GBDT is much faster, this advantage quickly diminishes in practice when hyperparameter tuning is needed or GPU acceleration for text embeddings is not available. Appendix F.4 provides dataset-level runtimes and F.5 analyzes costs as a function of the feature count.

## 6    Analysis

We analyze the factors contributing to TabSTAR's strong performance by addressing three key research questions: **Q1:** How important is the encoder language model unfreezing? **Q2:** Does the number of datasets during pretraining contribute to the downstream task performance? and **Q3:** How do different verbalization methods of numerical features impact performance?

To answer these questions, we pretrain several variants of TabSTAR for each analysis, limiting ourselves to a subset of the tabular datasets used for the main experiment (see §4). Specifically, each variant is pretrained over 256 datasets[17] including 30 datasets from our benchmark, and evaluated over the remaining 20 datasets (12 regression, 8 classification). This reduced setup allows leveraging transfer learning and exploiting our corpus, without the burden of training multiple folds per variant. Appendix G.1 lists the 20 datasets used for evaluation along with per-dataset results.

**Q1: The Role of the Encoder Unfreezing**    We examine the effect of unfreezing different numbers of textual encoder layers during pretraining. In finetuning, we keep all base layers frozen and instead apply LoRA adapters to the same layers that were unfrozen in pretraining.[18]  Figure 4 shows the validation loss during TabSTAR pretraining (left) and the normalized score on the downstream tasks (right) as a function of the number of unfrozen encoder layers. Notably, unfreezing even a single encoder layer significantly outperforms using static embeddings. Further substantial improvements are observed as more layers are tuned, with the best results achieved when unfreezing 6 layers. While unfreezing 9 layers shows lower performance, it is plausible that adding more datasets to the pretraining phase will affect this finding. See Appendix G.2 for more details.

---

[17]Except for variants of Q2, which analyze the effect of the number of datasets on pretraining.

[18]For simplicity, we will refer to them as unfrozen to differentiate them from layers without LoRA.

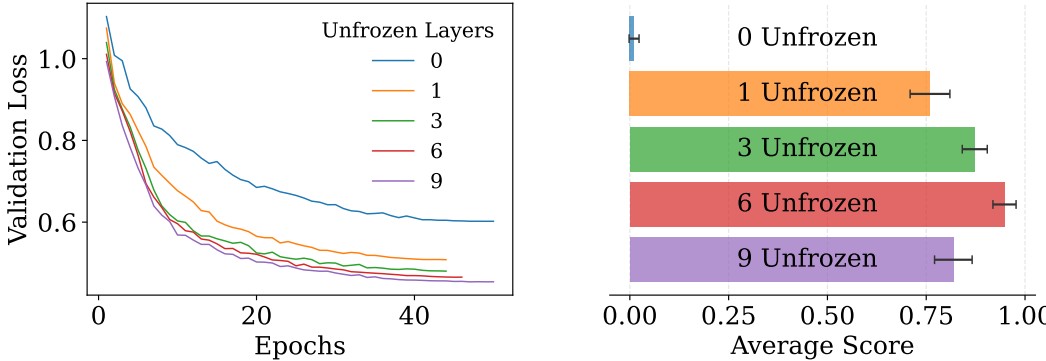

Figure 4: Performance as a function of the number of encoder layers unfrozen: Validation loss during TabSTAR's pretraining (left) and normalized scores with 95% CIs on the downstream tasks (right). Unfreezing even a single encoder layer significantly improves the performance of TabSTAR.

**Q2: The Effect of Pretraining**  To evaluate the impact of pretraining on TabSTAR's downstream performance, we compare a pretrained version of TabSTAR with a version that was finetuned from scratch.[19] In line with previous work [90, 86], the pretrained model performs significantly better, highlighting the critical role of transfer learning for TabSTAR's success. To further investigate the effect of the number of pretraining datasets on downstream task performance, we train two additional versions: one pretrained on 16 datasets and another on 64 datasets.

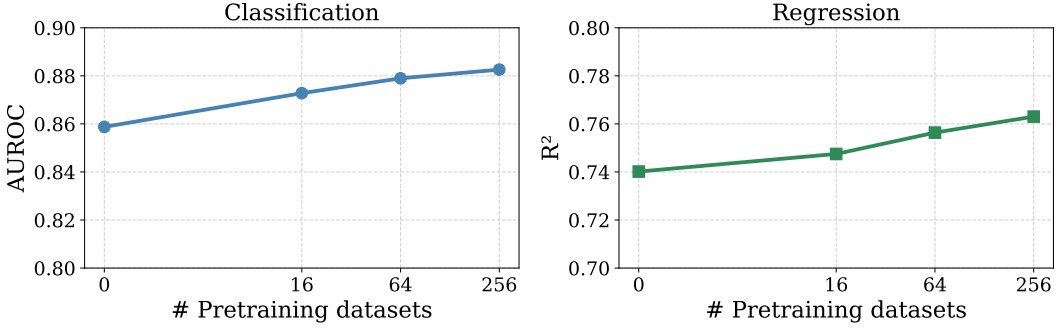

Figure 5: Average performance on downstream tasks as a function of the number of pretraining datasets (in log scale). We use AUROC for classification (left), and $R^2$ for regression (right).

As shown in Figure 5, increasing the number of pretraining datasets consistently improved performance in both classification and regression tasks. Notably, the substantial gain in regression tasks suggests that TabSTAR's downstream performance on §5 could improve with more pretraining data, potentially reaching SOTA performance with enough scale. See Appendix G.3 for more details.

**Q3: Numerical Verbalization**  A key challenge in integrating language models with numerical data is determining how to best represent numerical values within a linguistic framework. While some semantic tabular methods omit numerical features from the verbalization [83, 86], TP-BERTa [84] introduced *Relative Magnitude Tokenization* [84], which encodes numerical information through non-semantic special bin tokens. In contrast, TabSTAR injects semantic numerical information into the verbalization of numerical features, as illustrated in Table 2. To quantify the effect of our novel verbalization, we explore two thinner variants: (1) **Name + Bin**, which excludes the quantile information, and (2) **Name**, which omits numeric information entirely and verbalizes the feature name only.  Appendix G.4 shows an illustrative example for each variant and presents the full results. As demonstrated in Table 4, our findings reveal that incorporating numerical information significantly enhances performance, highlighting the importance of balancing numerical precision with a representation format that aligns with the language model's parametric knowledge.

---

[19]Since LoRA underperforms on random weights, we finetune the entire non-pretrained model.

Table 4: Normalized score with 95% CIs by the numerical verbalization method.

| Verbalization Method | Name | Name + Bin | TabSTAR |
|---|---|---|---|
| Classification | 0.386 ± 0.095 | 0.544 ± 0.093 | 0.593 ± 0.097 |
| Regression | 0.386 ± 0.081 | 0.584 ± 0.076 | 0.596 ± 0.079 |

## 7 Discussion and Conclusion

We introduce TabSTAR, a Tabular Foundation Model with Semantically Target-Aware Representations, which integrates textual features through an unfrozen pretrained encoder. In addition, its novel target-aware tokens enable efficient cross-dataset generalization without dataset-specific parameters. Despite limited pretraining data and a relatively small text encoder [80], TabSTAR sets the SOTA in tabular classification with textual features, significantly surpassing GBDTs and leading TFMs.

Since scaling laws in data and model size have proven themselves for LLMs [44] and TabSTAR improves with the number of pretraining datasets (see §6), future work should scale TabSTAR across both model and data dimensions. For model scaling, we envision a family of model sizes, common for LLMs [30, 74, 51], that will allow a trade-off between quality and costs. Data scaling might leverage self-supervised learning [62, 82] over large-scale table corpora [18], or realistic synthetic tabular data generators [10], which have proven successful [38, 2]. At scale, it could potentially unlock few-shot learning capabilities and develop automatic feature-engineering skills [37].

Beyond scaling, TabSTAR's semantic approach has tremendous potential to explicitly include world knowledge by leveraging LLMs, which to date have had a limited impact on tabular learning. As a few motivating examples, LLMs could improve TabSTAR's numerical verbalization binning approach by providing semantically informed thresholds, or by providing explicit world knowledge that could be injected as a strong prior in small data scenarios. While these directions seem like plausible research paths, they come with a risk of data leakage due to the memorization properties of LLMs [9]. Evaluating TFMs fairly while keeping benchmarks uncontaminated would be an important enabler for tabular research. As a step in this direction, we are releasing several TabSTAR variants, each with a different dataset withheld during pretraining, ensuring that for every dataset there is a TabSTAR model that has never seen it. We urge fellow researchers to adopt this approach in their own work.

While TabSTAR sets a new bar in classification, its regression results lag behind GBDTs, which outperform other TFMs as well. This gap could be narrowed through additional scaling, by exploring regression-via-classification techniques like [38, 2], and by enriching the numerical encoder with distribution-aware statistics per feature as done by [61]. In addition, similar to other TFMs, TabSTAR may encounter memory bottlenecks on datasets with hundreds of features, and its training speed lags behind untuned GBDTs and ICL-based TFMs. Yet TabSTAR achieves GBDT-level efficiency during inference, as opposed to TabPFN-v2 and TabICL. Furthermore, TabSTAR has not been extensively evaluated in few-shot scenarios and in purely numerical datasets.

Despite these limitations, TabSTAR offers a promising pathway toward improving performance on tabular datasets with textual fields, common in industries with high social impact (e.g., healthcare, education), or with significant economic value (e.g., banking, manufacturing). In addition, its architecture lends itself to multimodality and could be extended to tabular datasets that combine numerical, textual, and image features. We believe TabSTAR paves the way for a new generation of semantically enriched, Multimodal TFMs, and invite the research community to advance this vision.

## Acknowledgments and Disclosure of Funding

Roi Reichart and Eilam Shapira have been partially supported by a VATAT grant on data science.

We thank Omri Feldman for brainstorming since the very beginning; Elad Hoffer and Ofir Lindenbaum for consulting and feedback; David Holzmüller, Lennart Purucker, Myung Kim, and Gaël Varoquaux for assisting with evaluations and benchmarks; and Frank Hutter, Noah Hollmann, Léo Grinsztajn, and the rest of the Prior Labs team for providing extensive access to TabPFN-v2's API version.

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

# A   Architecture

This appendix provides additional technical details for the architecture introduced in §3. First, we discuss the verbalization module; next, we formally describe the architecture step-by-step; and finally, we present selected experiments on the TabSTAR architecture.

## A.1   The Verbalization Module

TabSTAR's verbalization module standardizes heterogeneous tabular inputs by converting each column, whether predictive feature or target variable, into templated text blocks. We first describe the detection of column types, and then detail the processing steps for each type.

**Feature Detection**   We classify each column as either numerical, referring to quantitative values, or semantic, referring to textual values including categorical and boolean fields encoded as text. We rely on heuristics involving both the primitive data type (e.g., string, float) and human annotation (e.g., OpenML metadata). However, real-world datasets pose challenges, as numerical features can often be stored as strings (e.g., "35 years", "unknown age") or may lack inherent order (e.g., country calling codes). Leveraging LLMs for contextualized data cleaning can be a promising direction [7].

A special case is the handling of timestamp and date columns. Similarly to [40], we rely on *skrub's DatetimeEncoder*[20] to detect datetime columns and decompose each one of them into a set of new features. Each extracted feature then undergoes its own processing: For example, the weekday is treated as semantic, while the total seconds since the Unix epoch is treated as numerical. Integrating date features more holistically remains an open research question.

Table 5: Illustrative verbalization of a numerical feature (*Age*) with 10 bins. Examples outside the range and missing values are considered as special bins.

| Bin | Range | Example Value | Illustrative Verbalization |
|---|---|---|---|
| – | Lower than 18 | 17 | Age: Lower than 18 (Quantile 0%) |
| 1 | 18–23 | 20 | Age: 18–23 (Quantile 0–10%) |
| 2 | 23–27 | 25 | Age: 23–27 (Quantile 10–20%) |
| 3 | 27–31 | 29 | Age: 27–31 (Quantile 20–30%) |
| 4 | 31–35 | 33 | Age: 31–35 (Quantile 30–40%) |
| 5 | 35–40 | 38 | Age: 35–40 (Quantile 40–50%) |
| 6 | 40–45 | 42 | Age: 40–45 (Quantile 50–60%) |
| 7 | 45–51 | 48 | Age: 45–51 (Quantile 60–70%) |
| 8 | 51–58 | 55 | Age: 51–58 (Quantile 70–80%) |
| 9 | 58–67 | 63 | Age: 58–67 (Quantile 80–90%) |
| 10 | 67–87 | 83 | Age: 67–87 (Quantile 90–100%) |
| – | Higher than 87 | 93 | Age: Higher than 87 (Quantile 100%) |
| – | Unknown | – | Age: Unknown Value |

**Numerical Features**   Numerical features are represented by both a numerical and a semantic representation. For the numerical representation, given a value $x$, we compute the clipped z-score $z' = \mathrm{clip}\big((x - \mu)/\sigma, -3, 3\big)$ where $\mu, \sigma$ are the training set mean and the standard deviation, and missing values are set to 0. For the semantic representation, we build $B = 10$ quantile bins over the training distribution to map the value accordingly. Table 5 shows an illustrative example for the feature *Age* from our running example in Table 2.

**Semantic Features**   Semantic features are sanitized (e.g., normalizing whitespaces) and verbalized using the template presented in Table 6. Missing values are mapped to "Unknown Value", just like for numerical features. If a text exceeds the model's context window (512 tokens for *e5-small-v2*), it is naively truncated to fit it. This limitation is far more pronounced for methods that serialize the entire example into a single textual sequence [84], thereby dramatically reducing the effective context size.

---

[20]https://skrub-data.org/

**Target Variables**   The verbalization templates for the target values are prepended to every example. For classification tasks, each possible label is verbalized, while for regression we verbalize a single element consisting solely of the feature name. Employing a binning strategy to treat regression as a classification task is a future work direction, as discussed in §7. For regression tasks, target values go through the same standardization with outlier clipping as numerical features, being used solely as the ground truth without going through the input.

Table 6: Verbalization templates for semantic features and target values.

| Element Type | Verbalization Template |
|---|---|
| Predictive feature | "Predictive Feature: {feature_name}" 
 "Feature Value: {feature_value}" |
| Classification target | "Target Feature: {target_name}" 
 "Feature Value: {target_value}" |
| Regression target | "Numerical Target Feature: {target_name}" |

## A.2   The Annotated TabSTAR

Table 7 describes the number of parameters per component in the TabSTAR architecture, when using *e5-small-v2* [80] as the text encoder. It has approximately 47.26M parameters, most of which come from the text encoder. When unfreezing 6 layers of the text encoder, about 24.70M parameters are tuned, with the remaining 11.92M embedding parameters and 10.65M layer ones being kept frozen.

Table 7: Parameter counts for TabSTAR components

| Module | # Parameters |
|---|---|
| Encoding: Semantic | 33,360,000 |
| Encoding: Numerical | 296,832 |
| Fusion | 1,774,464 |
| Interaction | 10,646,784 |
| Prediction | 1,185,794 |

To describe the architecture more precisely, we start by defining the dataset formally. Let $\mathcal{D} = \{(x_i, y_i)\}_{i=1}^n$ denote a tabular dataset with $n$ examples. Each example $x_i = [x_{i1}, \ldots, x_{im}]$ has $m$ features. The target variable $y_i$ is either continuous (regression) or discrete (classification) taking one of $C$ classes. For simplicity, we describe the architecture at the example level, though all computations are actually carried out on mini-batches of size $B$. The batches are always drawn from a single dataset in both pretraining and finetuning, removing any need for padding.

**Verbalization**   We denote by $t$ the number of target entries where $t = C$ for classification and $t = 1$ for regression. We then form a raw sequence of length $e = t + m$ by listing the $t$ target values, followed by the $m$ feature entries. Each element $j$ in this sequence is then verbalized into a semantic string $s_j$ and a numerical value $n_j$, set to be the clipped z-score for numerical non-missing features, and zero otherwise. The example is thus represented by parallel sequences $(\mathbf{s}, \mathbf{n})$ of length $e$.

**Encoding**   Each semantic string $s_j$ and numerical value $n_j$ are projected into a $d$-dimensional vector. Semantic strings are encoded with an encoder-only language model (*e5-small-v2 [80]*). Each string is tokenized, passed through the model, and pooled to produce its final embedding. This process is independent between elements, i.e. the attention is at the token level within a single element. In parallel, each numeric value is fed through a two-layer MLP that first projects from 1 to $2d$ dimensions, applies a ReLU and dropout, and then projects back down to $d$. This produces matching $d$-dimensional embeddings for each of the $e$ elements, ready to be fused.

**Fusion**   To unify semantic and numerical embeddings into a single representation, we apply a single-layer Transformer Encoder[21] over each element's pair of vectors. Concretely, for each element

---

[21]With 2 attention heads, a feed-forward hidden size of $4d$, dropout 0.1, and ReLU activation.

we stack its $d$-dimensional text and numeric embeddings and feed them through the encoder layer. For every element, the attention is applied between its two representations, and we average the two outputs to produce one fused $d$-dimensional embedding. This yields a final sequence of length $e$ and dimension $d$, which will serve as tokens for the Interaction block.

**Interaction**   The fused sequence of $e$ tokens is processed by a standard Transformer Encoder with model dimension $d = 384$, $L = 6$ layers, 6 attention heads per layer, feed-forward size $4d$, dropout 0.1, ReLU activation and using a pre-norm configuration. Unlike in language modeling, feature ordering is irrelevant, so no positional encodings are used. The encoder produces contextualized embeddings for every position, and we retain the $t$ target embeddings for the prediction.

**Prediction**   We initialize two identical MLP heads, one for regression and one for classification. Each of them consists of a hidden layer of size $4d$ (with ReLU activation) followed by a linear projection to a single output. For each dataset, we choose the relevant head and process the $t$ target token embeddings. For classification, we independently feed each one of the $t = C$ target tokens to the classification head to obtain a score (logit) per class. Notably, the same head is shared across classes and datasets. We apply softmax over these scores, yielding a probability distribution regardless of the number of classes. For regression, the single target token is projected into a real value. Note that the heads is shared between datasets, as regression outputs are always clipped z-scores.

### A.3   Architecture Experiments

In this section we explore the effect of different design choices for TabSTAR's architecture. For each experiment, we only vary the parameter of interest, keeping everything else fixed. We follow the same pretraining regime as in Appendix B.1, except that for computational efficiency we train only 25 epochs (instead of 50) with 128 pretraining datasets (instead of 390). We evaluate each variant relying solely on pretraining performance, as an approximation for downstream task performance. We acknowledge that our conclusions might depend on this limited scale, hence we discuss a subset of the experiments briefly to reflect the depth of our work and inspire future research.

**The Fusion Block's Mechanism**   For the fusion block, we consider two simpler alternatives to the attention mechanism, both of them underperforming: (1) Concatenation, by concatenating the semantic and numerical $d$-dimensional vectors into a $2d$-dimensional vector, and projecting them back via an MLP, and (2) Multiplication, by multiplying the semantic representation directly with the numerical value[22] in a straightforward, parameter-free manner as in [83, 86].

**The Number of Interaction Layers**   We experiment with the number of encoder layers, and observe that 3 yields anecdotally worse performance than 6, with lower parameter count. Nevertheless, we prioritize a deeper network as for datasets with very complex relationships we believe that this might be beneficial. Additionally, we try a 9-layer variant which performs significantly worse, while also increasing the parameter count.

**Row-Level Attention**   We experiment with adopting the architecture proposed by SAINT [69], which adds row-level attention to each encoder layer. Similar concepts are also employed by models that get the input the whole dataset, labels included [38, 49]. We run experiments with 2, 4 and 6 layers as they are roughly equivalent in parameter count to 3, 6, and 9 layers without row-attention. We observe no substantial gain, and thus we prioritize the simpler solution, as row-level attention is sensitive to the batch size and adds complexity to inference time.

**Verbalization**   We experiment with various verbalization strategies. For features, including the column name alongside the value consistently outperforms using the value alone. For target-aware tokens, we find that explicitly verbalizing all possible target values yields better performance than omitting them. In contrast, adding a dataset-level description prefix to improve contextualization offers no measurable advantage. Interestingly, TabSTAR maintains high performance even when no semantic metadata is present. We see this robustness as a strength.

---

[22]After rescaling it to be centered around 1 rather than 0, using a learned scaling factor.

# B  Training

In this section we elaborate on the two stages of TabSTAR's training: pretraining and finetuning, presented in §3. As in Appendix A.3, we summarize key pretraining experiments.

## B.1  Pretraining

TabSTAR is pretrained employing supervised learning in a multi-task regime, jointly learning regression, binary and multiclass classification tasks. The parameters of the architecture are fully-shared, without any need for dataset-specific parameters. Every example during the pretraining updates all the model's weights, with the sole exception of the prediction head, for which every example uses its respective head depending on the nature of the task (classification or regression).

**Sampling**  For computational efficiency, each dataset is subsampled once before the pretraining. At the example level, we sample up to 300,000 examples from each dataset, stratified by the target variable for classification tasks. Since we only use a fraction of each dataset for each pretraining epoch, this decision has negligible influence. In addition, we randomly sample up to 200 features per dataset. While straightforward, this decision is suboptimal as feature importance isn't taken into consideration. As this work does not focus on wide-feature datasets, we consider this trade-off acceptable. Importantly, this setup is enabled during finetuning as the TabSTAR architecture is agnostic to the number of features. We split each dataset into train-validation splits (95%-5%),[23] without any need for test splits, and cap the validation set at a maximum of 1,000 examples used for evaluating pretraining performance.

**Batching**  Every epoch, we randomly sample up to 2,048 examples from each dataset in mini-batches of 32, and shuffle all the batches. We conduct gradient accumulation and update the model every 4 steps to reduce the chances of a single update being dominated by a single dataset, so the global batch size is effectively 128. Appendix B.3.1 elaborates on the effect of batch size.

**Metrics**  Our loss function is cross-entropy for classification, and MSE for regression. With standardized targets, $R^2 \approx 1 - \text{MSE}$, although this equivalence is degraded by clipping targets in preprocessing. We train with mixed-precision and apply gradient clipping to stabilize training without task-specific weights, with Appendix B.3.2 discussing the limitations of this approach. We use as metrics AUROC for classification and $R^2$, so for each task the optimal metric value is 1. We average performance across all datasets into a single metric that reflects the pretraining performance.

**Training**  We pretrain for 50 epochs with the *OneCycleLR* [67] optimizer, with warmup during the first 5 epochs (10%) and cosine annealing. Early stopping is conducted after 3 epochs without improvement on the pretraining metric. The weight decay is set to 0.001, and a max learning rate of $lr = 5 \times 10^{-5}$ is applied uniformly across all layers. Appendix B.3.3 discusses experiments with differential learning rate. Pretraining running time varies depending on the number of epochs and the included datasets. The full models (390 datasets) reported in §5 train for less than 48 hours on a single NVIDIA A40 GPU (48GB memory), and we believe that this could be optimized much further.

## B.2  Finetuning

We finetune downstream tasks using LoRA's implementation of the *peft* package[24]. We use a rank of $r = 32$, set $\alpha = 2r = 64$ and $dropout = 0.1$. We employ the same scheduler as in the pretraining phase, with the only difference being that we set $lr = 0.001$, and increase the patience parameter for early stopping to 5. We apply a train-test split of 90%-10% and sample a validation set of 10%. As opposed to the pretraining, all batches are drawn from the same dataset. Therefore, we observe no effect from changing the mini-batch size when keeping the global batch size fixed to 128. We tune 1,597,440 out of TabSTAR's 47,263,874 parameters (3.4%), spanning all blocks of the architecture. The frozen layers of the text encoder do not receive LoRA adapters.

Finetuning hyperparameters are selected by pretraining TabSTAR over 256 datasets and performing grid-search over a held-out set of 25 downstream tasks disjoint from the 50 datasets in the benchmark

---

[23]We choose only 5% for efficiency, as we use hundreds of pretraining datasets.
[24]https://github.com/huggingface/peft

evaluated in §4. The search space is presented in Table 8, and we observe that average performance is relatively robust across this space. An interesting observation is that decreasing the number of parameters by setting $r = 16$ mildly hurts performance, but it has upsides on memory and latency aspects, allowing a future trade-off exploration. As a final note, we argue that providing a strong default configuration for TFMs is crucial for evaluating them, but for real-world applications, it is still recommendable to find the best hyperparameters tailored to the downstream task.

Table 8: LoRA hyperparameter tuning grid search for TabSTAR's finetuning.

| Hyper-parameter | Search Space |
|---|---|
| LoRA rank ($r$) | 16, 32, 64 |
| Learning Rate | 0.0005, 0.001, 0.002, 0.005, 0.01 |
| Dropout | 0, 0.1 |

The only experiment in this paper where we employ full finetuning instead of LoRA is for the non-pretrained variant discussed in the analysis in §6. For this variant we fully finetune the pretrained model on each downstream task. Compared to the pretraining, we use $lr = 2.5 \times 10^{-5}$ and increase the patience to 5. These hyperparameters are lightly tuned using the same procedure as for LoRA, and we observe that fully finetuning the model achieves comparable performance, except for small datasets, where training is more prone to overfitting.

## B.3 Pretraining Experiments

In this section, we briefly elaborate on some experiments performed over TabSTAR's pretraining protocol. As in Appendix A.3, we highlight only a subset of them in a high-level manner.

### B.3.1 Batch Size

During pretraining, we use a mini-batch size of 32, each of them drawn from a single dataset. Since we train with gradient accumulation and a global batch size of 128, varying the batch size affects the diversity of a single model update: lower batch sizes are likely to be exposed to more datasets. We decrease the batch size to 16 and 8 and observe an improvement at the cost of slower training. An interesting direction for future work is moving to mixed-datasets batches, which require more complex implementation but might benefit from more regularized learning. Such approach, however, goes against row-level attention methods and ICL, as discussed in Appendix A.3.

### B.3.2 Loss Weights

Pretraining the model over hundreds of datasets in a multi-task regime presents a key challenge: the loss scale of each dataset can vary substantially, depending on task difficulty or task type. For example, a multiclass classification task with dozens of classes will naturally yield a higher average loss than a binary task. These dynamics can also shift during training. Our default approach naively averages the loss across all datasets, which risks over-weighting tasks for potentially arbitrary reasons.

To address this, we explore two alternative weighting strategies: (1) Assigning a constant weight per task type, accounting for the number of classes in classification tasks, and (2) Normalizing each dataset's contribution by the best loss achieved by CatBoost [60] when being fitted to that dataset. While these strategies better reflect task-specific characteristics, they hardly impact performance and introduce additional complexity. Notably, adjusting loss weights across tasks impacts metric interpretability, as each weighting scheme implicitly optimizes a different objective.

We do not explore more sophisticated methods such as learning per-dataset weights, as these often require mixed-dataset batches and introduce additional learnable parameters. We believe, however, that multi-task pretraining over tabular datasets remains an open and important research question.

### B.3.3 Differential Learning Rate

TabSTAR's weights initialization is not balanced: the textual encoder is a pretrained embedding model while the rest of the architecture parameters are randomly initialized. To counteract this imbalance, we experiment with using differential learning rates for the textual encoder layers, and experiment

with scaling it by a factor of 0.5 and of 0.75. To our surprise, this decision hurts performance, so we stick to a uniform learning rate across all layers.

## C   Training Datasets

In this appendix we briefly expand on the pretraining corpus elaborated in §4. It is composed of 350 datasets, spanning all the datasets appearing in *AMLB* [25], *OpenML-CTR23* [23], *TabZilla* [55] and the ones presented by *Grinsztajn* [31]. After deduplication,[25] this results in 152 datasets (94 classification, 58 regression). Interestingly, only 6 of these 152 datasets have free-text or high-cardinality features. We manually add datasets from OpenML [78] and Kaggle, as well as from the *AutoML-Benchmark-Train* [22] corpus, and achieve a total of 350 datasets, with 49 textual datasets.

Table 9 details the 253 classification datasets and Table 10 the 97 regression ones. We elaborate the *Dataset* name, the number of examples $n$, the number of features $m$, and the number of classes $C$ for classification. In addition, we mark datasets that belong to one of the benchmarks, and the ones that have text features. Importantly, the textual flag is quite permissive, as it includes features with relatively short texts or potentially low predictive power (e.g., people names or addresses).

Table 9: The 253 Classification Datasets of the Pretraining Corpus, with their $n$ examples, $m$ features, $C$ classes, presence in a benchmark ($B$) and whether they are textual ($T$).

| Dataset | $n$ | $m$ | $C$ | B | T |
|---|---|---|---|---|---|
| KDDCup99 | 4,898,422 | 40 | 20 | ✓ | |
| mimic_extract_los_3 | 4,155,270 | 17 | 68 | | ✓ |
| Online-P2P-Lending | 2,875,146 | 16 | 5 | | |
| sf-police-incidents | 2,215,023 | 8 | 2 | ✓ | ✓ |
| physionet_sepsis | 1,552,210 | 42 | 2 | | |
| poker-hand | 1,025,009 | 10 | 10 | ✓ | |
| Higgs | 1,000,000 | 28 | 2 | ✓ | |
| BAF_base | 1,000,000 | 30 | 2 | | |
| Credit_Card_Fraud_ | 1,000,000 | 7 | 2 | | |
| Harry-Potter-fanfiction-data | 648,493 | 13 | 4 | | ✓ |
| porto-seguro | 595,212 | 57 | 2 | ✓ | |
| covertype | 581,012 | 54 | 7 | ✓ | |
| AVIDa-hIL6 | 573,891 | 3 | 2 | | ✓ |
| airlines | 539,383 | 7 | 2 | ✓ | ✓ |
| HolisticBias | 472,991 | 14 | 4 | | ✓ |
| albert | 425,240 | 78 | 2 | ✓ | |
| DBPedia | 342,781 | 3 | 219 | | ✓ |
| hcdr_main | 307,511 | 120 | 2 | | |
| Mental_Health_Dataset | 292,364 | 16 | 5 | | |
| Kuzushiji-49 | 270,912 | 784 | 49 | | |
| spoken-arabic-digit | 263,256 | 14 | 10 | | |
| cdc_diabetes | 253,680 | 21 | 2 | | |
| skin-segmentation | 245,057 | 3 | 2 | | |
| LT-Vehicle-Loan-Default-Prediction | 233,154 | 38 | 2 | | |
| Churn_Telco_Europa | 190,776 | 17 | 2 | | |
| ldpa | 164,860 | 6 | 11 | | |
| Give-Me-Some-Credit | 150,000 | 10 | 2 | ✓ | |
| walking-activity | 149,332 | 4 | 22 | | |
| social_bias_frames | 144,649 | 16 | 3 | | ✓ |
| Wikipedia_Talk_Labels | 140,379 | 12 | 15 | | ✓ |
| Municipal-Debt-Risk-Analysis | 138,509 | 13 | 2 | | |
| MiniBooNE | 130,064 | 50 | 2 | ✓ | |
| nba-shot-logs | 128,069 | 15 | 2 | | ✓ |
| | | | | | Continued on next page |

---

[25] And the exclusion of the *fifa* dataset, which is included in the benchmark.

Table 9: The 253 Classification Datasets of the Pretraining Corpus.

| Dataset | $n$ | $m$ | $C$ | B | T |
|---|---|---|---|---|---|
| college_scorecard | 124,699 | 117 | 2 | | |
| drug-directory | 120,215 | 16 | 7 | | ✓ |
| TVS_Loan_Default | 119,528 | 29 | 2 | | |
| road-safety | 111,762 | 32 | 2 | ✓ | |
| Diabetes130US | 101,766 | 46 | 3 | ✓ | |
| fars | 100,968 | 29 | 8 | | |
| Credit_Score_Classification | 100,000 | 26 | 3 | | ✓ |
| numerai28.6 | 96,320 | 21 | 2 | ✓ | |
| Run_or_walk_information | 88,588 | 6 | 2 | | |
| jannis | 83,733 | 54 | 4 | ✓ | |
| KDD98 | 82,318 | 477 | 2 | | |
| APSFailure | 76,000 | 169 | 2 | ✓ | |
| kick | 72,983 | 32 | 2 | ✓ | ✓ |
| human-choice-prediction | 71,579 | 20 | 2 | | ✓ |
| Traffic_violations | 70,340 | 20 | 3 | | ✓ |
| Fashion-MNIST | 70,000 | 784 | 10 | ✓ | |
| Cardiovascular-Disease-dataset | 70,000 | 11 | 2 | | |
| mnist_784 | 70,000 | 719 | 10 | | |
| connect-4 | 67,557 | 42 | 3 | ✓ | |
| mobile_churn | 66,469 | 63 | 2 | | |
| helena | 65,196 | 27 | 100 | ✓ | |
| LICD | 63,634 | 413 | 2 | | ✓ |
| CIFAR_10 | 60,000 | 3,072 | 10 | | |
| REASONER | 58,497 | 34 | 2 | | ✓ |
| volkert | 58,310 | 147 | 10 | ✓ | |
| shuttle | 58,000 | 9 | 7 | ✓ | |
| GTSRB-HueHist | 51,839 | 256 | 43 | | |
| okcupid-stem | 50,789 | 19 | 3 | ✓ | ✓ |
| KDDCup09-Upselling | 50,000 | 13,419 | 2 | ✓ | |
| KDDCup09_appetency | 50,000 | 207 | 2 | ✓ | |
| adult | 48,842 | 14 | 2 | ✓ | |
| League-of-Legends-Diamond | 48,651 | 14 | 2 | | |
| tamilnadu-electricity | 45,781 | 2 | 20 | | |
| bank-marketing | 45,211 | 16 | 2 | ✓ | |
| meta_stream_intervals | 45,164 | 74 | 11 | | |
| jungle_chess | 44,819 | 6 | 3 | ✓ | |
| Dynamically-Generated-Hate-Speech-Dataset | 41,144 | 8 | 2 | | ✓ |
| Breast-cancer-prediction | 39,998 | 11 | 2 | | |
| Click_prediction_small | 39,948 | 11 | 2 | ✓ | |
| Hotel-Reviews | 38,932 | 3 | 2 | | ✓ |
| electricity | 38,474 | 8 | 2 | ✓ | |
| nomao | 34,465 | 118 | 2 | ✓ | |
| Employee-Turnover-at-TECHCO | 34,452 | 9 | 2 | | |
| Amazon_employee_access | 32,769 | 9 | 2 | ✓ | |
| Credit-Risk-Dataset | 32,581 | 11 | 2 | | |
| Default-of-Credit-Card-Clients-Dataset | 30,000 | 23 | 2 | ✓ | |
| funpedia | 29,819 | 3 | 3 | | ✓ |
| credit_risk_china | 27,522 | 27 | 5 | | |
| Insurance | 23,548 | 10 | 2 | | |
| guillermo | 20,000 | 4,281 | 2 | ✓ | |
| riccardo | 20,000 | 4,283 | 2 | ✓ | |
| insurance_dataset | 20,000 | 26 | 4 | | |
| letter | 20,000 | 16 | 26 | | |
| game-of-thrones-script-all-seasons | 16,825 | 5 | 43 | | ✓ |

Table 9: The 253 Classification Datasets of the Pretraining Corpus.

| Dataset | $n$ | $m$ | $C$ | B | T |
|---|---|---|---|---|---|
| NewspaperChurn | 15,855 | 16 | 2 | | ✓ |
| mozilla4 | 15,545 | 5 | 2 | | |
| pol | 15,000 | 26 | 11 | ✓ | |
| eeg-eye-state | 14,980 | 14 | 2 | | |
| MagicTelescope | 13,376 | 10 | 2 | ✓ | |
| nursery | 12,958 | 8 | 4 | | |
| online-shoppers-intention | 12,330 | 17 | 2 | | |
| Disaster-Tweets | 11,370 | 4 | 2 | | ✓ |
| mammography | 11,183 | 6 | 2 | | |
| PhishingWebsites | 11,055 | 30 | 2 | ✓ | |
| Binary-Dataset-of-Phishing-and-Legitimate-URLs | 11,000 | 14 | 2 | | |
| pendigits | 10,992 | 16 | 10 | | |
| WBCAtt | 10,298 | 11 | 5 | | |
| artificial-characters | 10,218 | 7 | 10 | ✓ | |
| internet_usage | 10,108 | 71 | 46 | | ✓ |
| robert | 10,000 | 7,200 | 10 | ✓ | |
| dilbert | 10,000 | 2,000 | 5 | ✓ | |
| shrutime | 10,000 | 10 | 2 | | |
| JapaneseVowels | 9,961 | 14 | 9 | | |
| GesturePhaseSegmentationProcessed | 9,873 | 32 | 5 | ✓ | |
| FICO-HELOC-cleaned | 9,871 | 23 | 2 | ✓ | |
| IBRD_Loans_Classification | 9,215 | 6 | 10 | | |
| Indian_pines | 9,144 | 220 | 8 | | |
| SpeedDating | 8,378 | 120 | 2 | ✓ | ✓ |
| fabert | 8,237 | 795 | 7 | ✓ | |
| mushroom | 8,124 | 21 | 2 | | |
| isolet | 7,797 | 617 | 26 | | |
| eye_movements | 7,608 | 23 | 2 | ✓ | |
| twonorm | 7,400 | 20 | 2 | | |
| blastchar | 7,043 | 19 | 2 | | |
| musk | 6,598 | 167 | 2 | | |
| first-order-theorem-proving | 6,118 | 51 | 6 | ✓ | |
| HMEQ_Data | 5,960 | 12 | 2 | | |
| philippine | 5,832 | 308 | 2 | ✓ | |
| optdigits | 5,620 | 62 | 10 | | |
| BachChoralHarmony | 5,586 | 15 | 68 | | |
| page-blocks | 5,473 | 10 | 5 | | |
| wall-robot-navigation | 5,456 | 24 | 4 | | |
| christine | 5,418 | 1,611 | 2 | ✓ | |
| phoneme | 5,404 | 5 | 2 | ✓ | |
| Is_fraud | 5,227 | 19 | 2 | | ✓ |
| sylvine | 5,124 | 20 | 2 | ✓ | |
| Satellite | 5,100 | 36 | 2 | ✓ | |
| Multiclass_Classification_for_Corporate_Credit | 5,000 | 7 | 10 | | |
| Personal-Loan-Modeling | 5,000 | 12 | 2 | | |
| churn | 5,000 | 20 | 2 | ✓ | |
| waveform-5000 | 5,000 | 40 | 3 | | |
| air-quality-and-pollution-assessment | 5,000 | 9 | 4 | | |
| Heart_Failure_Prediction | 5,000 | 12 | 2 | | |
| compas-two-years | 4,966 | 11 | 2 | ✓ | |
| wine-quality-white | 4,898 | 11 | 7 | ✓ | |
| wilt | 4,839 | 5 | 2 | ✓ | |
| spambase | 4,601 | 57 | 2 | | |
| StackOverflow-polarity | 4,423 | 1 | 3 | | ✓ |

Table 9: The 253 Classification Datasets of the Pretraining Corpus.

| Dataset | $n$ | $m$ | $C$ | B | T |
|---|---|---|---|---|---|
| hiva_agnostic | 4,229 | 1,617 | 2 | | |
| Fraud-Detection-Updated | 4,156 | 27 | 2 | | |
| ada | 4,147 | 46 | 2 | ✓ | |
| analcatdata_supreme | 4,052 | 7 | 10 | | |
| hypothyroid | 3,770 | 27 | 3 | | |
| Bioresponse | 3,751 | 1,776 | 2 | ✓ | |
| Internet-Advertisements | 3,279 | 1,558 | 2 | ✓ | |
| led24 | 3,200 | 24 | 10 | | |
| kr-vs-kp | 3,196 | 36 | 2 | ✓ | |
| splice | 3,190 | 60 | 3 | ✓ | |
| dna | 3,186 | 180 | 3 | ✓ | |
| gina | 3,153 | 970 | 2 | ✓ | |
| madeline | 3,140 | 259 | 2 | ✓ | |
| jasmine | 2,984 | 144 | 2 | ✓ | |
| cjs | 2,796 | 29 | 6 | | |
| madelon | 2,600 | 500 | 2 | | |
| ozone-level-8hr | 2,534 | 72 | 2 | ✓ | |
| segment | 2,310 | 16 | 7 | ✓ | |
| cardiotocography | 2,126 | 23 | 10 | | |
| Estimation_of_Obesity_Levels | 2,111 | 16 | 7 | | |
| kc1 | 2,109 | 21 | 2 | ✓ | |
| Corporate-Credit-Rating | 2,026 | 30 | 8 | | ✓ |
| mfeat-factors | 2,000 | 216 | 10 | ✓ | |
| South_Asian_Churn_dataset | 2,000 | 13 | 2 | | |
| mfeat-zernike | 2,000 | 47 | 10 | ✓ | |
| mfeat-fourier | 2,000 | 76 | 10 | ✓ | |
| pbcseq | 1,945 | 18 | 3 | | |
| steel-plates-fault | 1,941 | 27 | 7 | ✓ | |
| car | 1,728 | 6 | 4 | ✓ | |
| GAMETES_Heterogeneity | 1,600 | 20 | 2 | | |
| one-hundred-plants-texture | 1,599 | 64 | 100 | ✓ | |
| audit-data | 1,552 | 35 | 2 | | |
| OVA_Breast | 1,545 | 10,935 | 2 | | |
| amazon-commerce-reviews | 1,500 | 10,000 | 50 | ✓ | |
| yeast | 1,484 | 8 | 10 | ✓ | |
| cmc | 1,473 | 9 | 3 | ✓ | |
| ibm-employee-attrition | 1,470 | 31 | 2 | | |
| pc4 | 1,458 | 37 | 2 | ✓ | |
| Data_Science_Nigeria_Telecoms_Churn | 1,400 | 14 | 2 | | |
| hepatitis_c_virus_hcv_for_egyptian_patients | 1,385 | 28 | 4 | | |
| Bank-Note-Authentication-UCI | 1,372 | 4 | 2 | | |
| baseball | 1,340 | 16 | 3 | | |
| Titanic | 1,309 | 13 | 2 | | ✓ |
| mental-health-in-tech-survey | 1,259 | 26 | 2 | | ✓ |
| hill-valley | 1,212 | 100 | 2 | | |
| Heart-Disease-Dataset-(Comprehensive) | 1,190 | 11 | 2 | ✓ | |
| volcanoes-e1 | 1,183 | 3 | 5 | | |
| Airlines-Tweets-Sentiments | 1,097 | 1 | 3 | | ✓ |
| MiceProtein | 1,080 | 77 | 8 | | |
| cnae-9 | 1,080 | 856 | 9 | ✓ | |
| solar_flare | 1,058 | 9 | 5 | ✓ | |
| qsar-biodeg | 1,055 | 41 | 2 | ✓ | |
| SOCC | 1,043 | 13 | 4 | | ✓ |
| rmftsa_sleepdata | 1,024 | 2 | 4 | | |

Continued on next page

Table 9: The 253 Classification Datasets of the Pretraining Corpus.

| Dataset | $n$ | $m$ | $C$ | B | T |
|---|---|---|---|---|---|
| autoUniv-au1-1000 | 1,000 | 20 | 2 | | |
| collins | 1,000 | 19 | 30 | | |
| credit-g | 1,000 | 20 | 2 | ✓ | |
| vowel | 990 | 12 | 11 | | |
| The-Estonia-Disaster-Passenger-List | 989 | 6 | 2 | | ✓ |
| xd6 | 973 | 9 | 2 | | |
| tokyo1 | 959 | 42 | 2 | | |
| tic-tac-toe | 958 | 9 | 2 | | |
| Tour-and-Travels-Customer-Churn-Prediction | 954 | 6 | 2 | | |
| acp-breast-cancer | 949 | 1 | 4 | | ✓ |
| oil_spill | 937 | 48 | 2 | | |
| anneal | 898 | 18 | 5 | | |
| Cervical_Cancer_Risk_Factors | 858 | 30 | 5 | | |
| vehicle | 846 | 18 | 4 | ✓ | |
| analcatdata_authorship | 841 | 70 | 4 | | |
| glioma_grading_clinical_and_mutation_features | 839 | 23 | 2 | | |
| analcatdata_dmft | 797 | 4 | 6 | | |
| regensburg_pediatric_appendicitis | 780 | 55 | 3 | | |
| QSAR_Bioconcentration_classification | 779 | 12 | 3 | | ✓ |
| Diabetes_Dataset | 768 | 8 | 2 | | |
| blood-transfusion-service-center | 748 | 4 | 2 | ✓ | |
| eucalyptus | 736 | 19 | 5 | ✓ | |
| breast-w | 699 | 9 | 2 | | |
| Australian | 690 | 14 | 2 | ✓ | |
| soybean | 683 | 35 | 19 | | |
| profb | 672 | 8 | 2 | ✓ | |
| Student_Performance | 666 | 11 | 4 | | |
| balance-scale | 625 | 4 | 3 | ✓ | |
| Loan-Predication | 614 | 11 | 2 | | |
| monks-problems-2 | 601 | 6 | 2 | ✓ | |
| synthetic_control | 600 | 60 | 6 | | |
| ilpd | 583 | 10 | 2 | | |
| micro-mass | 571 | 1,082 | 20 | ✓ | |
| wdbc | 569 | 30 | 2 | | |
| arsenic-male-lung | 559 | 4 | 2 | | |
| cylinder-bands | 540 | 34 | 2 | | |
| climate-model-simulation-crashes | 540 | 18 | 2 | | |
| water-treatment | 527 | 36 | 2 | | |
| Early-Stage-Diabetes-Risk-Prediction-Dataset | 520 | 16 | 2 | | |
| dresses-sales | 500 | 12 | 2 | | |
| irish | 500 | 5 | 2 | | |
| arrhythmia | 443 | 262 | 10 | | |
| wholesale-customers | 440 | 7 | 2 | | |
| vote | 435 | 16 | 2 | | |
| cars | 406 | 7 | 3 | | |
| chronic-kidney-disease | 400 | 25 | 2 | | |
| differentiated_thyroid_cancer_recurrence | 383 | 16 | 2 | | |
| colic | 368 | 26 | 2 | ✓ | |
| breast-cancer | 286 | 9 | 2 | | |
| qualitative-bankruptcy | 250 | 6 | 2 | | |
| us-2020-presidential-election-speeches | 245 | 5 | 7 | | ✓ |
| audiology | 192 | 57 | 8 | ✓ | |
| bone_marrow_transplant_children | 187 | 36 | 2 | | |
| darwin | 174 | 450 | 2 | | |

Table 9: The 253 Classification Datasets of the Pretraining Corpus.

| Dataset | $n$ | $m$ | $C$ | B | T |
|---|---|---|---|---|---|
| tae | 151 | 5 | 3 | | |
| EgyptianSkulls | 150 | 4 | 5 | | |
| lymph | 148 | 18 | 3 | ✓ | |
| arcene | 100 | 9,920 | 2 | ✓ | |

Table 10: The 93 Regression Datasets of the Pretraining Corpus, with their $n$ examples, $m$ features, presence in a benchmark ($B$) and whether they are textual ($T$).

| Dataset | $n$ | $m$ | B | T |
|---|---|---|---|---|
| delays_zurich_transport | 5,465,575 | 14 | ✓ | |
| New-York-Citi-Bike-Trip | 4,500,000 | 7 | | |
| USA-Airport-Dataset | 3,606,803 | 14 | | ✓ |
| New-York-Taxi-Trip | 2,083,778 | 21 | | |
| Buzzinsocialmedia_Twitter | 583,250 | 77 | ✓ | |
| nyc-taxi-green-dec-2016 | 581,835 | 18 | ✓ | |
| 515K-Hotel-Reviews-Data-in-Europe | 515,738 | 16 | | ✓ |
| dionis | 416,188 | 54 | ✓ | |
| Yolanda | 400,000 | 100 | ✓ | |
| Allstate_Claims_Severity | 188,318 | 130 | ✓ | |
| Football_players_Fifa_stats | 183,142 | 37 | | |
| black_friday | 166,821 | 9 | ✓ | |
| medical_charges | 163,065 | 3 | ✓ | |
| football-manager-data | 159,541 | 87 | | ✓ |
| wave_energy | 72,000 | 32 | ✓ | |
| video_transcoding | 68,784 | 18 | ✓ | |
| dating_profile | 59,946 | 30 | | ✓ |
| diamonds | 53,940 | 9 | ✓ | |
| sarcos | 48,933 | 21 | ✓ | |
| physiochemical_protein | 45,730 | 9 | ✓ | |
| fried | 40,768 | 10 | | |
| 2dplanes | 40,768 | 10 | | |
| mv | 40,768 | 10 | | |
| Perth-House-Prices | 33,656 | 17 | | ✓ |
| cps88wages | 28,155 | 6 | ✓ | |
| fps_benchmark | 24,624 | 39 | ✓ | |
| news_popularity2 | 24,007 | 4 | | ✓ |
| house_16H | 22,784 | 16 | ✓ | |
| health_insurance | 22,272 | 11 | ✓ | |
| house_sales | 21,613 | 21 | ✓ | |
| superconductivity | 21,263 | 81 | ✓ | |
| california_housing | 20,640 | 8 | ✓ | |
| avocado-sales | 18,249 | 11 | | |
| Bike_Sharing_Demand | 17,379 | 12 | ✓ | |
| elevators | 16,599 | 18 | ✓ | |
| FIFA20-Players | 14,999 | 72 | | ✓ |
| miami_housing | 13,932 | 15 | ✓ | |
| naval_propulsion_plant | 11,934 | 14 | ✓ | |
| Brazilian_houses | 10,692 | 11 | ✓ | |
| German-House-Prices | 10,552 | 24 | | ✓ |
| sulfur | 10,081 | 5 | ✓ | |
| climate_change_impact | 10,000 | 14 | | |
| grid_stability | 10,000 | 12 | ✓ | |

Continued on next page

Table 10: The 93 Regression Datasets of the Pretraining Corpus.

| Dataset | $n$ | $m$ | B | T |
|---|---|---|---|---|
| Credit-Card-Dataset-for-Clustering | 8,949 | 16 | | |
| topo_2_1 | 8,885 | 261 | ✓ | |
| yprop_4_1 | 8,885 | 212 | ✓ | |
| seoul_bike_sharing_demand_cat | 8,760 | 13 | | |
| pumadyn32nh | 8,192 | 32 | ✓ | |
| kin8nm | 8,192 | 8 | ✓ | |
| cpu_activity | 8,192 | 21 | ✓ | |
| bank32nh | 8,192 | 32 | | |
| Pollen-Luxembourg-1992-2018 | 7,784 | 36 | | |
| colleges | 7,063 | 44 | ✓ | ✓ |
| wind | 6,574 | 14 | | |
| QSAR-TID-10980 | 5,766 | 1,024 | ✓ | |
| QSAR-TID-11 | 5,742 | 1,024 | ✓ | |
| Myanmar-Air-Quality | 5,122 | 10 | | |
| Santander_transaction_value | 4,459 | 4,735 | ✓ | |
| SAT11-HAND-runtime-regression | 4,440 | 114 | ✓ | |
| Mercedes_Benz_Greener_Manufacturing | 4,209 | 364 | ✓ | |
| abalone | 4,177 | 8 | ✓ | |
| pollen | 3,848 | 4 | | |
| space_ga | 3,107 | 6 | ✓ | |
| scotch-whiskey-reviews-update-2020 | 2,247 | 4 | | ✓ |
| quake | 2,178 | 3 | ✓ | |
| auction_verification | 2,043 | 7 | ✓ | |
| us_crime | 1,994 | 126 | ✓ | |
| airfoil_self_noise | 1,503 | 5 | ✓ | |
| house_prices | 1,460 | 80 | | |
| house_prices_nominal | 1,460 | 79 | ✓ | |
| NBA-PLAYERS–2016-2019 | 1,408 | 43 | | ✓ |
| Insurance-Premium-Data | 1,338 | 6 | | |
| Moneyball | 1,232 | 14 | ✓ | |
| socmob | 1,156 | 5 | ✓ | |
| MIP-2016-regression | 1,090 | 116 | ✓ | |
| geographical_origin_of_music | 1,059 | 116 | ✓ | |
| concrete_compressive_strength | 1,030 | 8 | ✓ | |
| Household-monthly-electricity-bill | 1,000 | 9 | | |
| stock | 950 | 9 | | |
| QSAR_fish_toxicity | 908 | 6 | ✓ | |
| cars | 804 | 17 | ✓ | |
| energy_efficiency | 768 | 8 | ✓ | |
| kdd_el_nino-small | 709 | 8 | | |
| student_performance_por | 649 | 30 | ✓ | |
| strikes | 625 | 6 | | |
| sensory | 576 | 11 | ✓ | |
| meta | 528 | 21 | | |
| forest_fires | 517 | 12 | ✓ | |
| rmftsa_ladata | 508 | 10 | | |
| boston | 506 | 13 | ✓ | |
| no2 | 500 | 7 | | |
| Diabetes(scikit-learn) | 442 | 10 | | |
| NBA-2k20-player-dataset | 439 | 14 | | ✓ |
| baseball-hitter | 263 | 22 | | |
| bodyfat | 252 | 14 | | |
| Lisbon-House-Prices | 246 | 13 | | |
| tecator | 240 | 124 | ✓ | |

## D   Benchmark Datasets

This appendix elaborates on the benchmark presented in §4. We consider all datasets proposed by *AutoML Multimodal Benchmark* (SHI) [63], *Vectorizing* (VEC) [32], and *CARTE-Benchmark* (CRT) [47], resulting in a final set of 50 datasets. We deduplicate datasets that appear as-is in more than one benchmark. In addition, since CARTE explores the concept of multi-table learning, they introduce highly-overlapping datasets for which we remove one variant (see 4.3 and B.2 in their paper).

Table 11 presents the classification datasets and Table 12 the regression ones. Each table includes an internal *ID* used for reference, the *Dataset* name, the number of examples $n$ and of features $m$, and the number of classes $C$ for classification.[26] Finally, we also indicate the benchmark sources where each dataset appears. In addition, Table 13 presents the full benchmark with a short description per dataset, and Table 14 details the datasets removed during the deduplication process. Most of the excluded datasets are regression datasets from the *CARTE-Benchmark*, because of its high-overlapping nature.

Table 11: The 14 classification datasets of the benchmark, with their $n$ examples, $m$ features, $C$ classes, and presence in the SHI, VEC and CRT benchmarks.

| ID | Dataset | $n$ | $m$ | $C$ | SHI | VEC | CRT |
|----|---------|-----|-----|-----|-----|-----|-----|
| C01 | women_clothing_review | 18,788 | 10 | 5 | ✓ | | |
| C02 | us-accidents | 7,728,394 | 42 | 4 | | ✓ | ✓ |
| C03 | data_scientist_salary | 15,841 | 6 | 6 | ✓ | | |
| C04 | imdb_genre_prediction | 800 | 11 | 2 | ✓ | | |
| C05 | product_sentiment_machine_hack | 5,091 | 2 | 4 | ✓ | | |
| C06 | google_qa_question_type_reason | 4,863 | 39 | 5 | ✓ | | |
| C07 | michelin-guide-restaurants-2021 | 17,735 | 11 | 5 | | | ✓ |
| C08 | fake_job_postings2 | 12,725 | 5 | 2 | ✓ | | |
| C09 | jigsaw_unintended_bias100K | 100,000 | 40 | 2 | ✓ | | |
| C10 | yelp-reviews-dataset | 10,000 | 5 | 5 | | | ✓ |
| C11 | news_channel | 20,284 | 17 | 6 | ✓ | | |
| C12 | wine_reviews | 84,123 | 5 | 30 | ✓ | ✓ | ✓ |
| C13 | kick_starter_funding | 86,502 | 9 | 2 | ✓ | | |
| C14 | melbourne_airbnb | 18,316 | 89 | 10 | ✓ | | |

## E   Baselines

In this appendix we first discuss models excluded from the evaluation due to data leakage concerns, and then cover implementation details for the baselines used in our main experiments §4.

### E.1   Excluded Baselines

As opposed to GDBTs or single-dataset deep learning methods, evaluating pretrained tabular models introduces additional complexity. Indeed, leakage can come in multiple forms. When LLMs are involved, there is a risk of memorization [9], and models trained on synthetic datasets [38] which try to mimic real-world distributions, can be unintentionally biased towards popular benchmarks.

While these two forms of leakage are subtle and hard to detect, a more direct form must be strictly avoided: When the same dataset (or a variant of it) is used during pretraining, and then it is evaluated as a downstream task. In such scenario there is inevitable severe data leakage, especially when running with multiple random test splits. The rest of the section explains how both *TP-BERTa* [84] and *CM2* [86] suffer from such contamination with respect to our benchmark. As we briefly mention in §7, we advocate for improving TFM research by encouraging models that are practical to evaluate, by releasing several versions of each model, and providing default hyperparameters.

**TP-BERTa**   We exclude TP-BERTa from our evaluation for two key reasons. First, their implementation assumes that every example is treated as a serialized single sequence, which allows for

---

[26]We treat ranking problems with up to 10 discrete values as multiclass problems.

Table 12: The 36 regression datasets of the benchmark, with their $n$ examples, $m$ features, and presence in the SHI, VEC and CRT benchmarks.

| ID | Dataset | $n$ | $m$ | SHI | VEC | CRT |
|----|---------|-----|-----|-----|-----|-----|
| R01 | used-cars-dataset-cardekho | 37,814 | 112 | | | ✓ |
| R02 | second-hand-mercedes-benz | 16,392 | 7 | | | ✓ |
| R03 | animeplanet-recommendation | 14,391 | 14 | | | ✓ |
| R04 | ML/DS-Salaries | 119,628 | 9 | | | ✓ |
| R05 | Babies-R-Us | 5,085 | 12 | | | ✓ |
| R06 | employee_salaries | 9,228 | 11 | | ✓ | ✓ |
| R07 | spotify-tracks-dataset | 114,000 | 18 | | ✓ | |
| R08 | california_house_price | 37,951 | 39 | ✓ | | |
| R09 | fifa | 19,178 | 28 | | | ✓ |
| R10 | coffee-scrap-coffeereview | 2,440 | 17 | | | ✓ |
| R11 | BikeWale | 9,003 | 6 | | ✓ | ✓ |
| R12 | used-car-prices-in-pakistan | 72,655 | 9 | | | ✓ |
| R13 | bookprice_prediction | 4,989 | 8 | ✓ | | |
| R14 | ae_price_prediction | 22,662 | 12 | ✓ | | |
| R15 | Employee-remuneration | 44,574 | 5 | | ✓ | ✓ |
| R16 | filmtv-movies-dataset | 41,399 | 17 | | | ✓ |
| R17 | free-7-million-company-dataset | 7,173,426 | 7 | | ✓ | ✓ |
| R18 | museums | 22,290 | 21 | | | ✓ |
| R19 | vivino-wine-data | 8,650 | 6 | | | ✓ |
| R20 | wikiliq-dataset | 12,569 | 12 | | | ✓ |
| R21 | beer-profile-and-ratings | 3,197 | 24 | | | ✓ |
| R22 | korean-drama | 1,647 | 9 | | | ✓ |
| R23 | videogamesales | 16,598 | 5 | | | ✓ |
| R24 | zomato-bangalore-restaurants | 41,665 | 15 | | ✓ | ✓ |
| R25 | the-movies-dataset | 45,460 | 20 | | | ✓ |
| R26 | nba-draft-basketball | 1,669 | 22 | | | ✓ |
| R27 | Goodreads | 3,967 | 14 | | ✓ | |
| R28 | Rotten-Tomatoes | 7,158 | 15 | | | ✓ |
| R29 | saudi-arabia-used-cars-dataset | 8,035 | 12 | | | ✓ |
| R30 | top-ramen-ratings-2022 | 4,105 | 4 | | ✓ | ✓ |
| R31 | Journal-Score-SJR | 31,136 | 21 | | | ✓ |
| R32 | chocolate-bar-ratings | 1,795 | 8 | | | ✓ |
| R33 | mercari_price_suggestion100K | 100,000 | 9 | ✓ | | |
| R34 | wine-price-on-polish-market | 2,247 | 18 | | | ✓ |
| R35 | clear-corpus | 4,724 | 30 | | ✓ | ✓ |
| R36 | jc_penney_products | 10,860 | 5 | ✓ | | |

a maximum length of 512 tokens, as elaborated in Appendix A.1. While this decision is efficient for datasets with a low amount of features and no free-text presence, around half of the datasets in our benchmark are too long for that limitation, as they either contain too many features or long free-texts. Second, TP-BERTa's pretraining uses datasets that appear in our evaluation set, as listed in Table 6 of their paper [84]. It is evident that several datasets overlap directly with datasets in our benchmark §4 (e.g., *1510_fifa*, *1368_IMDb-Ratings*, *1639_Melbourne*), disqualifying them for our purposes. Furthermore, we observe a concerning overlap between their pretraining and downstream task datasets (e.g., *airlines*, *sf police* and *diabetes*). We believe that this questions the validity of their evaluation, and that such contamination poses a serious challenge for the TFM community which could be substantially addressed by better tabular data repositories [76].

**CM2** CM2 was pretrained over *OpenTabs*, a compilation of more than 2,000 datasets drawn from public tabular data repositories, including OpenML and Kaggle. While this collection is valuable, pretraining a model over these datasets compromises further evaluation of any of them. Naturally, the overlap with our benchmark here is extremely high, making it infeasible to use as a

Table 13: Benchmark Datasets Description

| ID | Description |
|---|---|
| C01 | Women Clothing E-Commerce Reviews |
| C02 | US Accidents between 2016 and 2023 |
| C03 | Indian Data Scientist Salary Prediction |
| C04 | IMDB Movies Genre Prediction |
| C05 | Product Sentiment Analysis |
| C06 | Google QA Question Type Reason Explanation |
| C07 | Michelin Guide Restaurants Awards |
| C08 | Fake Job Posting Detection |
| C09 | Online Social Media Comments Toxicity |
| C10 | YELP Dataset Reviews |
| C11 | News Channel Prediction |
| C12 | Wine Reviews for Variety Prediction |
| C13 | Kickstarter Funding Prediction |
| C14 | Melbourne AirBnB Listings |
| R01 | User cars and listing price in the website Cardekho |
| R02 | Second-hand cars Mercedes Benz price Italy |
| R03 | Anime-Planet Recommendation Database 2020 |
| R04 | Salaries of ML/DS Professionals Worldwide |
| R05 | Prices Prediction for baby product from Babies R Us website |
| R06 | Employee Salary in Montgomery County, MD |
| R07 | Spotify Tracks Popularity |
| R08 | California Houses 2020 Prices |
| R09 | FIFA 2022 Players Wages |
| R10 | Coffee Review Rating |
| R11 | Bike and scooters from bikewale website in India |
| R12 | Used car prices in Pakistan 2021 |
| R13 | Book Price Prediction |
| R14 | American Eagle Retailer Price Prediction |
| R15 | Employee Remuneration and Expenses - Vancouver |
| R16 | FilmTV movies ataset rating |
| R17 | Company size prediction |
| R18 | General information on the US museums |
| R19 | Vivino Spanish Wine Data |
| R20 | WikiliQ - Alcohol dataset (May, 2022) |
| R21 | Tasting profiles and consumer reviews for beers |
| R22 | Korean Dramas |
| R23 | Video Games Sales |
| R24 | Zomato Restaurants in Bengaluru |
| R25 | Metadata of movies released until 2017 for box-office revenues |
| R26 | NBA Draft Basketball Player Data 1989-2021 |
| R27 | Books ratings |
| R28 | Rotten Tomatoes Movie Ratings |
| R29 | Saudi Arabia Used Cars Price from Syarah Website |
| R30 | Ramen Ratings |
| R31 | Academic impact for Scientific Journals |
| R32 | Chocolate Bar expert ratings |
| R33 | Mercari Online Marketplace Product Prices |
| R34 | Information about wines on the polish market |
| R35 | Readability scores for text passages spanning various genres and time periods |
| R36 | JC Penney Product Prices in Retailer Website |

Table 14: Excluded datasets, with their benchmark origin and reason for removal: (1) *Duplicate* dataset, (2) *Unavailable*, for datasets with inconvenient or unavailable hosting outside tabular repositories, and (3) *Pretraining*, for two (regression) datasets mistakenly used for the pretraining.

| Dataset | Benchmark | Reason | Duplicate |
|---|---|---|---|
| google_qa_answer_type | SHI | Duplicate | google_qa_question_type |
| news_popularity2 | SHI | Pretraining | |
| US Presidential | VEC | Unavailable | |
| Journal Influence | VEC | Duplicate | Journal-Score-SJR |
| Buy Buy Baby | CRT | Duplicate | Babies-R-Us |
| Bikedekho | CRT | Duplicate | BikeWale |
| Journal Score JCR | CRT | Duplicate | Journal-Score-SJR |
| Japanese Anime | CRT | Duplicate | animeplanet-recommendation |
| Mydramalist | CRT | Duplicate | korean-drama |
| Prescription Drugs | CRT | Unavailable | |
| Roger Ebert | CRT | Unavailable | |
| US Presidential | CRT | Unavailable | |
| Used Cars 24 | CRT | Duplicate | used-car-prices-in-pakistan |
| Whisky | CRT | Pretraining | |
| Wine.com | CRT | Duplicate | wine_reviews |
| WineEnthusiasts | CRT | Duplicate | wine_reviews |

baseline. Interestingly, their repository[27] lists TP-BERTa as a method trained on a subset of OpenTabs, reinforcing that the leakage is shared between the models.

## E.2 Baselines Implementation and Hyperparameters

This section outlines the implementation and hyperparameter tuning strategy used for the baselines reported in §4. While each baseline has its own model-specific preprocessing pipeline, we apply two shared preprocessing steps to both TabSTAR (as detailed in Appendix A.1) and all the baselines: (1) We perform date preprocessing by using skrub's *DatetimeEncoder*, and (2) Apply a clipped z-score transformation for target variables in regression datasets.

**Textual Feature Handling** CARTE natively supports textual inputs, and the TabPFN-v2 API client[28] does as well, although its implementation details remain undisclosed. As all other baselines do not natively support free-text features,[29] we preprocess these features into fixed-size embeddings using *skrub*'s *TextEncoder*, which internally applies a frozen *e5-small-v2* encoder to each semantic column. This aligns with the encoder used in TabSTAR, enabling a fair comparison across models. There are, however, two key differences in how TabSTAR handles these embeddings: First, the embeddings are specifically finetuned for the task, contributing significantly to its strong performance as shown in §5 and further analyzed in §G.2. The second detail is that skrub applies dimensionality reduction to 30 dimensions, as proposed by [32]. This compressed representation performs comparably to the full embedding space, while offering improved inference efficiency.

**Hyperparameter Tuning** For the tuned models we use the *Optuna* package[30] with random search, with a budget of 4 hours for every run and parallelizing trials on 8 CPU cores. We use 5-fold cross-validation and take the best configuration selected based on this mean score. We use it then to retrain the model on the full training data.

---

[27]https://github.com/Chao-Ye/CM2

[28]https://github.com/PriorLabs/tabpfn-client

[29]CatBoost includes a built-in text module, but it underperforms compared to dense text embeddings.

[30]https://pypi.org/project/optuna/

### E.2.1  TabPFN-v2

We run TabPFN-v2 using their API client which supports text features.[31] While the intrinsic details of their textual handling remain undocumented, it's reasonable to assume that it resembles the processing we apply to GBDTs, as their model leverages ICL and their architecture has no textual encoder.

### E.2.2  TabICL

We run TabICL using its package[32] with the default configuration. Since TabICL can only be evaluated on classification tasks, we exclude it from the regression analysis.

### E.2.3  TabDPT

We run TabDPT using its package[33] with the default configuration. While it achieves state-of-the-art performance on some regression tasks, we encounter severe performance issues on others. Due to this inconsistency, we exclude TabDPT from the reported results and are in contact with the authors to investigate potential implementation issues.

### E.2.4  CARTE

We run CARTE using its package,[34] which inherently performs $k$-fold cross-validation. After consulting with the authors, we set $k = 5$ for efficiency instead of 10. We do grid search over their recommended learning rates,[35] and we take the best-performing variant per dataset split.

### E.2.5  RealMLP

We run RealMLP using its official implementation in the pytabkit package.[36] Following the advice of its authors, we disable label smoothing and optimize for *cross_entropy* for binary classification and *1-auc_ovr* for multiclass classification, and keep all other default hyperparameters. For the tuned version, we follow the hyperparameter search space suggested by [39] as detailed in Table 15.

Table 15: RealMLP-Tuned Hyperparameters Search Space

| Hyperparameter | Search Space |
|---|---|
| num_emb_type | Choice([None, PBLD, PL, PLR]) |
| add_front_scale | Choice([True, False], p=[0.6, 0.4]) |
| lr | $\log \mathcal{U}(2e-2, 3e-1)$ |
| p_drop | Choice([0.0, 0.15, 0.3], p=[0.3, 0.5, 0.2]) |
| act | Choice([ReLU, SELU, Mish]) |
| hidden_sizes | Choice([[256, 256, 256], [64, 64, 64, 64], [512]], p=[0.6, 0.2, 0.2]) |
| wd | Choice([0.0, 2e-2]) |
| plr_sigma | $\log \mathcal{U}(0.05, 0.5)$ if classification else 0.1 |

### E.2.6  CatBoost

We run CatBoost using the *catboost* package[37] and run the default configuration suggested by [26] by setting $early\_stopping\_rounds = 50$, $od\_pval = 0.001$, $iterations = 2000$. For the tuned version, we follow the hyperparameter search suggested by [38] as detailed in Table 16.

---

[31]We use v2.0.8, the latest version available at the time of running the experiments.

[32]https://pypi.org/project/tabicl/

[33]https://pypi.org/project/tabdpt/

[34]https://github.com/soda-inria/carte

[35]$\{2.5 \times 10^{-4},\ 5 \times 10^{-4},\ 7.5 \times 10^{-4},\ 2.5 \times 10^{-3},\ 5 \times 10^{-3},\ 7.5 \times 10^{-3}\}$

[36]https://pypi.org/project/pytabkit/

[37]https://pypi.org/project/catboost/

Table 16: CatBoost-Tuned Hyperparameters Search Space

| Hyperparameter | Search Space |
|---|---|
| learning_rate | $\log \mathcal{U}(e^{-5}, 1)$ |
| random_strength | $\mathcal{U}\{1, 2, \ldots, 20\}$ |
| l2_leaf_reg | $\log \mathcal{U}(1, 10)$ |
| bagging_temperature | $\mathcal{U}(0.0, 1.0)$ |
| leaf_estimation_iterations | $\mathcal{U}\{1, 2, \ldots, 20\}$ |
| iterations | $\mathcal{U}\{100, 101, \ldots, 4000\}$ |

### E.2.7 XGBoost

We run XGBoost using the *xgboost* package.[38] For the default configuration, we follow the suggestion of [26] and use: $booster = "gbtree"$, $early\_stopping\_rounds = 50$, $n\_estimators = 2000$. For the tuned variant, we follow the hyperparameter search space suggested by [38], as shown in Table 17.

Table 17: XGBoost-Tuned Hyperparameters Search Space

| Hyperparameter | Search Space |
|---|---|
| learning_rate | $\log \mathcal{U}(e^{-7}, 1)$ |
| max_depth | $\mathcal{U}\{1, 2, \ldots, 10\}$ |
| subsample | $\mathcal{U}(0.2, 1)$ |
| colsample_bytree | $\mathcal{U}(0.2, 1)$ |
| colsample_bylevel | $\mathcal{U}(0.2, 1)$ |
| min_child_weight | $\log \mathcal{U}(e^{-16}, e^5)$ |
| alpha | $\log \mathcal{U}(e^{-16}, e^2)$ |
| reg_lambda | $\log \mathcal{U}(e^{-16}, e^2)$ |
| gamma | $\log \mathcal{U}(e^{-16}, e^2)$ |
| n_estimators | $\mathcal{U}\{100, 101, \ldots, 4000\}$ |

### E.2.8 LightGBM

We run LightGBM using the *lightgbm* package.[39] and its default implementation. For the tuned variant, we follow the hyperparameter search space suggested by [38], as shown in Table 18.

Table 18: LightGBM-Tuned Hyperparameters Search Space

| Hyperparameter | Search Space |
|---|---|
| num_leaves | $\mathcal{U}\{5, 6, \ldots, 50\}$ |
| max_depth | $\mathcal{U}\{3, 4, \ldots, 20\}$ |
| learning_rate | $\log \mathcal{U}(e^{-3}, 1)$ |
| n_estimators | $\mathcal{U}\{50, 51, \ldots, 2000\}$ |
| min_child_weight | $\{10^{-5}, 10^{-3}, 10^{-2}, 10^{-1}, 1, 10, 10^2, 10^3, 10^4\}$ |
| subsample | $\mathcal{U}(0.2, 0.8)$ |
| colsample_bytree | $\mathcal{U}(0.2, 0.8)$ |
| reg_alpha | $\{0, 10^{-1}, 1, 2, 5, 7, 10, 50, 100\}$ |
| reg_lambda | $\{0, 10^{-1}, 1, 5, 10, 20, 50, 100\}$ |

### E.2.9 Random Forest

We treat Random Forest as a weak baseline to establish a lower-bound reference for each dataset split. We run it with the sklearn package [40] and use its default configuration with $n\_estimators = 100$.

---

[38]https://pypi.org/project/xgboost/

[39]https://pypi.org/project/lightgbm/

[40]https://scikit-learn.org/

Table 19: Classification performance per dataset (up to 10K). Best score before rounding is bolded. We report average AUROC with 95% CIs. Models: CARTE (CRT), CatBoost (CTB), TabICL (ICL), LightGBM (LGB), RealMLP (MLP), TabDPT (DPT), TabM (TBM), TabPFNv2 (PFN), RandomForest (RF), TabSTAR (STR), XGBoost (XGB). Tuned models are marked with a '+'.

| ID | CRT | CTB | CTB+ | DPT | ICL | LGB | LGB+ | MLP | MLP+ | PFN | RF | STR | TBM | TBM+ | XGB | XGB+ |
|---|---|---|---|---|---|---|---|---|---|---|---|---|---|---|---|---|
| C01 | 88.4 ±0.3 | 90.2 ±0.3 | 90.3 ±0.4 | 90.2 ±0.3 | 90.7 ±0.4 | 89.5 ±0.4 | 89.6 ±0.4 | 90.2 ±0.5 | 90.2 ±0.3 | 90.3 ±0.3 | 88.8 ±0.3 | **90.8** ±**0.3** | 90.5 ±0.3 | 90.5 ±0.3 | 89.3 ±0.4 | 90.2 ±0.4 |
| C02 | – | 97.3 ±0.5 | 97.4 ±0.4 | 92.3 ±0.6 | 95.2 ±0.4 | 97.2 ±0.5 | 97.3 ±0.5 | 96.4 ±0.5 | 96.6 ±0.5 | – | 96.3 ±0.5 | **97.9** ±**0.5** | 96.8 ±0.5 | 97.2 ±0.4 | 97.2 ±0.3 | 97.6 ±0.3 |
| C03 | 82.5 ±0.3 | 82.0 ±0.3 | 81.2 ±0.6 | **85.4** ±**0.3** | 81.4 ±0.3 | 79.8 ±0.3 | 79.9 ±0.3 | 82.6 ±0.4 | 82.7 ±0.3 | 82.4 ±0.3 | 77.3 ±0.3 | 83.0 ±0.3 | 83.3 ±0.3 | 83.8 ±0.3 | 80.5 ±0.3 | 82.1 ±0.3 |
| C04 | – | 84.5 ±1.9 | 85.4 ±1.8 | 83.3 ±2.8 | 86.9 ±2.3 | 83.0 ±2.3 | 82.8 ±2.1 | 82.4 ±3.1 | 82.8 ±2.2 | **88.3** ±**1.5** | 82.4 ±3.0 | 83.7 ±2.3 | 85.0 ±3.1 | 85.1 ±2.6 | 80.7 ±2.9 | 85.4 ±2.0 |
| C05 | 88.5 ±0.9 | 90.6 ±1.0 | 90.9 ±0.5 | 92.2 ±0.6 | **92.9** ±**0.5** | 88.2 ±1.3 | 88.4 ±0.6 | 91.0 ±0.8 | 91.4 ±0.6 | 91.2 ±0.7 | 88.0 ±1.1 | 91.3 ±0.8 | 91.1 ±0.7 | 90.7 ±0.6 | 88.1 ±1.1 | 90.3 ±0.6 |
| C06 | 80.9 ±1.5 | 81.3 ±1.3 | 82.6 ±1.1 | 73.5 ±2.2 | 74.3 ±1.8 | 81.9 ±1.1 | 82.8 ±0.9 | 75.6 ±1.4 | 77.5 ±0.5 | **87.7** ±**0.5** | 73.6 ±1.2 | 87.0 ±0.6 | 79.0 ±1.7 | 79.2 ±1.5 | 81.5 ±1.0 | 83.7 ±0.9 |
| C07 | 90.1 ±0.4 | 90.3 ±0.3 | 90.6 ±0.3 | 89.9 ±0.2 | 86.7 ±0.3 | 88.9 ±0.4 | 89.3 ±0.4 | 90.0 ±0.3 | 90.2 ±0.2 | 89.8 ±0.4 | 85.1 ±0.7 | **91.5** ±**0.3** | 91.1 ±0.3 | 91.1 ±0.3 | 87.2 ±0.6 | 89.3 ±0.5 |
| C08 | 90.8 ±1.6 | 93.0 ±0.9 | 93.2 ±1.0 | 93.2 ±1.4 | **95.1** ±**0.7** | 93.1 ±0.7 | 93.2 ±0.9 | 92.1 ±0.9 | 91.9 ±1.3 | 91.3 ±1.1 | 90.2 ±1.1 | 93.5 ±1.3 | 92.1 ±1.5 | 92.1 ±1.3 | 91.9 ±1.1 | 94.4 ±0.7 |
| C09 | – | 82.4 ±1.3 | 82.6 ±1.4 | 81.3 ±1.2 | 84.3 ±1.0 | 81.1 ±1.2 | 82.9 ±1.3 | 82.6 ±1.4 | 81.3 ±2.5 | 82.5 ±1.1 | 76.4 ±1.4 | **94.0** ±**0.8** | 81.8 ±4.1 | 83.6 ±1.0 | 79.4 ±1.6 | 83.2 ±1.2 |
| C10 | – | 86.7 ±0.2 | 87.0 ±0.2 | 87.2 ±0.2 | 87.7 ±0.2 | 85.7 ±0.3 | 85.9 ±0.3 | 87.7 ±0.2 | 87.6 ±0.2 | 87.6 ±0.4 | 84.6 ±0.2 | **89.1** ±**0.2** | 87.5 ±0.3 | 87.5 ±0.3 | 85.4 ±0.3 | 86.8 ±0.2 |
| C11 | 78.4 ±0.5 | 79.7 ±0.5 | 80.7 ±0.4 | 79.6 ±0.4 | **81.5** ±**0.5** | 78.9 ±0.5 | 79.2 ±0.4 | 80.4 ±0.4 | 79.6 ±0.5 | 81.4 ±0.4 | 76.2 ±0.6 | 79.2 ±0.5 | 81.2 ±0.3 | 81.5 ±0.3 | 78.2 ±0.6 | 80.2 ±0.5 |
| C12 | 96.0 ±0.2 | 97.5 ±0.1 | 97.7 ±0.1 | 93.8 ±0.3 | 97.8 ±0.1 | 96.7 ±0.2 | 96.7 ±0.1 | 97.6 ±0.1 | 97.6 ±0.1 | – | 95.3 ±0.2 | **98.3** ±**0.1** | 97.8 ±0.1 | 97.9 ±0.1 | 96.6 ±0.2 | 97.3 ±0.1 |
| C13 | 70.9 ±0.8 | 73.7 ±1.1 | 74.1 ±1.0 | 70.7 ±0.7 | 74.0 ±0.8 | 72.6 ±1.0 | 72.7 ±1.0 | 73.0 ±0.7 | 72.4 ±1.2 | 72.3 ±0.9 | 71.1 ±1.1 | **75.0** ±**0.7** | 74.3 ±0.6 | 74.4 ±0.7 | 70.2 ±0.8 | 74.1 ±1.1 |
| C14 | – | 83.5 ±0.3 | 84.0 ±0.5 | 82.4 ±0.3 | 80.4 ±0.3 | 81.9 ±0.4 | 82.3 ±0.4 | 81.9 ±0.5 | 83.2 ±0.4 | – | 80.1 ±0.3 | 84.0 ±0.3 | 84.8 ±0.2 | **85.4** ±**0.3** | 81.4 ±0.4 | 83.2 ±0.4 |

# F   Extended Main Results

In this appendix we provide the main results for the experiment as reported in §5. As elaborated in §4, each model is evaluated on each dataset across 10 splits. Since performance scales vary between datasets, we follow the normalization approach proposed by [38], rescaling all scores to the $[0, 1]$ range. The final reported performance for each model is the average over these normalized runs, and we compute 95% confidence intervals using the standard normal approximation: $\hat{\mu} \pm 1.96 \frac{\hat{\sigma}}{\sqrt{n}}$.

## F.1   Technical Limitations

As discussed in §5, TabPFN-v2 is unable to run on 4 datasets: C12, because it is a multiclass problem with more than 10 classes, and C02, C14 and R01 because they support inference for up to 500,000 cells. Attempts to run the model over a subset of the examples led to a significantly worse performance, and thus we decide not to report them to allow a fair comparison.

Additionally, CARTE is unable to run over 15 of the datasets in the benchmark due to a known bug[41] in their implementation for the *PowerTransformation*, which struggles in the presence of features with too less unique values. Furthermore, TabDPT's performance over a few regression datasets is especially low. Since we focus on classification tasks, we exclude this model's performance on this task. Finally, RealMLP, fails to run on R01 in the unlimited setting due to memory constraints.

## F.2   Dataset Level Performance

We report AUROC for classification and $R^2$ for regression, with 95% CIs computed over the 10 runs for each dataset. Tables  19 and 20 summarize classification performance on datasets with up to 10K and over 10K examples, respectively. Tables 21 and 22 to the same for regression tasks. For conciseness, datasets are referred by their ID from Appendix D.

---

[41]https://github.com/soda-inria/carte/issues/23

Table 20: Classification performance per dataset (above 10K). Best score before rounding is bolded. We report average AUROC with 95% CIs. Models: CARTE (CRT), CatBoost (CTB), TabICL (ICL), LightGBM (LGB), RealMLP (MLP), TabDPT (DPT), TabM (TBM), TabPFNv2 (PFN), RandomForest (RF), TabSTAR (STR), XGBoost (XGB). Unlimited models are marked with a '!'.

| ID | CRT | CTB | CTB! | DPT | ICL | ICL! | LGB | LGB! | MLP | MLP! | PFN | STR | STR! | TBM | TBM! | XGB | XGB! |
|---|---|---|---|---|---|---|---|---|---|---|---|---|---|---|---|---|---|
| C01 | 88.4 ±0.3 | 90.3 ±0.4 | 90.5 ±0.3 | 90.2 ±0.4 | 90.7 ±0.4 | 90.9 ±0.4 | 89.6 ±0.4 | 89.8 ±0.3 | 90.2 ±0.4 | 90.6 ±0.3 | 90.3 ±0.3 | 90.8 ±0.3 | **91.2** **±0.3** | 90.5 ±0.3 | 90.8 ±0.4 | 90.2 ±0.4 | 90.4 ±0.3 |
| C02 | – | 97.4 ±0.4 | 98.3 ±0.3 | 92.3 ±0.6 | 95.2 ±0.4 | 96.9 ±0.5 | 97.3 ±0.5 | 98.3 ±0.3 | 96.6 ±0.5 | 98.5 ±0.2 | – | 97.9 ±0.5 | 98.4 ±0.2 | 97.2 ±0.4 | **98.5** **±0.3** | 97.6 ±0.3 | 98.2 ±0.3 |
| C03 | 82.5 ±0.3 | 81.2 ±0.6 | 81.5 ±0.8 | **85.4** **±0.3** | 81.4 ±0.3 | 81.2 ±0.3 | 79.9 ±0.3 | 80.4 ±0.3 | 82.7 ±0.3 | 83.4 ±0.3 | 82.4 ±0.3 | 83.0 ±0.3 | 83.8 ±0.5 | 83.8 ±0.3 | 84.1 ±0.4 | 82.1 ±0.3 | 82.3 ±0.3 |
| C07 | 90.1 ±0.4 | 90.6 ±0.3 | 91.0 ±0.4 | 89.9 ±0.2 | 86.7 ±0.3 | 87.1 ±0.3 | 89.3 ±0.4 | 90.0 ±0.3 | 90.2 ±0.2 | 90.5 ±0.4 | 89.8 ±0.4 | 91.5 ±0.3 | **91.9** **±0.3** | 91.1 ±0.3 | 91.6 ±0.3 | 89.3 ±0.5 | 89.9 ±0.5 |
| C08 | 90.8 ±1.6 | 93.2 ±1.0 | 93.1 ±1.2 | 93.2 ±1.4 | 95.1 ±0.7 | **95.3** **±0.6** | 93.2 ±0.9 | 93.4 ±0.9 | 91.9 ±1.3 | 91.9 ±1.3 | 91.3 ±1.1 | 93.5 ±1.5 | 95.1 ±0.9 | 92.1 ±1.5 | 93.9 ±1.1 | 94.4 ±0.7 | 94.4 ±0.9 |
| C09 | – | 82.6 ±1.4 | 85.9 ±1.2 | 81.3 ±1.2 | 84.3 ±1.0 | 86.3 ±1.1 | 82.9 ±1.1 | 85.6 ±1.3 | 81.3 ±1.2 | 84.8 ±0.9 | 82.5 ±1.1 | 94.0 ±0.8 | **96.2** **±0.3** | 83.6 ±1.0 | 86.1 ±1.3 | 83.2 ±1.2 | 85.5 ±1.2 |
| C11 | 78.4 ±0.5 | 80.7 ±0.4 | 81.8 ±0.4 | 79.6 ±0.4 | 81.5 ±0.5 | 82.3 ±0.4 | 79.2 ±0.4 | 80.5 ±0.5 | 79.6 ±0.5 | 81.0 ±0.3 | 81.4 ±0.4 | 79.2 ±0.5 | 81.0 ±0.5 | 81.5 ±0.3 | **82.8** **±0.4** | 80.2 ±0.5 | 81.4 ±0.4 |
| C12 | 96.0 ±0.2 | 97.7 ±0.1 | 98.5 ±0.1 | 93.8 ±0.3 | 97.8 ±0.1 | 98.6 ±0.1 | 96.7 ±0.1 | 79.7 ±1.1 | 97.6 ±0.1 | 98.5 ±0.1 | – | 98.3 ±0.1 | **99.1** **±0.1** | 97.9 ±0.1 | 98.7 ±0.1 | 97.3 ±0.1 | 98.4 ±0.1 |
| C13 | 70.9 ±0.8 | 74.1 ±1.0 | 76.7 ±0.8 | 70.7 ±0.7 | 74.0 ±0.8 | 76.7 ±0.8 | 72.7 ±1.0 | 75.8 ±0.9 | 72.4 ±1.2 | 76.4 ±0.6 | 72.3 ±0.9 | 75.0 ±0.7 | **77.9** **±1.0** | 74.4 ±0.7 | 77.6 ±1.0 | 74.1 ±1.1 | 76.9 ±0.9 |
| C14 | – | 84.0 ±0.5 | 84.8 ±0.4 | 82.4 ±0.3 | 80.4 ±0.3 | 81.3 ±0.3 | 82.3 ±0.4 | 83.5 ±0.3 | 83.2 ±0.4 | 84.3 ±0.3 | – | 84.0 ±0.3 | 85.1 ±0.3 | 85.4 ±0.3 | **86.2** **±0.3** | 83.2 ±0.4 | 84.2 ±0.4 |

Table 21: Regression performance per dataset (up to 10K). Best score before rounding is bolded. We report average $R^2$ with 95% CIs. Models: CARTE (CRT), CatBoost (CTB), LightGBM (LGB), RealMLP (MLP), TabM (TBM), TabPFNv2 (PFN), RandomForest (RF), TabSTAR (STR), XGBoost (XGB). Tuned models are marked with a '+'.

| ID | CRT | CTB | CTB+ | LGB | LGB+ | MLP | MLP+ | PFN | RF | STR | TBM | TBM+ | XGB | XGB+ |
|---|---|---|---|---|---|---|---|---|---|---|---|---|---|---|
| R01 | 100.0 ±0.0 | 100.0 ±0.0 | 100.0 ±0.0 | 100.0 ±0.0 | 100.0 ±0.0 | 99.9 ±0.1 | 100.0 ±0.0 | – | **100.0** **±0.0** | 100.0 ±0.0 | 99.8 ±0.1 | 99.8 ±0.1 | 99.9 ±0.1 | 100.0 ±0.0 |
| R02 | – | 98.3 ±1.0 | 98.3 ±0.9 | 98.1 ±1.0 | 98.1 ±1.0 | 98.2 ±1.0 | 98.2 ±1.0 | 98.4 ±0.9 | 98.0 ±1.0 | 98.0 ±1.1 | **99.0** **±0.8** | 99.0 ±0.8 | 97.9 ±1.1 | 98.3 ±0.9 |
| R03 | 71.8 ±0.5 | 73.8 ±0.4 | 74.1 ±0.3 | 72.2 ±0.4 | 72.4 ±0.4 | 69.7 ±0.7 | 70.1 ±0.8 | 74.9 ±0.4 | 68.4 ±0.6 | 70.9 ±0.8 | 75.3 ±0.5 | **75.4** **±0.5** | 69.1 ±0.5 | 73.7 ±0.5 |
| R04 | – | 86.2 ±7.4 | 85.1 ±7.1 | 85.7 ±6.5 | 84.6 ±6.2 | 88.7 ±4.4 | **90.3** **±3.2** | 86.1 ±5.2 | 85.1 ±6.8 | 81.5 ±8.7 | 84.0 ±3.1 | 84.1 ±3.1 | 85.8 ±7.5 | 87.5 ±3.7 |
| R05 | **93.2** **±0.8** | 89.5 ±1.3 | 90.0 ±1.2 | 87.4 ±1.5 | 88.2 ±1.4 | 90.9 ±1.2 | 91.2 ±1.5 | 92.7 ±0.9 | 86.7 ±1.3 | 93.2 ±1.2 | 92.1 ±1.1 | 92.3 ±1.1 | 87.1 ±1.4 | 90.1 ±1.3 |
| R06 | 97.7 ±0.8 | 98.1 ±0.5 | 98.2 ±0.5 | 97.7 ±0.7 | 97.8 ±0.7 | 97.5 ±0.9 | 97.8 ±0.9 | **98.5** **±0.6** | 97.2 ±1.0 | 98.1 ±0.3 | 97.8 ±0.8 | 97.9 ±0.8 | 97.1 ±0.7 | 97.9 ±0.7 |
| R07 | **71.6** **±1.3** | 66.9 ±0.8 | 67.8 ±0.8 | 62.4 ±1.0 | 63.4 ±1.0 | 68.9 ±0.8 | 67.4 ±1.3 | 62.2 ±1.3 | 61.5 ±1.2 | 71.1 ±1.8 | 67.9 ±0.7 | 69.6 ±0.9 | 61.9 ±1.5 | 68.8 ±0.8 |
| R08 | 93.2 ±0.8 | 93.0 ±0.7 | 93.2 ±0.7 | 92.9 ±0.7 | 93.0 ±0.7 | 92.9 ±0.8 | 92.7 ±0.8 | **93.9** **±0.7** | 92.2 ±0.7 | 92.8 ±0.6 | 93.4 ±0.7 | 93.3 ±0.7 | 92.3 ±0.7 | 93.1 ±0.7 |
| R09 | 89.2 ±0.5 | 89.2 ±0.6 | 89.3 ±0.4 | 88.7 ±0.5 | 88.9 ±0.5 | 88.6 ±0.4 | 89.2 ±0.5 | **89.8** **±0.5** | 88.6 ±0.7 | 88.8 ±0.5 | 89.5 ±0.7 | 89.7 ±0.7 | 88.2 ±0.7 | 89.3 ±0.5 |
| R10 | 99.5 ±0.2 | 99.0 ±0.6 | 98.9 ±0.6 | 98.5 ±0.3 | 98.5 ±0.3 | 99.3 ±0.2 | 99.4 ±0.2 | 99.1 ±0.3 | 99.5 ±0.3 | **99.5** **±0.1** | 98.9 ±0.7 | 99.1 ±0.4 | 98.6 ±0.3 | 99.4 ±0.2 |
| R11 | – | 94.2 ±0.9 | 94.0 ±0.9 | 93.6 ±0.9 | 93.6 ±1.0 | 94.2 ±1.0 | 94.4 ±1.0 | 94.5 ±0.9 | 92.5 ±1.3 | 93.9 ±1.1 | 94.5 ±0.8 | **94.6** **±0.8** | 93.3 ±1.1 | 94.2 ±0.8 |
| R12 | – | 98.5 ±0.3 | 98.5 ±0.2 | 98.3 ±0.3 | 98.4 ±0.2 | 98.4 ±0.3 | 98.4 ±0.2 | 98.5 ±0.2 | 98.0 ±0.3 | 98.3 ±0.2 | 98.6 ±0.2 | **98.7** **±0.2** | 98.2 ±0.3 | 98.5 ±0.2 |
| R13 | 52.8 ±2.1 | 57.6 ±2.7 | 58.2 ±2.6 | 54.7 ±2.8 | 55.0 ±2.6 | 55.4 ±2.5 | 55.1 ±2.5 | 53.8 ±2.7 | 51.8 ±3.5 | 53.4 ±3.0 | **58.7** **±1.9** | 58.4 ±2.1 | 49.0 ±3.3 | 57.5 ±2.7 |
| R14 | 96.6 ±0.2 | 97.3 ±0.1 | **97.3** **±0.1** | 97.0 ±0.1 | 97.0 ±0.1 | 96.8 ±0.1 | 96.7 ±0.2 | 96.2 ±0.2 | 96.8 ±0.1 | 96.4 ±0.1 | 96.3 ±0.1 | 96.3 ±0.1 | 97.1 ±0.1 | 97.1 ±0.1 |
| R15 | – | 79.5 ±0.9 | 79.9 ±1.1 | 77.8 ±0.9 | 77.7 ±0.9 | 76.2 ±1.4 | 77.4 ±1.5 | 79.4 ±1.0 | 77.3 ±1.2 | 78.9 ±1.5 | **80.6** **±0.7** | 79.9 ±0.7 | 76.8 ±1.2 | 80.3 ±1.0 |

Continued on next page

Table 21: Regression performance per dataset (up to 10K). Best score before rounding is bolded. We report average $R^2$ with 95% CIs. Models: CARTE (CRT), CatBoost (CTB), LightGBM (LGB), RealMLP (MLP), TabM (TBM), TabPFNv2 (PFN), RandomForest (RF), TabSTAR (STR), XGBoost (XGB). Tuned models are marked with a '+'.

| ID | CRT | CTB | CTB+ | LGB | LGB+ | MLP | MLP+ | PFN | RF | STR | TBM | TBM+ | XGB | XGB+ |
|---|---|---|---|---|---|---|---|---|---|---|---|---|---|---|
| R16 | 98.7 ±0.0 | 98.7 ±0.0 | 98.7 ±0.0 | 98.7 ±0.0 | 98.7 ±0.0 | 97.6 ±0.1 | 97.8 ±0.1 | **98.8** ±**0.0** | 97.7 ±0.1 | 98.7 ±0.0 | 97.9 ±0.1 | 97.9 ±0.1 | 97.7 ±0.1 | 97.8 ±0.1 |
| R17 | 95.3 ±2.0 | 94.8 ±2.4 | 95.2 ±2.0 | 94.6 ±2.3 | 95.2 ±2.0 | 94.4 ±2.4 | 95.4 ±1.9 | 95.1 ±2.1 | 94.2 ±2.5 | 94.7 ±2.3 | 96.7 ±1.1 | **96.7** ±**1.1** | 94.5 ±2.6 | 95.4 ±1.9 |
| R18 | 98.1 ±0.4 | 98.2 ±0.4 | 98.2 ±0.4 | 98.1 ±0.4 | 98.1 ±0.4 | 98.0 ±0.5 | 97.7 ±0.5 | 93.1 ±1.3 | 97.7 ±0.4 | 97.4 ±0.6 | 98.2 ±0.3 | 98.1 ±0.4 | 97.5 ±0.8 | **98.3** ±**0.4** |
| R19 | – | 85.5 ±0.8 | **85.8** ±**0.9** | 85.2 ±0.9 | 85.2 ±0.9 | 82.7 ±0.8 | 83.3 ±1.3 | 84.0 ±1.1 | 85.3 ±0.9 | 83.8 ±1.1 | 85.0 ±1.0 | 85.1 ±0.9 | 83.9 ±1.0 | 82.4 ±2.0 |
| R20 | **96.4** ±**0.5** | 96.0 ±0.6 | 96.2 ±0.5 | 95.7 ±0.5 | 95.8 ±0.5 | 95.3 ±0.8 | 96.0 ±0.5 | 93.9 ±1.0 | 95.7 ±0.6 | 95.6 ±0.9 | 95.9 ±0.7 | 95.8 ±0.7 | 95.3 ±0.9 | 96.2 ±0.5 |
| R21 | 92.3 ±1.2 | 92.5 ±1.1 | 92.6 ±1.1 | 92.3 ±1.2 | 92.3 ±1.2 | 92.0 ±1.1 | 91.7 ±1.3 | **93.3** ±**1.1** | 91.8 ±1.1 | 92.3 ±1.0 | 91.6 ±1.4 | 92.1 ±1.2 | 91.6 ±1.0 | 92.6 ±1.0 |
| R22 | – | 45.2 ±5.5 | 45.2 ±5.9 | 42.4 ±5.6 | 44.7 ±5.5 | 43.6 ±6.4 | 44.4 ±6.3 | 43.8 ±5.5 | 39.1 ±6.4 | 39.7 ±5.5 | 47.9 ±5.5 | **48.1** ±**5.4** | 36.1 ±7.1 | 46.8 ±5.8 |
| R23 | 85.0 ±1.8 | 85.4 ±1.2 | 85.5 ±1.1 | 84.3 ±1.5 | 85.5 ±1.2 | 83.3 ±1.8 | 85.6 ±1.5 | 84.8 ±1.6 | 81.8 ±1.5 | 83.0 ±1.6 | 86.4 ±0.9 | **86.6** ±**0.8** | 83.3 ±1.2 | 85.1 ±1.2 |
| R24 | **86.0** ±**0.6** | 84.0 ±0.8 | 85.8 ±0.7 | 78.8 ±0.9 | 79.5 ±0.9 | 84.6 ±0.8 | 82.8 ±0.9 | 70.3 ±1.6 | 79.7 ±1.0 | 81.8 ±1.8 | 84.9 ±0.9 | 85.5 ±0.8 | 82.9 ±0.7 | 85.9 ±0.9 |
| R25 | 94.3 ±0.6 | **95.4** ±**0.5** | 95.3 ±0.5 | 95.2 ±0.6 | 95.2 ±0.5 | 93.2 ±0.6 | 94.1 ±0.7 | 86.7 ±1.0 | 94.8 ±0.5 | 94.8 ±0.5 | 94.3 ±0.4 | 94.5 ±0.4 | 94.6 ±0.6 | 95.4 ±0.5 |
| R26 | 99.8 ±0.1 | 99.4 ±0.1 | 99.4 ±0.1 | 99.2 ±0.2 | 99.3 ±0.2 | 99.9 ±0.1 | **99.9** ±**0.0** | 99.8 ±0.1 | 99.0 ±0.2 | 99.6 ±0.1 | 98.1 ±0.7 | 98.7 ±0.8 | 99.1 ±0.2 | 99.5 ±0.1 |
| R27 | 81.9 ±1.6 | 85.2 ±1.3 | 85.3 ±1.4 | 84.4 ±1.4 | 84.8 ±1.3 | 83.3 ±1.5 | 76.7 ±11.2 | 82.2 ±1.6 | 84.8 ±1.3 | 82.0 ±1.6 | 83.8 ±1.5 | 84.0 ±1.7 | 83.2 ±1.2 | **85.5** ±**1.2** |
| R28 | 52.5 ±2.6 | 53.4 ±2.7 | 53.3 ±2.8 | 50.0 ±3.0 | 51.0 ±2.9 | 52.3 ±3.2 | 52.6 ±3.0 | **61.7** ±**2.1** | 46.0 ±2.7 | 51.5 ±2.9 | 54.7 ±2.7 | 55.6 ±2.6 | 45.3 ±2.3 | 53.5 ±2.7 |
| R29 | – | 94.4 ±0.7 | 94.4 ±0.7 | 94.8 ±0.7 | 94.8 ±0.8 | 94.3 ±0.7 | 94.4 ±0.8 | **95.7** ±**0.6** | 93.0 ±1.0 | 94.3 ±0.9 | 94.9 ±0.8 | 95.0 ±0.7 | 93.4 ±0.8 | 94.5 ±0.8 |
| R30 | 23.8 ±4.2 | 22.7 ±4.1 | 24.6 ±3.9 | 22.3 ±3.7 | 23.9 ±4.2 | 15.7 ±3.9 | 18.8 ±5.4 | 20.9 ±4.8 | 23.5 ±3.5 | 15.7 ±5.0 | 22.9 ±5.0 | 23.4 ±4.8 | 17.5 ±4.2 | **25.5** ±**3.7** |
| R31 | 92.1 ±0.3 | 92.1 ±0.4 | 92.0 ±0.4 | 91.6 ±0.4 | 91.6 ±0.4 | 91.5 ±0.3 | 91.5 ±0.4 | **93.2** ±**0.3** | 89.9 ±0.5 | 91.7 ±0.4 | 92.4 ±0.5 | 92.6 ±0.5 | 90.3 ±0.5 | 92.0 ±0.4 |
| R32 | – | 28.8 ±6.3 | 28.2 ±5.8 | 25.0 ±6.4 | 27.3 ±5.7 | 19.8 ±8.8 | 19.8 ±7.1 | 26.2 ±5.4 | 28.6 ±5.9 | 19.4 ±5.6 | 25.8 ±5.3 | 27.3 ±6.0 | 22.7 ±7.3 | **31.6** ±**6.2** |
| R33 | 46.6 ±1.5 | 47.4 ±1.7 | 47.8 ±1.7 | 44.8 ±1.8 | 45.2 ±1.6 | 40.4 ±2.1 | 44.6 ±1.6 | 44.9 ±1.6 | 40.2 ±1.6 | 46.0 ±1.8 | 48.4 ±1.1 | **48.4** ±**1.2** | 40.3 ±2.1 | 47.9 ±1.6 |
| R34 | – | 90.9 ±2.5 | 91.4 ±2.1 | 88.9 ±2.9 | 89.9 ±2.4 | 88.6 ±3.0 | 90.7 ±1.4 | **92.3** ±**1.8** | 88.7 ±3.1 | 89.1 ±2.9 | 91.1 ±1.7 | 90.9 ±1.8 | 88.9 ±3.6 | 91.6 ±2.1 |
| R35 | **85.9** ±**0.6** | 84.4 ±0.5 | 84.5 ±0.5 | 83.7 ±0.6 | 83.7 ±0.6 | 83.1 ±0.9 | 84.2 ±0.8 | 85.2 ±0.7 | 81.2 ±0.4 | 85.7 ±0.8 | 84.5 ±0.6 | 84.5 ±0.5 | 81.6 ±0.8 | 84.4 ±0.3 |
| R36 | 96.3 ±0.9 | 95.2 ±0.9 | 95.5 ±0.9 | 94.7 ±1.0 | 94.9 ±1.0 | 95.4 ±1.1 | 95.7 ±0.9 | 91.2 ±1.1 | 94.3 ±1.0 | 95.7 ±0.8 | 96.2 ±0.6 | **96.4** ±**0.6** | 94.4 ±0.9 | 95.5 ±0.9 |

### F.3 Head-to-head comparisons

We compare the performance of TabSTAR against each of the models in head-to-head comparisons. We report win rate, which can be seen as a private case of the normalized metric with only two models. We exclude failed runs when comparing to CARTE and TabPFN-v2. Table 23 shows the performance of TabSTAR against all models competing up to 10K examples, for both regression and classification, with 95% CIs over the win rate. Table 24 does the same for TabSTAR-Unlimited.

### F.4 Running Times and Compute Information

**Hardware** All baselines are evaluated using a single *NVIDIA A100-SXM4* GPU with 40GB memory, and 8 CPU cores of type *AMD EPYC 7742 64-Core Processor*. The only exclusions are TabPFN-v2,

Table 22: Regression performance per dataset (above 10K). Best score before rounding is bolded. We report average $R^2$ with 95% CIs. Models: CARTE (CRT), CatBoost (CTB), LightGBM (LGB), RealMLP (MLP), TabM (TBM), TabPFNv2 (PFN), RandomForest (RF), TabSTAR (STR), XGBoost (XGB). Unlimited models are marked with a '!'.

| ID | CRT | CTB | CTB! | LGB | LGB! | MLP | MLP! | PFN | STR | STR! | TBM | TBM! | XGB | XGB! |
|---|---|---|---|---|---|---|---|---|---|---|---|---|---|---|
| R01 | 100.0 ±0.0 | 100.0 ±0.0 | 100.0 ±0.0 | 100.0 ±0.0 | **100.0** ±**0.0** | 100.0 ±0.0 | – | – | 100.0 ±0.0 | 100.0 ±0.0 | 99.8 ±0.1 | 100.0 ±0.0 | 100.0 ±0.0 | 100.0 ±0.0 |
| R02 | – | 98.3 ±0.9 | 98.8 ±0.9 | 98.1 ±1.0 | 98.8 ±1.0 | 98.2 ±1.0 | 98.8 ±1.0 | 98.4 ±0.9 | 98.0 ±1.1 | 98.4 ±1.0 | **99.0** ±**0.8** | 98.5 ±0.9 | 98.3 ±0.9 | 98.8 ±0.9 |
| R03 | 71.8 ±0.5 | 74.1 ±0.3 | 75.1 ±0.4 | 72.4 ±0.4 | 72.7 ±0.5 | 70.1 ±0.8 | 71.0 ±0.6 | 74.9 ±0.4 | 70.9 ±0.8 | 72.3 ±0.6 | 75.4 ±0.5 | **76.4** ±**0.5** | 73.7 ±0.5 | 74.6 ±0.6 |
| R04 | – | 85.1 ±7.1 | 91.3 ±0.8 | 84.6 ±6.2 | 91.0 ±1.3 | 90.3 ±3.2 | **91.4** ±**0.8** | 86.1 ±5.2 | 81.5 ±8.7 | 91.0 ±0.7 | 84.1 ±3.1 | 91.0 ±0.7 | 87.5 ±3.7 | 91.3 ±0.6 |
| R07 | 71.6 ±1.3 | 67.8 ±0.8 | 85.2 ±0.3 | 63.4 ±1.0 | 70.1 ±0.5 | 67.4 ±1.3 | 85.0 ±0.3 | 62.2 ±1.3 | 71.1 ±1.8 | 79.9 ±1.8 | 69.6 ±0.9 | 84.0 ±1.3 | 68.8 ±0.8 | **86.1** ±**0.7** |
| R08 | 93.2 ±0.8 | 93.2 ±0.7 | 93.8 ±0.6 | 93.0 ±0.7 | 93.6 ±0.5 | 92.7 ±0.8 | 93.5 ±0.7 | **93.9** ±**0.7** | 92.8 ±0.6 | 93.3 ±0.6 | 93.3 ±0.7 | 93.9 ±0.6 | 93.1 ±0.7 | 93.9 ±0.5 |
| R09 | 89.2 ±0.5 | 89.3 ±0.4 | 89.5 ±0.4 | 88.9 ±0.5 | 89.2 ±0.5 | 89.2 ±0.5 | 89.3 ±0.5 | **89.8** ±**0.5** | 88.8 ±0.5 | 89.2 ±0.6 | 89.7 ±0.7 | 89.3 ±0.4 | 89.3 ±0.5 | 89.5 ±0.4 |
| R12 | – | 98.5 ±0.2 | 98.9 ±0.1 | 98.4 ±0.2 | 98.9 ±0.2 | 98.4 ±0.2 | 99.0 ±0.1 | 98.5 ±0.2 | 98.3 ±0.2 | 98.4 ±0.2 | 98.7 ±0.2 | **99.0** ±**0.2** | 98.5 ±0.2 | 98.9 ±0.2 |
| R14 | 96.6 ±0.2 | 97.3 ±0.1 | **97.8** ±**0.1** | 97.0 ±0.1 | 97.5 ±0.1 | 96.7 ±0.2 | 97.4 ±0.1 | 96.2 ±0.2 | 96.4 ±0.1 | 96.8 ±0.2 | 96.3 ±0.1 | 97.2 ±0.1 | 97.1 ±0.1 | 97.5 ±0.1 |
| R15 | – | 79.9 ±1.1 | 86.5 ±0.7 | 77.7 ±0.9 | 82.0 ±0.6 | 77.4 ±1.5 | 85.7 ±0.6 | 79.4 ±1.0 | 78.9 ±1.5 | 83.9 ±1.4 | 79.9 ±0.7 | **87.2** ±**0.8** | 80.3 ±1.0 | 87.0 ±0.7 |
| R16 | 98.7 ±0.0 | 98.7 ±0.0 | **98.8** ±**0.0** | 98.7 ±0.0 | 98.8 ±0.0 | 97.8 ±0.1 | 98.0 ±0.0 | 98.8 ±0.0 | 98.7 ±0.0 | 98.8 ±0.0 | 97.9 ±0.0 | 98.1 ±0.1 | 97.8 ±0.1 | 98.0 ±0.1 |
| R17 | 95.3 ±2.0 | 95.2 ±2.0 | 97.6 ±0.6 | 95.2 ±2.0 | 97.6 ±0.7 | 95.4 ±1.9 | 97.5 ±0.6 | 95.1 ±2.1 | 94.7 ±2.3 | 97.6 ±0.6 | 96.7 ±1.1 | 97.2 ±0.7 | 95.4 ±1.9 | **97.7** ±**0.6** |
| R18 | 98.1 ±0.4 | 98.2 ±0.4 | **99.0** ±**0.2** | 98.1 ±0.4 | 98.8 ±0.3 | 97.7 ±0.5 | 98.4 ±0.4 | 93.1 ±1.3 | 97.4 ±0.6 | 97.7 ±0.6 | 98.1 ±0.4 | 98.7 ±0.3 | 98.3 ±0.4 | 98.9 ±0.3 |
| R20 | **96.4** ±**0.5** | 96.2 ±0.5 | 96.3 ±0.4 | 95.8 ±0.5 | 95.8 ±0.5 | 96.0 ±0.5 | 95.6 ±0.5 | 93.9 ±1.0 | 95.6 ±0.9 | 95.3 ±1.0 | 95.8 ±0.7 | 95.9 ±0.6 | 96.2 ±0.5 | 96.3 ±0.5 |
| R23 | 85.0 ±1.8 | 85.5 ±1.1 | 86.3 ±0.7 | 85.5 ±1.2 | 85.8 ±0.7 | 85.6 ±1.5 | 86.6 ±0.8 | 84.8 ±1.6 | 83.0 ±1.6 | 85.2 ±1.7 | 86.6 ±0.8 | **88.3** ±**0.4** | 85.1 ±1.2 | 86.1 ±0.8 |
| R24 | 86.0 ±0.6 | 85.8 ±0.7 | 97.1 ±0.3 | 79.5 ±0.9 | 85.4 ±0.6 | 82.8 ±0.9 | 96.5 ±0.4 | 70.3 ±1.6 | 81.8 ±1.8 | 95.5 ±0.7 | 85.5 ±0.8 | **97.8** ±**0.3** | 85.9 ±0.9 | 97.3 ±0.3 |
| R25 | 94.3 ±0.6 | 95.3 ±0.5 | **95.9** ±**0.4** | 95.2 ±0.5 | 95.7 ±0.5 | 94.1 ±0.7 | 95.0 ±0.5 | 86.7 ±1.0 | 94.8 ±0.5 | 95.1 ±0.5 | 94.5 ±0.4 | 95.9 ±0.4 | 95.4 ±0.5 | 95.8 ±0.5 |
| R31 | 92.1 ±0.3 | 92.0 ±0.4 | 93.0 ±0.3 | 91.6 ±0.4 | 92.5 ±0.3 | 91.5 ±0.4 | 92.5 ±0.4 | 93.2 ±0.3 | 91.7 ±0.4 | 92.7 ±0.4 | 92.6 ±0.5 | **93.6** ±**0.3** | 92.0 ±0.4 | 92.9 ±0.3 |
| R33 | 46.6 ±1.5 | 47.8 ±1.7 | 55.9 ±1.1 | 45.2 ±1.6 | 49.8 ±1.3 | 44.6 ±1.6 | 54.5 ±1.3 | 44.9 ±1.6 | 46.0 ±1.8 | 57.2 ±1.3 | 48.4 ±1.2 | **57.8** ±**1.2** | 47.9 ±1.6 | 55.7 ±1.3 |
| R36 | 96.3 ±0.9 | 95.5 ±0.9 | 95.5 ±0.9 | 94.9 ±1.0 | 94.9 ±1.0 | 95.7 ±0.9 | 95.8 ±0.9 | 91.2 ±1.1 | 95.7 ±0.8 | 95.5 ±0.5 | **96.4** ±**0.6** | 96.1 ±0.9 | 95.5 ±0.9 | 95.5 ±0.9 |

Table 23: Win rates of TabSTAR (up to 10K) against baselines. Win rate with 95% CI.

| Model | Classification | Regression |
|---|---|---|
| CARTE | 93.3 ± 5.2 | 35.7 ± 5.9 |
| CatBoost | 80.7 ± 6.6 | 33.6 ± 4.9 |
| CatBoost-Tuned | 75.0 ± 7.2 | 32.2 ± 4.8 |
| LightGBM | 92.1 ± 4.5 | 49.7 ± 5.2 |
| LightGBM-Tuned | 90.7 ± 4.8 | 45.3 ± 5.1 |
| RandomForest | 95.7 ± 3.4 | 66.9 ± 4.9 |
| RealMLP | 82.1 ± 6.4 | 51.4 ± 5.2 |
| RealMLP-Tuned | 83.6 ± 6.2 | 49.4 ± 5.2 |
| TabDPT | 73.6 ± 7.3 | 66.7 ± 4.9 |
| TabICL | 70.0 ± 7.6 | – |
| TabM | 64.3 ± 8.0 | 39.2 ± 5.0 |
| TabM-Tuned | 63.3 ± 8.0 | 37.4 ± 5.0 |
| TabPFN-v2 | 66.4 ± 8.9 | 40.9 ± 5.2 |
| XGBoost | 96.4 ± 3.1 | 69.4 ± 4.8 |
| XGBoost-Tuned | 78.6 ± 6.8 | 30.3 ± 4.8 |

Table 24: Win rates of TabSTAR-Unlimited (above 10K) against baselines. Win rate with 95% CI.

| Model | Classification | Regression |
|---|---|---|
| CARTE | 98.6 ± 2.8 | 63.5 ± 7.5 |
| CatBoost-Tuned | 96.0 ± 3.9 | 53.0 ± 6.9 |
| CatBoost-Tuned-Unlimited | 77.0 ± 8.3 | 22.0 ± 5.8 |
| LightGBM-Tuned | 98.0 ± 2.8 | 69.5 ± 6.4 |
| LightGBM-Tuned-Unlimited | 92.0 ± 5.3 | 44.0 ± 6.9 |
| RealMLP-Tuned | 97.0 ± 3.4 | 72.0 ± 6.2 |
| RealMLP-Tuned-Unlimited | 86.0 ± 6.8 | 38.9 ± 7.0 |
| TabDPT | 85.0 ± 7.0 | – |
| TabICL | 85.0 ± 7.0 | – |
| TabICL-Unlimited | 82.0 ± 7.6 | – |
| TabM-Tuned | 76.8 ± 8.4 | 65.2 ± 6.7 |
| TabM-Tuned-Unlimited | 53.0 ± 9.8 | 22.5 ± 5.8 |
| TabPFN-v2 | 90.0 ± 7.1 | 61.1 ± 7.0 |
| TabSTAR | 94.0 ± 4.7 | 82.0 ± 5.3 |
| XGBoost-Tuned | 95.0 ± 4.3 | 59.5 ± 6.8 |
| XGBoost-Tuned-Unlimited | 81.0 ± 7.7 | 28.5 ± 6.3 |

which runs on its API version, and CARTE, for which we employ several *NVIDIA GeForce GTX 1080 Ti* GPUs with 11GB memory, as their high latency and lack of default hyperparameters forced us to rely on more accessible hardware.

**Running Times**   Table 25 presents the average running times of the different models per dataset, with up to 10,000 examples, excluding the tuned variants, TabPFN-v2 and CARTE. TabSTAR's average running time for a downstream task ranges from 59s (C04) to 9,332s (R24). Although higher than a single run of other baselines, these times remain far below the 14,400s (4 hours) typically required for their tuned counterparts. TabICL achieves considerably faster running times; however, their ICL-based approach limits them to smaller datasets, and inference is as costly as training (see §5). TabPFN-v2, though unreported, is subject to the same limitation.

## F.5   TabSTAR's Cost Analysis by Number of Features

While §5 reports average computational costs, these values vary considerably across datasets, depending on factors such as the number of examples, the feature count, and the amount of textual fields. In this subsection, we examine how TabSTAR's computational cost scales with the number of features.

Table 25: Average model training running time in seconds, per dataset, for up to 10K examples.

| ID | CatBoost | LightGBM | RandomForest | RealMLP | TabICL | TabSTAR | XGBoost |
|---|---|---|---|---|---|---|---|
| C01 | 116 | 35 | 48 | 148 | 37 | 397 | 36 |
| C02 | 251 | 57 | 75 | 179 | 58 | 1,135 | 58 |
| C03 | 106 | 51 | 61 | 147 | 49 | 553 | 48 |
| C04 | 29 | 17 | 19 | 25 | 20 | 59 | 18 |
| C05 | 26 | 13 | 16 | 63 | 13 | 132 | 13 |
| C06 | 170 | 79 | 83 | 126 | 77 | 938 | 81 |
| C07 | 163 | 78 | 98 | 186 | 80 | 1,044 | 78 |
| C08 | 89 | 72 | 86 | 180 | 74 | 1,057 | 69 |
| C09 | 33 | 25 | 33 | 128 | 29 | 414 | 25 |
| C10 | 49 | 35 | 43 | 126 | 36 | 572 | 35 |
| C11 | 43 | 22 | 29 | 121 | 20 | 521 | 20 |
| C12 | 614 | 39 | 37 | 145 | 28 | 729 | 36 |
| C13 | 70 | 55 | 68 | 163 | 55 | 551 | 50 |
| C14 | 1,938 | 307 | 333 | 418 | 311 | 4,068 | 329 |
| R01 | 488 | 161 | 1,204 | 368 | – | 2,908 | 161 |
| R02 | 28 | 10 | 45 | 128 | – | 283 | 10 |
| R03 | 111 | 62 | 282 | 171 | – | 592 | 61 |
| R04 | 21 | 9 | 17 | 138 | – | 240 | 7 |
| R05 | 65 | 20 | 68 | 90 | – | 366 | 23 |
| R06 | 60 | 24 | 119 | 129 | – | 501 | 23 |
| R07 | 66 | 39 | 285 | 145 | – | 673 | 44 |
| R08 | 213 | 105 | 951 | 222 | – | 1,448 | 112 |
| R09 | 12 | 8 | 56 | 115 | – | 604 | 7 |
| R10 | 102 | 45 | 104 | 72 | – | 255 | 47 |
| R11 | 26 | 10 | 45 | 98 | – | 197 | 10 |
| R12 | 34 | 10 | 42 | 133 | – | 335 | 11 |
| R13 | 78 | 44 | 145 | 94 | – | 484 | 46 |
| R14 | 67 | 24 | 91 | 159 | – | 1,296 | 27 |
| R15 | 38 | 18 | 102 | 132 | – | 390 | 17 |
| R16 | 108 | 76 | 316 | 193 | – | 898 | 80 |
| R17 | 62 | 54 | 1,110 | 160 | – | 261 | 53 |
| R18 | 124 | 74 | 1,705 | 201 | – | 553 | 68 |
| R19 | 22 | 13 | 38 | 107 | – | 198 | 12 |
| R20 | 70 | 52 | 401 | 176 | – | 705 | 49 |
| R21 | 53 | 29 | 86 | 62 | – | 222 | 30 |
| R22 | 57 | 31 | 62 | 52 | – | 171 | 33 |
| R23 | 41 | 19 | 137 | 140 | – | 281 | 18 |
| R24 | 461 | 396 | 650 | 520 | – | 9,332 | 392 |
| R25 | 191 | 104 | 2,583 | 223 | – | 750 | 110 |
| R26 | 29 | 12 | 22 | 30 | – | 205 | 11 |
| R27 | 103 | 49 | 164 | 92 | – | 334 | 49 |
| R28 | 175 | 92 | 460 | 158 | – | 647 | 97 |
| R29 | 34 | 8 | 21 | 100 | – | 308 | 7 |
| R30 | 26 | 14 | 47 | 56 | – | 114 | 14 |
| R31 | 148 | 78 | 505 | 200 | – | 797 | 77 |
| R32 | 20 | 11 | 23 | 31 | – | 82 | 11 |
| R33 | 89 | 55 | 271 | 161 | – | 608 | 47 |
| R34 | 66 | 22 | 66 | 50 | – | 122 | 24 |
| R35 | 99 | 50 | 146 | 101 | – | 411 | 50 |
| R36 | 58 | 38 | 223 | 146 | – | 823 | 39 |

Table 26: Cost Analysis for TabSTAR per number of features. Median training and inference times and peak memory usage on GPU and CPU, aggregated across 45 datasets.

| Feature Group | Train | | | Inference | | |
|---|---|---|---|---|---|---|
| | Time (s) | GPU (GB) | CPU (GB) | Time (s) | GPU (GB) | CPU (GB) |
| Up to 10 | 85.6 | 1.9 | 4.4 | 0.5 | 1.6 | 4.4 |
| 30-50 | 228.8 | 3.1 | 4.5 | 1.0 | 1.7 | 4.4 |
| 100+ | 417.9 | 7.8 | 4.5 | 2.5 | 2.3 | 4.4 |

Table 26 summarizes both latency and memory consumption of TabSTAR, aggregated over a group of up to 10 features, a group of 30-50 features and a group of more than 100 features. These datasets were randomly sampled from the pretraining corpus; consequently, the reported times may differ from those observed on the benchmark datasets. We observe that latency can increase by a factor of 5, and GPU memory consumption by a factor of 4. This behavior stems from the transformer's architecture, whose computational complexity is quadratic in the sequence length. This raises concerns for training TabSTAR over datasets with hundreds of features, and calls for improvements to expand to datasets of thousands of features.

Additionally, TabSTAR's memory consumption in classification tasks is affected by the number of classes due to its target-aware tokens. We assume that for TabSTAR, the effective number of features equals the actual number of features plus the number of classes. For highly multiclass datasets, this can become a limiting factor.

### F.6 Number of Trials for Tuned models

The tuned models are optimized with a budget of 4 hours using 8 CPU cores. Each trial is executed on a single core, ensuring that at least one trial could be completed within the allocated time. Table 27 presents the number of hyperparameter trials per dataset.

## G    Extended Analysis

In this section we expand on the analysis results discussed in §6.

### G.1    Evaluation Datasets for Analysis

All experiments are conducted on 20 datasets from the benchmark described in Appendix D. Each experiment reports performance using AUROC for classification and $R^2$ for regression. Each tables reports both regression and classification tasks, distinguishable by their ID. Furthermore, each experiment compares its variants to other models, excluding TabPFN-v2 and CARTE, which could not be executed on all datasets. Although this comparison covers only a subset of the benchmark and should not be interpreted as conclusive, the results nonetheless offer valuable reference points.

### G.2    The Role of the Encoder Unfreezing (Q1)

Table 28 shows the results for each variant of the experiment presented in §6. It is evident that unfreezing the textual encoder yields a significant performance boost across datasets, as seen in Figure 6: The frozen TabSTAR variant is much worse than any other baseline! Furthermore, while finetuning a single layer gives a significant boost, it underperforms compared to 6 unfrozen layers.

### G.3    The Effect of Pretraining (Q2)

We pretrain three TabSTAR variants on nested dataset subsets of size 16, 64, and 256. The 64-dataset variant contains the original 16 plus 48 new datasets, and the 256-dataset variant builds on those 64 by adding another 192. This cumulative design minimizes variance between variants so that performance differences reflect only the effect of increasing data volume.

While LoRA [42] is a powerful technique, it can't be applied to a randomly initialized model. Therefore, we perform full finetuning of the non-pretrained model, as explained in Appendix B.2.

Table 27: Average number of hyperparameter trials per dataset. Tuned models: CatBoost (CTB), LightGBM (LGB), RealMLP (MLP), XGBoost (XGB). Unlimited models are marked with a '!'.

| Dataset | CTB | CTB! | LGB | LGB! | MLP | MLP! | XGB | XGB! |
|---------|-----|------|-----|------|-----|------|-----|------|
| C01 | 27.1 | 24.4 | 770.2 | 812.0 | 56.0 | 26.7 | 97.7 | 90.0 |
| C02 | 18.1 | 9.4 | 417.5 | 51.3 | 31.9 | 8.0 | 47.2 | 19.6 |
| C03 | 18.0 | 15.8 | 312.0 | 338.1 | 37.4 | 25.2 | 42.4 | 39.5 |
| C04 | 105.8 | – | 11283.8 | – | 642.1 | – | 458.5 | – |
| C05 | 122.1 | – | 3373.3 | – | 98.3 | – | 228.3 | – |
| C06 | 18.1 | – | 406.9 | – | 67.7 | – | 44.6 | – |
| C07 | 17.3 | 18.5 | 335.6 | 376.6 | 31.8 | 23.7 | 53.4 | 37.7 |
| C08 | 101.1 | 67.8 | 2911.4 | 2394.6 | 43.1 | 37.3 | 508.3 | 566.1 |
| C09 | 169.0 | 81.3 | 3969.5 | 475.7 | 50.5 | 9.2 | 883.9 | 207.2 |
| C10 | 73.7 | – | 1598.1 | – | 55.4 | – | 165.2 | – |
| C11 | 58.3 | 48.5 | 812.4 | 863.4 | 44.8 | 26.0 | 124.1 | 96.3 |
| C12 | 9.9 | 8.5 | 207.2 | 34.4 | 35.3 | 8.2 | 41.8 | 17.3 |
| C13 | 85.8 | 27.4 | 1848.3 | 329.6 | 40.8 | 8.7 | 234.6 | 105.7 |
| C14 | 8.6 | 8.4 | 50.9 | 36.3 | 14.9 | 9.7 | 13.7 | 11.6 |
| R01 | 14.2 | 12.2 | 278.5 | 169.6 | 11.3 | – | 32.8 | 15.4 |
| R02 | 138.9 | 118.6 | 4463.0 | 5214.6 | 48.0 | 30.7 | 742.4 | 1054.5 |
| R03 | 55.5 | 64.2 | 1033.6 | 1265.9 | 35.8 | 30.0 | 94.3 | 97.8 |
| R04 | 123.8 | 31.6 | 4309.9 | 714.9 | 44.1 | 8.7 | 1324.0 | 364.8 |
| R05 | 78.8 | – | 4428.5 | – | 87.9 | – | 307.0 | – |
| R06 | 88.8 | – | 2305.3 | – | 47.5 | – | 232.2 | – |
| R07 | 109.9 | 58.8 | 1370.0 | 344.1 | 42.0 | 8.5 | 145.6 | 72.5 |
| R08 | 30.5 | 30.1 | 568.6 | 122.9 | 21.4 | 10.4 | 65.7 | 47.1 |
| R09 | 194.6 | 208.7 | 6391.0 | 5975.1 | 50.4 | 31.2 | 589.0 | 680.6 |
| R10 | 53.7 | – | 2806.9 | – | 135.9 | – | 241.8 | – |
| R11 | 142.9 | – | 4669.1 | – | 58.7 | – | 662.6 | – |
| R12 | 126.8 | 45.6 | 2990.9 | 814.7 | 46.9 | 8.6 | 385.8 | 178.0 |
| R13 | 80.9 | – | 2814.6 | – | 91.2 | – | 204.8 | – |
| R14 | 59.9 | 67.6 | 1725.3 | 1542.6 | 39.5 | 15.2 | 200.6 | 177.0 |
| R15 | 107.4 | 69.2 | 2896.1 | 1177.4 | 52.5 | 14.0 | 337.5 | 242.7 |
| R16 | 65.8 | 46.3 | 1446.3 | 663.7 | 34.9 | 10.9 | 160.9 | 104.9 |
| R17 | 79.7 | 44.6 | 1517.3 | 316.3 | 45.3 | 9.1 | 254.0 | 104.4 |
| R18 | 39.1 | 43.8 | 1572.4 | 1041.4 | 29.4 | 16.6 | 258.8 | 144.5 |
| R19 | 117.5 | – | 4046.2 | – | 63.1 | – | 1063.2 | – |
| R20 | 62.6 | 89.1 | 1611.5 | 2176.4 | 36.4 | 30.1 | 250.5 | 258.8 |
| R21 | 101.6 | – | 3284.9 | – | 122.3 | – | 159.4 | – |
| R22 | 67.8 | – | 3584.3 | – | 259.9 | – | 127.2 | – |
| R23 | 91.8 | 105.4 | 3179.2 | 2773.0 | 53.4 | 27.6 | 420.1 | 314.5 |
| R24 | 52.4 | 38.1 | 867.2 | 417.7 | 35.2 | 10.4 | 104.7 | 60.3 |
| R25 | 40.4 | 33.5 | 806.9 | 357.6 | 23.6 | 10.8 | 122.4 | 72.0 |
| R26 | 166.5 | – | 12567.7 | – | 279.4 | – | 625.0 | – |
| R27 | 53.9 | – | 2034.2 | – | 85.3 | – | 155.5 | – |
| R28 | 40.9 | – | 670.8 | – | 37.5 | – | 59.3 | – |
| R29 | 135.3 | – | 7615.2 | – | 63.8 | – | 1024.3 | – |
| R30 | 194.3 | – | 5275.3 | – | 114.8 | – | 431.7 | – |
| R31 | 36.6 | 39.5 | 774.5 | 504.3 | 26.4 | 12.6 | 90.3 | 54.1 |
| R32 | 143.2 | – | 6605.7 | – | 191.7 | – | 337.3 | – |
| R33 | 69.3 | 30.8 | 1053.9 | 212.0 | 36.7 | 8.3 | 119.8 | 51.1 |
| R34 | 69.1 | – | 3962.2 | – | 148.9 | – | 314.9 | – |
| R35 | 65.8 | – | 3351.6 | – | 76.3 | – | 139.9 | – |
| R36 | 108.0 | 120.2 | 2633.9 | 4634.0 | 49.9 | 48.0 | 289.9 | 228.8 |

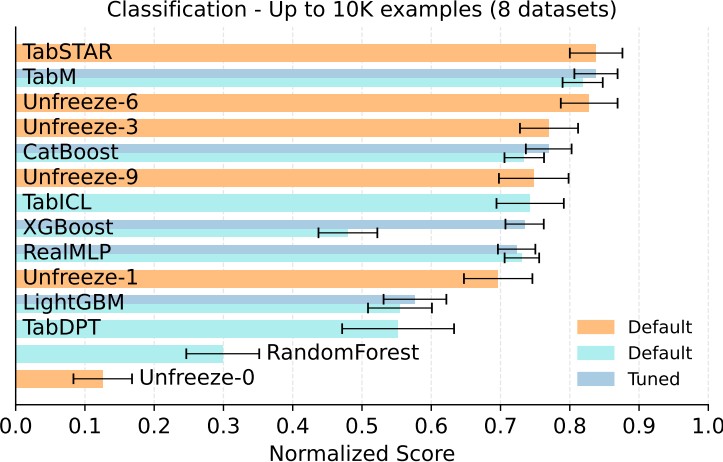

Figure 6: Comparison of normalized scores with 95% CIs for the encoder unfreezing experiment, comparing TabSTAR variants against baseline models in classification tasks.

Table 28: Downstream performance for Q1: The Role of the Encoder Unfreezing. Results for 20 datasets with 95% CI, for varying number of unfrozen layers. The top performance score is bolded first, and then all scores are rounded. We report AUROC for classification and $R^2$ for regression.

| ID | 0 | 1 | 3 | 6 | 9 |
|----|-----|-----|-----|-----|-----|
| C01 | 87.8±0.3 | 90.4±0.4 | 90.8±0.4 | **91.0±0.4** | 90.8±0.4 |
| C02 | 94.9±1.2 | 97.8±0.3 | 97.8±0.3 | **98.1±0.3** | 97.7±0.3 |
| C03 | 77.6±0.9 | 82.4±0.2 | 82.8±0.3 | **83.0±0.5** | 83.0±0.4 |
| C05 | 87.0±0.7 | 90.2±0.9 | 90.8±0.6 | **91.5±0.9** | 89.4±0.8 |
| C07 | 82.1±1.4 | 89.1±0.9 | 90.6±0.5 | **91.3±0.4** | 91.0±0.4 |
| C11 | 77.4±0.5 | 78.4±0.6 | 78.7±0.7 | **78.8±0.5** | 78.3±0.6 |
| C12 | 94.4±0.2 | **98.3±0.1** | 98.3±0.1 | 98.3±0.1 | 98.2±0.1 |
| C13 | 66.5±0.9 | 71.7±1.1 | 72.9±0.7 | **74.1±0.7** | 73.0±0.8 |
| R02 | 97.0±1.7 | 97.9±1.2 | **98.0±1.1** | 97.9±1.2 | 98.0±1.1 |
| R03 | 67.2±0.8 | 69.8±0.8 | 70.9±0.9 | **71.2±0.8** | 70.8±0.6 |
| R05 | 80.8±2.6 | 91.8±1.5 | **93.2±0.5** | 93.1±0.7 | 92.9±0.8 |
| R09 | 88.1±0.7 | 88.7±0.7 | **89.0±0.5** | 89.0±0.6 | 88.9±0.8 |
| R12 | 97.0±0.6 | 98.2±0.3 | 98.2±0.2 | **98.3±0.3** | 98.2±0.3 |
| R13 | 37.6±3.6 | 45.7±2.1 | 51.6±3.2 | 51.4±2.9 | **52.1±1.8** |
| R18 | 96.3±1.1 | 97.1±0.6 | **97.4±0.6** | 97.3±0.6 | 97.0±0.6 |
| R23 | 79.3±2.4 | 81.6±1.6 | 82.8±1.6 | **83.2±2.2** | 82.1±1.9 |
| R27 | 81.3±1.6 | 81.3±1.6 | **81.5±1.6** | 81.3±1.7 | 81.4±1.6 |
| R30 | 14.2±4.6 | 15.6±5.6 | **19.2±3.1** | 19.1±4.0 | 17.3±4.5 |
| R33 | 36.0±2.0 | 43.6±1.5 | 44.3±1.4 | **46.0±1.8** | 43.5±2.2 |
| R34 | 84.3±2.8 | 87.4±2.9 | 88.4±3.4 | 87.9±3.5 | **88.7±2.7** |

Table 29 show the normalized results for classification and regression, and Table 30 shows the dataset level results. It is evident that for most of the datasets, improvement is observed when scaling, with the 256 datasets variant winning for almost all datasets. Figure 7 highlights this effect, showing that adding more datasets enables TabSTAR to outperform the baselines.

### G.4 Numerical Verbalization (Q3)

We show the full results for the experiment in §6, with Table 31 illustrating the verbalizations in each variant. Note that we do not include an exact value verbalization, since it would increase the number of unique text inputs and place extra memory demands. The two variants which integrate numerical information into the verbalization dominate the experiment, although the improvement seems to be

Table 29: Normalized score with 95% CIs by the number of datasets used during TabSTAR pretraining.

| Pretraining Datasets | 0 | 16 | 64 | 256 |
|---|---|---|---|---|
| Classification | $0.352 \pm 0.086$ | $0.450 \pm 0.084$ | $0.558 \pm 0.086$ | $0.786 \pm 0.076$ |
| Regression | $0.338 \pm 0.073$ | $0.395 \pm 0.068$ | $0.642 \pm 0.066$ | $0.811 \pm 0.055$ |

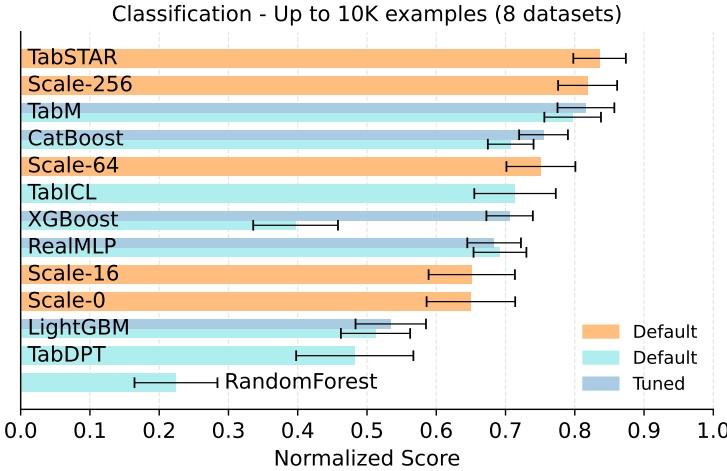

Figure 7: Comparison of normalized scores with 95% CIs for the pretraining effect experiment, comparing TabSTAR variants against baseline models in classification tasks.

marginal for some datasets. Interestingly, some datasets significantly underperform, with the *R27* dataset completely failing the task. The addition of the quantile information on top of the bin seems to have limited impact, although marginally winning on the average performance. Figure 8 presents a comparison with baseline models, highlighting the importance of incorporating richer numerical verbalizations beyond relying solely on the column name.

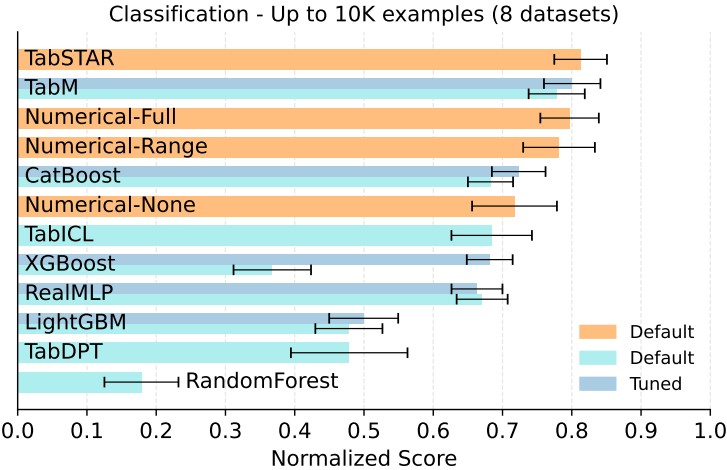

Figure 8: Comparison of normalized scores with 95% CIs for the numerical verbalization experiment, comparing TabSTAR variants against baseline models in classification tasks.

Table 30: Downstream performance for Q2: The Effect of Pretraining. Results for 20 datasets with 95% CI, for varying number of pretraining datasets. The top performance score is bolded first, and then all scores are rounded. We report AUROC for classification and $R^2$ for regression.

| ID | 0 | 16 | 64 | 256 |
|----|-----|-----|-----|-----|
| C01 | 90.8±0.4 | 90.7±0.4 | 90.7±0.3 | **91.0±0.4** |
| C02 | 98.0±0.4 | 97.4±0.8 | 97.8±0.3 | **98.1±0.3** |
| C03 | 69.2±8.9 | 83.1±0.4 | **83.2±0.3** | 83.0±0.5 |
| C05 | 90.7±0.8 | 90.3±1.1 | 90.6±0.5 | **91.5±0.9** |
| C07 | 87.9±0.5 | 87.6±0.8 | 90.9±0.3 | **91.3±0.4** |
| C11 | 78.2±0.6 | 77.4±0.9 | 78.0±0.4 | **78.8±0.5** |
| C12 | **98.3±0.1** | 98.2±0.1 | 98.2±0.1 | 98.3±0.1 |
| C13 | 74.0±0.5 | 73.5±0.8 | 73.7±1.0 | **74.1±0.7** |
| R02 | 97.2±1.6 | 97.6±1.3 | 97.9±1.2 | **97.9±1.2** |
| R03 | 67.3±1.9 | 68.5±1.0 | 71.2±0.7 | **71.2±0.8** |
| R05 | 88.0±2.8 | 90.6±2.2 | 92.2±1.0 | **93.1±0.7** |
| R09 | 88.1±1.0 | 88.4±0.7 | 88.8±0.5 | **89.0±0.6** |
| R12 | 97.7±0.3 | 97.9±0.3 | 98.1±0.2 | **98.3±0.3** |
| R13 | 49.4±2.7 | 49.1±3.6 | 48.0±3.0 | **51.4±2.9** |
| R18 | 95.0±2.2 | 94.9±1.2 | 96.7±0.6 | **97.3±0.6** |
| R23 | 82.2±1.6 | 81.2±2.3 | 81.8±2.3 | **83.2±2.2** |
| R27 | 80.9±1.7 | 81.3±1.6 | **81.5±1.6** | 81.3±1.7 |
| R30 | 13.1±4.1 | 18.5±4.9 | 18.5±3.9 | **19.1±4.0** |
| R33 | 45.4±2.2 | 42.8±3.3 | 45.0±1.5 | **46.0±1.8** |
| R34 | 83.8±4.3 | 86.0±3.3 | **88.0±3.4** | 87.9±3.5 |

Table 31: Illustrative verbalization of a numerical feature (*Age*) for the Q3: Numerical Verbalization experiment.

| Value | Name | Name + Bin | TabSTAR |
|-------|------|------------|---------|
| 17 | Age: Numeric | Age: Lower than 18 | Age: Lower than 18 (Quantile 0%) |
| 20 | Age: Numeric | Age: 18–23 | Age: 18–23 (Quantile 0–10%) |
| 25 | Age: Numeric | Age: 23–27 | Age: 23–27 (Quantile 10–20%) |
| 29 | Age: Numeric | Age: 27–31 | Age: 27–31 (Quantile 20–30%) |
| 33 | Age: Numeric | Age: 31–35 | Age: 31–35 (Quantile 30–40%) |
| 38 | Age: Numeric | Age: 35–40 | Age: 35–40 (Quantile 40–50%) |
| 42 | Age: Numeric | Age: 40–45 | Age: 40–45 (Quantile 50–60%) |
| 48 | Age: Numeric | Age: 45–51 | Age: 45–51 (Quantile 60–70%) |
| 55 | Age: Numeric | Age: 51–58 | Age: 51–58 (Quantile 70–80%) |
| 63 | Age: Numeric | Age: 58–67 | Age: 58–67 (Quantile 80–90%) |
| 83 | Age: Numeric | Age: 67-87 | Age: 67–87 (Quantile 90–100%) |
| 93 | Age: Numeric | Age: Higher than 87 | Age: Higher than 87 (Quantile 100%) |
| – | Age: Unknown Value | Age: Unknown Value | Age: Unknown Value |

Table 32: Downstream performance for Q3: Numerical Verbalization. Results for 20 datasets with 95% CI, for different verbalizations. The top performance score is bolded first, and then all scores are rounded. We report AUROC for classification and $R^2$ for regression.

| ID | Name | Name + Bin | TabSTAR |
|---|---|---|---|
| C01 | 91.0±0.4 | **91.2±0.4** | 91.0±0.4 |
| C02 | **98.1±0.2** | 97.9±0.3 | 98.1±0.3 |
| C03 | 83.3±0.4 | **83.3±0.3** | 83.0±0.5 |
| C05 | 90.8±0.9 | 91.1±1.2 | **91.5±0.9** |
| C07 | 91.2±0.2 | **91.4±0.4** | 91.3±0.4 |
| C11 | 78.2±0.5 | 78.2±0.6 | **78.8±0.5** |
| C12 | 98.2±0.1 | **98.4±0.1** | 98.3±0.1 |
| C13 | 71.6±0.7 | 73.8±0.9 | **74.1±0.7** |
| R02 | **98.0±1.1** | 98.0±1.1 | 97.9±1.2 |
| R03 | 67.4±0.7 | **71.3±0.6** | 71.2±0.8 |
| R05 | 93.1±1.1 | **93.2±0.8** | 93.1±0.7 |
| R09 | 88.9±0.5 | 88.9±0.6 | **89.0±0.6** |
| R12 | 98.2±0.2 | **98.3±0.2** | 98.3±0.3 |
| R13 | 50.1±3.3 | 49.7±3.4 | **51.4±2.9** |
| R18 | 97.1±0.6 | 97.1±0.7 | **97.3±0.6** |
| R23 | 81.9±2.4 | 82.7±1.9 | **83.2±2.2** |
| R27 | 16.7±6.6 | **82.0±1.5** | 81.3±1.7 |
| R30 | 16.4±4.4 | 17.7±5.3 | **19.1±4.0** |
| R33 | 45.6±1.1 | **46.2±2.2** | 46.0±1.8 |
| R34 | 88.0±3.6 | **88.6±2.5** | 87.9±3.5 |

