# OpenReview forum: "TabSTAR: A Tabular Foundation Model for Tabular Data with Text Fields"
_NeurIPS.cc/2025/Conference — NeurIPS 2025 poster_

### Official Review · Reviewer_qMRw · 2025-06-24

**Clarity:** 3
**Significance:** 3
**Originality:** 3
**Rating:** 5
**Confidence:** 3

**Summary:**

The paper introduces a new model called TabSTAR. The model achieves competitive
performance compared to GBDTs, neural nets and in context learner models on
both regression and classification tasks. This is done via a two phase
pre-train and finetune process. In terms of architecture, an unfrozen language
model is used, with layers for fusion, interaction and prediction. A novel
approach is the use of target-aware tokens that enable the model to work with
any number of classes. Finally, the paper justifies the architecture choices
through experiments.

**Questions:**

- Are the final models trained with 6 unfrozen layers for the text encoder?
- The NLP community has been using LoRA for finetuning, why the authors choose
  to unfreeze the text encoder instead? Note that LoRA is already used for
  finetuning stage. Figure 4 shows that 9 unfrozen layers underperforms 6
  unfrozen layers. It seems like overfitting is a concern (validation is lower
  but test is not). Note that the amount of text here is tiny compared to LLM's
  pretraining dataset and things like catastrophic forgetting can be
  concerning.

**Ethical Concerns:**

["NO or VERY MINOR ethics concerns only"]

**Final Justification:**

I originally liked the paper and gave it rating 5. After discussion with authors and reading other reviews I still like the paper and maintain my score.

**Limitations:**

Yes

**Paper Formatting Concerns:**

The formatting is fine.

**Quality:**

3

**Strengths And Weaknesses:**

## Quality

- The experiments are thoroughly conducted with lots of results in the
  Appendix that show competitive performance and justify many choices
  made for training the model. Furthermore, the Appendix highlights some
  efforts that did not lead to significant results, which will help the
  community explore the space more efficiently.
- The paper's claim about being a foundation model is not fully justified.
  Foundation models such as TabPFN, TabDPT, and TabICL achieve their results
  without finetuning on each dataset. This paper, however, uses five models
  that are separately trained (contrary to the foundation model claim) and then
  finetunes on each dataset. A fairer comparison would have been to use one
  model and still finetune on each dataset but with baselines using
  inference-time calibration (as done in LocaLPFN). Alternatively, more
  emphasis should be put on the inference-time compute required for each model,
  and that section should be in the main text, not the Appendix (F.3).

## Clarity

- The paper reads well and explains the ideas clearly and justifies most of
  them with experiments.
- The authors could touch upon the relation between number of classes and
  performance and efficiency, since target-aware tokens scale with the number of
  classes. This seems important as when comparing training dataset to
  evaluation dataset, one observes the training datasets have many examples
  with large number of classes while evaluation dataset (on which computational
  cost/performance analysis is done) lacks such datasets.
- In the Appendix A.3 Row-Level Attention is discussed. It seems like the
  authors tried in context learning approach. Although, the details are
  missing. What was the number of in context samples? Did the authors see any
  sign of overfitting as reported in TabDPT (which motivates their use of SSL
  methods)? This is a tangent in the Appendix and I am not raising this as a
  negative point about the paper. But I believe this approach is of interest to
  the community and adding the details to the Appendix would be valuable.


## Significance

- The model trains fast (<48h on A40), opening the door for community training
  their own variant.
- TabSTAR is good on classification tasks (possibly due to verbalization and
  the text encoder and target-aware token being more helpful here) but not as
  good in regression, even compared to TabPFNv2 which doesn't have per dataset
  finetuning (on less than 10K setting).


## Originality

- The architecture is novel.
- The method used to mix the verbalization with numeric values is novel.
- Target-aware tokens are novel.

---

> ### Author Rebuttal · Authors · 2025-07-30
>
> Dear reviewer qMRw,
>
> We appreciate your thorough review and the hard work, as well as the warm feedback. We believe that some of your points are highly valuable and will contribute to the camera-version paper. We especially appreciate that you've gone thoroughly over the additional experiments described in the appendix, and that you found them insightful and worthy.
>
> # Foundation Model Property
>
> > The paper's claim about being a foundation model is not fully justified. Foundation models such as TabPFN, TabDPT, and TabICL achieve their results without finetuning on each dataset.
>
> The term Foundation Model coined by Bommasani et al. in "On the Opportunities and Risks of Foundation Models", explicitly defines it in their first paragraph: "A foundation model is any model that is trained on broad data (generally using self-supervision at scale) that can be adapted (e.g., fine-tuned) to a wide range of downstream tasks; current examples include BERT". This message is supported in the paper "Why Tabular Foundation Models Should Be a Research Priority".
>
> This is not necessarily a weakness, but a careful design choice with clear upsides. TabPFN and other ICL methods have the magical property of working "out of the box" without training, as they don't do parameter update. However, this comes with the price of having limited context size (and thus can process up to 10,000 examples). While some engineering efforts led to TabICL being able to process 100,000 examples, this method would likely not be able to scale to millions of examples. In addition, inference is much more complex and comes with a huge price of latency/memory, as every example attends the whole dataset.
>
> We believe that this approach is not necessarily the right one, and definitely not the only valid one. Our model does finetune, yes, but its inference is similar to GBDTs in terms of latency, which can be a crucial aspect in realistic scenarios. While our training is more expensive, for most datasets we can efficiently tune it thanks to the usage of LoRA, and we are faster than GBDTs as we offer out-of-the-box SOTA performance without tuning hyperparameters.
>
> > This paper, however, uses five models that are separately trained (contrary to the foundation model claim) and then finetunes on each dataset.
>
> This is inaccurate, and we were probably not clear enough. Every downstream task is only finetuned once, using only one of the 5 pretrained models. We write in the paper:  *"To maximize the value of cross-dataset learning, instead of pretraining TabSTAR once, we create five dataset splits. Each variant is pretrained on the 350 pretraining datasets and 40 of the benchmark datasets, while the other 10 serve exclusively as its test set."*.
>
> We understand the confusion and we probably not clear enough, and also our evaluation setup is indeed a bit complex. For some additional details, you could read the section "TabSTAR Eval Complexity" in our response to reviewer Cb5W
>
> As a side note, please note that TabSTAR is actually underoptimized during training/inference time: both tuned baselines, TabPFN-v2 and TabICL, fit more than one model and average predictions. We consider this optimization worthy, and definitely necessary for real-world production use-cases, but we believe that foundation models should be easy to use and we preferred to avoid such internal optimizations to make the model as light as we can for future benchmarking.
>
> # Inference-time compute
>
> > Alternatively, more emphasis should be put on the inference-time compute required for each model, and that section should be in the main text, not the Appendix (F.3).
>
> We thank you for this comment and it will be well addressed, and we will include a thorough analysis as part of the main text. To justify our rationale, we would add that we were driven by peak performance, and that TabSTAR could be optimized further with inference efficiency in mind. Nevertheless, this request come from multiple reviewers and we took it very seriously. We changed our code to meticulously log CPU/GPU memory and running times for both training and inference time. This analysis will complement the results section in the main paper.
>
> Due to the rebuttal time constraints, we can only share a partial, qualitative analysis of TabSTAR vs the best open-source representative for other categories: CatBoost, RealMLP and TabICL. We present here our two main conclusions so far:
>
> - GPU vs CPU: We found that for tabular data with text features that require embedding models, GBDTs must use a GPU to be competitive in terms of latency. Serving embeddings models with CPU-only makes both training and inference orders of magnitude slower. This claim is very important and non-trivial, as it narrows the gap from more efficient methods.
>
> - While TabSTAR is GPU memory-hungry during training (3x more than others), it is memory efficient with comparative memory and latency to GBDTs. On the contrary, while TabICL has virtually no training time, its inference is prohibitively slower (~10x). As the community rapidly adapts ICL as the leading method for doing Foundation Models, the upside of no training time comes with the cost of GPU-based low-latency/high-memory inference. TabSTAR, on the contrary, is inference-time efficient.
>
> # Multiclass performance
>
> > The authors could touch upon the relation between number of classes and performance and efficiency, since target-aware tokens scale with the number of classes.
>
> Valid concern. Since this problem is very similar to the "multiple features" scenario as pointed by reviewer mFUa, we analyze your question with that mindset. If you consider it insufficient, we can specifically benchmark number of classes as well for the camera-ready paper.
>
> We performed a partial analysis over groups of "<10", "30-50" and "100-200" features. We observe that transformer based models scale worse, a problem shared by both TabSTAR and TabICL. It seems that metrics increase up to 5x times in a multi-feature scenario, and become infeasible or too costly with 200+ datasets for TabSTAR. While this is a problem for virtually all foundation models (as they all use transformers), we agree that TabSTAR is also sensitive to the number of classes, and this is an additional limitation we should mention in the paper, and future work material.
>
> # Row-level Attention
>
> > In the Appendix A.3 Row-Level Attention is discussed. It seems like the authors tried in context learning approach.
>
> For a long time we explored using row-level attention as presented in SAINT, i.e. batches of samples of the dataset while still doing finetuning. We never took the ICL path. While we believe that this direction is interesting, and especially appealing for small datasets, we wanted to put emphasis on big datasets to show that foundation models should scale.
>
> > What was the number of in context samples? Did the authors see any sign of overfitting as reported in TabDPT (which motivates their use of SSL methods)?
>
> Anecdotically, and here we rely on memory only as this direction was neglected, we did see signs of overfitting during pretraining, something relatively rare when shuffling between many datasets over different samples of the data. We experimented with batches of very different sizes both during pretraining and finetuning. Eventually, as we've seen limited value and too much overhead (overfitting, unclear batch size during both phases, unclear inference method) we decided to leave this path completely, and we currently believe that finetuning with row-level attention is too complex, and thus was probably not widely adopted: it should be either IID finetuning like TabSTAR, or pure ICL like TabPFN et. al.
>
> # Regression Performance
>
> > TabSTAR is good on classification tasks ... but not as good in regression, even compared to TabPFNv2 which doesn't have per dataset finetuning (on less than 10K setting).
>
> Although this is not explicitly stated as a weakness, and although we believe that this is definitely an area for improvement, we would like to highlight that TabPFN-v1 didn't support regression tasks and that TabPFN-v2 was trained over 130M+ datasets, carefully generated to simulate a wide range of numerical distributions. We believe that given the right data, TabSTAR could be scaled, as its architecture lends itself to that and shows consistent improvement, also for regression tasks.
>
> # Frozen Layers
>
> >Are the final models trained with 6 unfrozen layers for the text encoder?
>
> During pretraining, the final models unfreeze the 6 top layers. During finetuning, LoRA is applied to this layers. We neither touch the other bottom 6 layers, nor the word embeddings.
>
> > The NLP community has been using LoRA for finetuning, why the authors choose to unfreeze the text encoder instead? Note that LoRA is already used for finetuning stage.
>
> The term "unfrozen" is misleading and our fault, we'll rephrase. During pretraining, these 6 layers are completely unfrozen. During finetuning, we apply LoRA, so the pretrained weights are actually frozen. We used the term unfrozen to differentiate them from the bottom 6 layers that remain completely frozen without LoRA, of from other methods that don't tune (in any way) these embeddings. But this was unclear, thanks!
>
> > Figure 4 shows that 9 unfrozen layers underperforms 6 unfrozen layers. It seems like overfitting is a concern (validation is lower but test is not). Note that the amount of text here is tiny compared to LLM's pretraining dataset and things like catastrophic forgetting can be concerning.
>
> Very valid point. We believe that with more data we'll pretrain the whole text encoder, and that when using a stronger base language model, we'll need to add stronger regularization. This is aligned with some of our thoughts for future works, thank you!

---

> > ### Comment · Reviewer_qMRw · 2025-08-03
> >
> > Thanks for the explanations. I have a better understanding of the method now. I think this is a good paper and I will keep my high score.
> > The only points remaining are about the foundation model aspect:
> > My general point was  regarding comparison with other such models and the fairness there. Models like TabPFN can adapt to new datasets without any fine-tuning. Regarding large datasets, things like indices can be used (TabPFN-KNN from LocaLPFN or TabDPT). If we have more time and resources and want to fine-tune, we can use the full method in LocaLPFN for a fair comparison. But in this case fine-tuning is optional and enhances performance as opposed to a necessary step for each new dataset. Note that these methods also allow handling large datasets.
> > Regarding the five model ensemble, my point is the following: Assume  all these five models are released and there is a new dataset, which model should be used on it? It seems like an ensemble would be an option. But non of these five models are the foundation model.

---

> ### Author Response · Authors · 2025-08-04
>
> Thank you for your kind words. We are happy that our response makes sense, and hope it raises your confidence that we deserve your high score.
>
> We understand your claim about comparison and mostly agree: definitions aside, it’s a fact that TabSTAR’s finetuning is not optional, and that models could be tuned even further. We want to add that we aimed for fairness by not optimizing ourselves per downstream dataset, i.e. used default hyperparameters and no model ensembling. As mentioned, we will add to the main paper an analysis of the trade-offs which will emphasize your valid point.
>
> Regarding your second question: We plan to release a single model pretrained over 400 datasets, which should be finetuned (once) for any novel downstream task. For datasets appearing in the corpus, evaluation should be done using the matching model that excludes it from its pretraining (only one of the 5 per dataset). We would like to reiterate that there are no ensembles, and in each case, we only use a single model. This complexity was added to allow our model to be fairly evaluated over benchmarks, as opposed to other models for them this was impossible. We hope this can become a best practice for models pretrained on real datasets.

---

### Official Review · Reviewer_SJjF · 2025-06-29

**Clarity:** 3
**Significance:** 2
**Originality:** 2
**Rating:** 4
**Confidence:** 4

**Summary:**

The paper proposes a tabular foundation model for datasets with textual features. Both numeric and textual features are summarized into verbalizations together with targets and pass through an encoder to build vector representations. These representations are then passed through a second interaction encoder that attends over all inputs. Finally, attended representations of targets are used to apply the regression/classification heads and generate predictions. Results on real-world datasets with textual features indicate that this approach can generalize to unseen tasks.

**Questions:**

-Is fine-tuning require for each downstream task? Table 3 shows the impact of pre-training but I couldn't find results where full pre-training is done but model is then applied as is without fine tuning. I think these results are important to show.
-Why is regression performance so weak? Authors say that "regression is not our main focus" but there are quite a few regression datasets in pre-training yet the model is not generalizing well after. Analysis on why that happens would be useful to see.

**Ethical Concerns:**

["NO or VERY MINOR ethics concerns only"]

**Limitations:**

Yes

**Quality:**

3

**Strengths And Weaknesses:**

Strengths:
-The paper is well written and easy to follow. It proposes an innovative approach to represent numerical data through verbalizations of quantile binning.
-Proposed architecture seamlessly integrates targets without making any assumptions on cardinality and LLM embeddings for textual data with end-to-end fine-tuning.
-Experiments on real-world data show promising results on generalization against some of the leading baselines.

Weaknesses:
-Novelty is somewhat limited, other than numeric data verbalizations, the rest of the architecture are fairly standard components that have been explored.
-Proposed model requires fine-tuning for each downstream task and is not really a "foundation" model like TabPFN where no finetuning is needed.
-Experimental results are mixed, even though TabSTAR has almost 100 regression datasets in pre-training, the regression performance is  quite weak.

---

> ### Author Rebuttal · Authors · 2025-07-30
>
> Dear reviewer SJjF,
>
> We thank you for reviewing our work and for acknowledging our strengths. We found your questions interesting, and we are sure that they will lead to a fruitful discussion. We believe that some of your questions should have been explained more clearly in the paper, and we plan to improve the camera-ready version accordingly. We hope that we can also highlight some of our strengths which perhaps were not noted enough, and turn some of your perceived weaknesses into actual strengths, resulting into an even higher rating of our work.
>
> # Novelty
>
> > Novelty is somewhat limited, other than numeric data verbalizations, the rest of the architecture are fairly standard components that have been explored.
>
> We humbly believe that our work contributes a few more important novelties, and we would like to highlight them:
>
> 1. Target-Aware Tokens: We introduce "target aware tokens", a novel component that cleverly introduces semantics about the target values and can build a representation from its early layers accordingly. This component has interesting properties both for ICL approaches, as well as for models such as XTab or Tp-BERTa which used dataset-specific prediction heads, making pretraining in scale infeasible. We would like to emphasize that each one of the other 4 reviewers specifically mentioned that this concept is creative and novel.
>
> 2. Semantically Target-Aware Represenations: we prove that TabSTAR could have been a fairly limited model had it used frozen, static embeddings as an initialization point, as done by Tp-BERTa and CARTE, which are semantically-aware foundation models. We believe that this misses the potential and show it even with a small text encoder and a fairly limited amount of data. As a bonus, this actually leads to SOTA performance on classification tabular tasks with text.
>
> 3. We are perhaps the only tabular foundation models which puts emphasis on free text, rather than strings. We can note that CARTE shines on categorical, high-cardinality features, and that Tp-BERTa encode examples as sequences. None of these works put multimodal-ish free-text datasets as a research priority, and we believe that true tabular foundation models should be able to properly handle these very common type of datasets which were historically neglected.
>
> 4. Our architecture is designed to scale over downstream big datasets, without any special limitations. This is in contrary to the popular ICL approach which shines on small-medium datasets, but struggle as they grow. We believe that we are the only foundation model that exhibits such behavior.
>
> To summarize, we truly believe this paper has actually high novelty and this weakness should be recognized as a strength. We hope that our arguments make sense and we'd be glad to engage in further discussion.
>
> # Foundation Model Properties
>
> > Proposed model requires fine-tuning for each downstream task
>
> This is not necessarily a weakness, but a careful design choice with clear upsides. TabPFN and other ICL methods have the magical property of working "out of the box" without training, as they don't do parameter update. However, this comes with the price of having limited context size (and thus can process up to 10,000 examples). While some engineering efforts led to TabICL being able to process 100,000 examples, this method would likely not be able to scale to millions of examples. In addition, inference is much more complex and comes with a huge price of latency/memory, as every example attends the whole dataset.
>
> We believe that this approach is not necessarily the right one, and definitely not the only valid one. Our model does finetune, yes, but its inference is similar to GBDTs in terms of latency, which can be a crucial aspect in realistic scenarios. While our training is more expensive, for most datasets we can efficiently tune it thanks to the usage of LoRA, and the fact that we achieve out-of-the-box SOTA performance without tuning hyperparameters.
>
> > is not really a "foundation" model like TabPFN where no finetuning is needed.
>
> We believe that this statement is not accurate.
>
> The term Foundation Model coined by Bommasani et al. in "On the Opportunities and Risks of Foundation Models", explicitly defines it in their first paragraph: *"A foundation model is any model that is trained on broad data (generally using self-supervision at scale) that can be adapted (e.g., fine-tuned) to a wide range of downstream tasks; current examples include BERT"*. This message is supported in the paper "Why Tabular Foundation Models Should Be a Research Priority".
>
> While we are used to having foundation models that require no finetuning in other modalities, it's unclear whether this property is a requisite for Tabular Foundation Models, especially as we are still in early days.
>
> > Is fine-tuning require for each downstream task? Table 3 shows the impact of pre-training but I couldn't find results where full pre-training is done but model is then applied as is without fine tuning. I think these results are important to show.
>
> TabSTAR indeed requires finetuning per downstream task, and this is a feature, not a bug. We believe that updating the parameters of the network makes it more practical for inference purposes and purposely avoided using ICL. Having said that, we believe that future versions of TabSTAR, being trained with 130M+ datasets like TabPFN and potentially leveraging larger language models, could potentially be flexible enough to both support ICL (for very small datasets) and do finetuning (for large ones), but we leave this as an exciting future area of research.
>
> We are open to hear your thoughts about this matter, and we believe that this approach should have been presented with more emphasis in the paper, and will emphasize this at the beginning of the "Tabular Foundation Models" paragraph in section 2.
>
> # Model Performance
>
> > Experimental results are mixed
>
> While the current model of TabSTAR underperforms in regression tasks (for now), we respectfully believe that this should not shadow that the model achieves SOTA for classification, and these are excellent results. We believe that foundation models should be able to tackle both tasks, although we could easily publish a classification-only model, like TabPFN-v1 or TabICL. We had the courage to publish the regression results, and we believe but this shouldn't stand against our work.
>
> > even though TabSTAR has almost 100 regression datasets in pre-training, the regression performance is quite weak.
>
> We would like to remind that TabPFN-v2 was trained over 130M+ datasets, carefully generated to simulate a wide range of numerical distributions. Pretraining over 100 datasets is a very modest decision, constrained by both budget and data availability. We believe that given the right data, TabSTAR could be scaled, as its architecture lends itself to that and shows consistent improvement, also for regression tasks.
>
> > Why is regression performance so weak? Authors say that "regression is not our main focus" but there are quite a few regression datasets in pre-training yet the model is not generalizing well after. Analysis on why that happens would be useful to see.
>
> We acknowledge that this is a weakness and area for improvement. We discuss in the paper several potential reasons:
>
> 1. Lack of data: we use ~100 regression datasets, this is probably not enough to generalize and transfer learn this complex task. Scaling the training set with millions of synthetic regression tasks (e.g. TabICL's public data) could provide a huge impact.
>
> 2. Better numerical modeling: we used a naive single transformation for numerical features, which potentially limits TabSTAR's numerical capabilities. We believe this could be challenged, similarly to how "On embeddings for numerical features in tabular deep learning" expanded FT-Transformer.
>
> 3. Regression-as-classification: discretizing the target value and using binned (target) tokens, similarly to CHRONOS and other TS FMs.
>
> To summarize: yes, TabSTAR's performance in regression is not great, but this is only an exciting area for future research. It's unlikely to expect a new architecture at small scale to shine for every problem. To demonstrate that, we would like to quote the "Conclusions & Future Work" section of TabPFN-v1, which we believe is super-transparent about its weaknesses and limitations, and that papers should be valued for that spirit:
>
> *"The TabPFN still has important limitations... which motivates work on (1) scaling up to large datasets.. (2) improved handling
> of categorical features... (3) missing values... (4) robustness to unimportant features... (9) regression tasks"*
>
> We hope that we have convinced you that our performance is a strength, and not a weakness, and are open to discuss it further.

---

### Official Review · Reviewer_mFUa · 2025-07-01

**Clarity:** 3
**Significance:** 2
**Originality:** 3
**Rating:** 4
**Confidence:** 4

**Summary:**

This paper proposes TabSTAR, a tabular foundation model that integrates the semantic understanding capabilities of language models. Its core innovation lies in the introduction of target-aware tokens, where all possible target values are included as input tokens, and logits are generated for each of them.

**Questions:**

1. Please clarify how TabSTAR can be scaled to larger dataset volumes, especially in scenarios where labeled data is scarce. (W1)

2. Provide time and memory usage analysis to assess computational efficiency. (W2)

3. Include a rationale for encoder unfreezing, and describe how the number of unfrozen layers is selected. (W3)

4. Add an ablation study of each component. Also consider evaluating whether the current architecture performs better than a simpler version that just stacks the two modules with equivalent Transformer layers. (W4)

5. Improve the evaluation. Instead of only fine-tuning, test common unsupervised evaluation settings - such as aggregating embeddings and training XGBoost or linear probes on top. Also include comparisons with few-shot or unsupervised tabular learning methods such as SimSiam [1], SCARF [2], STAB [3], MET [4], STUNT [5], LFR [6], and TST-LLM [7]. If finishing these experiments is hard in time, at minimum, the paper should include a methodological comparison and explain why such methods were not used. (W5, W6)

[1] SimSiam: Exploring Simple Siamese Representation Learning (CVPR 2021)

[2] SCARF: Self-supervised Contrastive Learning Using Random Feature Corruption (ICLR 2022)

[3] STAB: Self-supervised Learning for Tabular Data (NeurIPS Workshop 2022)

[4] MET: Masked Encoding for Tabular Data (NeurIPS Workshop 2022)

[5] STUNT: Few-shot Tabular Learning with Self-generated Tasks from Unlabeled Tables (ICLR 2023)

[6] LFR: Self-supervised Representation Learning from Random Data Projectors (ICLR 2024)

[7] TST-LLM: LLM-Guided Self-Supervised Tabular Learning With Task-Specific Pre-text Tasks (TMLR 2025)

**Ethical Concerns:**

["NO or VERY MINOR ethics concerns only"]

**Final Justification:**

After reviewing the authors' rebuttal, I have decided to raise my score. The responses were detailed and addressed most of my initial concerns:

* Framing as Supervised Learning: While the authors’ perspective is understandable, the use of terms like "pretraining" made the approach appear close to tabular representation learning, prompting my request for comparisons to related baselines. Although I still find this boundary somewhat unclear, I acknowledge that this issue is secondary given the resolution of other major points. I encourage the authors to clarify this distinction in the paper to avoid potential confusion.

* Scalability: This concern is resolved.

* Encoder Unfreezing: The rationale provided is reasonable, and the authors’ observation regarding training stability would be a valuable addition to the paper.

* Fusion vs. Interaction: This concern is resolved.

Overall, while I retain some reservations regarding the positioning of baselines, all other concerns were thoroughly addressed. The paper has improved in clarity and scope as a result of the rebuttal, and I have updated my score accordingly.

**Limitations:**

Yes

**Paper Formatting Concerns:**

No concerns

**Quality:**

2

**Strengths And Weaknesses:**

**Strengths**
1. The problem addressed in this paper is important, and the high-level direction is well-motivated.

2.The introduction of target-aware tokens is a meaningful and creative idea.

**Weaknesses**
1. The proposed method relies entirely on supervised pretraining. In practice, this raises concerns about how the model handles datasets without labels. There is a reason that language foundation models typically use next-token prediction with unlabeled data. If TabSTAR depends solely on supervised learning, scalability may be limited by labeled data availability. For example, TabPFN mitigates this issue using synthetic data. It is unclear whether TabSTAR employs a novel strategy to address this limitation.

2. The method may struggle when the number of features grows large (e.g., 100+). As each feature is embedded (e.g., into 128 dimensions), memory usage increases rapidly, especially when applying Transformer layers for both fusion and interaction. This could raise scalability concerns.

3. The paper repeatedly emphasizes the importance of encoder unfreezing, but offers little explanation beyond empirical evidence. A theoretical rationale would help justify this design choice. If such justification is not possible, it may be better to reduce the emphasis. Moreover, if the number of unfrozen layers is treated as a hyperparameter, the paper should clarify whether a principled selection strategy exists, or if it is simply tuned using a held-out validation set.

4. Both the fusion and interaction modules process embeddings through Transformer layers. It is unclear how the paper ensures that fusion occurs during the fusion stage, and interaction occurs during the interaction stage, given the architectural similarity.

5. The evaluation section underexplores the implications of pretraining. Since pretraining is the core contribution, it would be useful to assess the model under different usage paradigms beyond fine-tuning.

6. The set of baselines is limited. As a pretraining-based approach, TabSTAR should be compared against recent few-shot or unsupervised tabular learning methods. Currently, none are included.

---

> ### Author Rebuttal · Authors · 2025-07-30
>
> Dear reviewer mFUa,
>
> We thank you for taking the time to review our paper and raise weaknesses and questions, which we took very seriously. While we believe that some of your concerns are spot-on and led to fruitful additional experiments which will make the paper better, some of your questions and weaknesses confused us, and we are afraid that the paper was misunderstood and framed as a paper that belongs to a different research branch, affecting your feedback and rating accordingly.
>
> # TabSTAR: A model for supervised learning
>
> > As a pretraining-based approach, TabSTAR should be compared against recent few-shot or unsupervised tabular learning methods. Currently, none are included. Improve the evaluation ... include comparisons with few-shot or unsupervised tabular learning methods ... explain why such methods were not used.
>
> Before we specifically address each one of your questions, we want to clarify that TabSTAR is a (pretrained) model used for tabular supervised learning, which is one of the downstream tasks of a higher-level problem of tabular representation. Very relevant papers that are strictly connected to TabSTAR are:
>
> * Why Tabular Foundation Models Should Be a Research Priority
> * TabArena: A Living Benchmark for Machine Learning on Tabular Data
>
> As well as the papers of TabPFN, TabPFN-v2, TabICL, CARTE and Tp-BERTa. We highlight that NONE of them cites ANY of the 7 papers you mentioned. In addition, none of the other 4 reviewers asked anything that hints that these methods are closely relevant to our work. With that in mind, we will try to answer your questions.
>
> # Evaluation
>
> > Improve the evaluation... include comparisons with few-shot or unsupervised tabular learning methods ... at minimum, the paper should include a methodological comparison and explain why such methods were not used.
>
> Since we only pretrain with supervision, it's unlikely that TabSTAR could be competitive vs models that employ more suitable training objectives, like SCARF or STAB. Accordingly, to the best of our knowledge, this method aren't widely adopted for supervised tabular learning. We'll add to this distinction to the related work section, to make TabSTAR's scope clearer.
>
> # Supervised pretraining
>
> > The proposed method relies entirely on supervised pretraining. In practice, this raises concerns about how the model handles datasets without labels. There is a reason that language foundation models typically use next-token prediction with unlabeled data. If TabSTAR depends solely on supervised learning, scalability may be limited by labeled data availability.
>
> The pretraining relies on supervised learning as the model is used for downstream tasks that are supervised as well, and thus we don't expect to encounter unlabeled data as a downstream task. Still, it is true that scaling the pretraining might be hard due to lower availability of labeled data, and we write in the discussion: "*Data scaling might leverage self-supervised learning [59] over large-scale table corpora [17], or realistic synthetic tabular data generators [9], which have proven successful [37, 2]*".
>
> > Please clarify how TabSTAR can be scaled to larger dataset volumes, especially in scenarios where labeled data is scarce.
>
> For future work and data scaling, we have a few options, which are not mutually exclusive.
> 1. Adding even more labeled datasets, as growing from 400 by a factor of 10x can be possible with Kaggle.
> 2. Leveraging unlabeled datasets or tables in the wild by:
> a. Assigning a target variable that could be predicted by the other features.
> b. Choosing a SSL strategy, for which we will could explore the SSL objectives from the papers you mentioned.
> 3. Creating/leveraging synthetic labeled datasets, involving LLMs due to TabSTAR's semantic nature.
>
> > For example, TabPFN mitigates this issue using synthetic data. It is unclear whether TabSTAR employs a novel strategy to address this limitation.
>
> From their paper: *"Data generation: we define a generative process (referred to as our prior) to synthesize diverse tabular datasets with varying relationships between features and targets ... For each dataset, a subset of samples has their target values masked, simulating a supervised prediction problem."*
>
> TabPFN created synthetic datasets, which can easily be employed for supervised learning, and they would be perfect for training TabSTAR, up to the fact that they contain no semantic information. Nevertheless, it is perfectly possible that we could first pre-pretrain over TabPFN's datasets without semantics, and only then continue with semantic real datasets.
>
> # Number of features
>
> > The method may struggle when the number of features grows large (e.g., 100+) ... This could raise scalability concerns.
>
> This concern is valid, and indeed one of our limitations. As we state in the paper, during pretraining "*we randomly sample up to 200 features per dataset.*" and we mention in the discussion that "*it may struggle with memory constraints on datasets containing hundreds of features*". Indeed, the problem you mentioned is a common one for transformer based methods, and similarly to other transformer based foundation models, TabSTAR doesn't scale well with lots of features. This could be mitigated by future work investing in reducing the feature dimensionality.
>
> > Provide time and memory usage analysis to assess computational efficiency.
>
> As requested by other reviewers, we plan to include in the paper an analysis of runtime and memory consumption during both training and inference time. Please see the section "Computational Costs" in the response to reviewer Cb5W for more details.
>
> In addition and per your request, we'll make sure to also break-down the comparison into different groups of number of features. Since memory consumption depends on many factors, we'll need to run a wide variety of datasets for strong conclusions, but we can say that TabSTAR can't really scale above 200 features in a A40 GPU and that time and memory metrics grow by a factor of 3-5x when moving from the 30-50 features bin to the 100-200 bin.
>
> # Encoder unfreezing
>
> > Include a rationale for encoder unfreezing.
>
> Since word embeddings are trained to predict masked or next token, opposite concepts might be very close to each other in the embedding space, but for tabular learning we would like to have a very different geometric structure. Given a categorical feature, like country of birth, we would ideally like to eliminate both the shared information as well as differences that are irrelevant to our target variable. Empirically, we indeed observe that the geometry varies as we keep training, building a new "correlation space" that is more suitable for tabular learning. An evidence for this theory is that the average distance of strings dramatically drops during training, from +0.9 for almost any pair to much lower numbers.
>
> > Describe how the number of unfrozen layers is selected
>
> It seems that this detail is missing from the paper: for downstream, we apply LoRA on the 6 unfrozen layers from pretraining, which selection was explained in section 6, under "The Role of the Encoder Unfreezing".
>
> # Fusion vs Interaction
>
> > Both the fusion and interaction modules process embeddings through Transformer layers. It is unclear how the paper ensures that fusion occurs during the fusion stage, and interaction occurs during the interaction stage, given the architectural similarity.
>
> While the Fusion and Interaction blocks share architecture similarity, their different roles stem from their input. The Fusion block unifies the semantic and numerical vector into one vector, for each feature independently. This means that for every example, we use this block potentially multiple times. On the contrary, the Interaction block operates over all the elements (features and targets) and is applied only once. De-facto, the first block applies “intra-feature” attention and the second one applies “cross-feature” attention. Note that this means that the effective number of inputs for each block is very different (2 vs num features + target tokens). We acknowledge that this should have been clearer in the paper, and we will rephrase accordingly.
>
> > Add an ablation study of each component. Also consider evaluating whether the current architecture performs better than a simpler version that just stacks the two modules with equivalent Transformer layers.
>
> We would first want to highlight a pretraining experiment reported in Appendix A3 ("The Fusion Block’s Mechanism"). In this experiment we consider simpler approaches to replace the Fusion component. The pretraining validation metric is:
> * Fusion (0.8351)
> * Multiplication (0.8280)
> * Concatenation (0.8241)
>
> The concatenation approach causes losing 1% on average per dataset. While multiplication is a non-parametric simpler alternative, we believe that the fusion component is more flexible, will scale better and lends itself to multimodality (e.g., adding image features, or richer numerical representations).
>
> Following your concern, we conducted a new experiment and suggested two variants that effectively skip the fusion block:
> 1. “Numeric Only”: for numerical features, we avoid using any semantic information and go directly to the interaction block with a single vector per element.
> 2. “Double Representation”: instead of a separate block for fusing, we pass to the interaction encoder two elements for numerical features. Note that this approach inflates the number of "features", potentially by a factor of two, making it a problematic choice in terms of memory.
>
> The pretraining performance is:
>
> * Fusion & Interaction: 0.8404
> * Numeric Only: 0.8356
> * Double Representation: 0.8249
>
> To summarize, we claim that the fusion block has a strong rational as well as an empirical validation, and can even be seen as a novel approach. We believe that our arguments could transform this perceived weakness to a strength, and we hope that you agree.

---

> ### Comment · Reviewer_mFUa · 2025-08-03
>
> As noted in the final justification, the choice of baselines remains a point of hesitation for me. Even if not through additional experiments, I strongly encourage the authors to clarify the methodological boundary to distinguish their approach. Nonetheless, given the resolution of all other concerns, I have increased my score.

---

> > ### Author Response · Authors · 2025-08-04
> >
> > We are very glad that we have resolved almost all of your concerns and that you recommend accepting the paper. We humbly believe that this reflects the quality of our work.
> >
> >
> > As we stated, we believe that TabSTAR should primarily (or solely) be evaluated on a supervised regime, due to its objective pretraining type and scale. This is, as we demonstrated, very inline with other papers, and this shouldn't be considered as a  weakness of our work. We believe that future research could explore expanding TabSTAR to include also a SSL pre-pretraining stage, and then evaluate on those setups.
> >
> > We would also like to emphasize that following the request of reviewers Cb5W and 1N8m, we added TabICL, RealMLP and LightGBM to our evaluations, all significantly underperforming (see rebuttal of reviewer Cb5W under the header “Baselines”). By now, we have compared TabSTAR to TabPFN-v2, TabICL, CARTE, RealMLP, CatBoost, XGBoost, LightGBM and RandomForest. We feel that this evaluation suite is quite complete and resembles TabArena, a new leaderboard backed by leading tabular learning researchers.
> >
> > **We’ll make sure that TabSTAR’s scope is very clear in the revised related work section**, and believe future readers will benefit from it. We would love to keep discussing the subject, as we feel that your perception of our work as well as your rating could be even more positive if we solve this disagreement.

---

### Official Review · Reviewer_1N8m · 2025-07-02

**Clarity:** 2
**Significance:** 2
**Originality:** 2
**Rating:** 3
**Confidence:** 4

**Summary:**

The paper presents TabSTAR, a new foundation model designed for tabular datasets that contain text features. The central idea is to create more meaningful, task-oriented feature representations by making the model "target-aware." This is achieved by feeding the model not just the input features but also tokens representing the possible target values (for classification).

The architecture involves five main steps:

Verbalization: All features (numerical, categorical, and text) are converted into human-readable strings. For example, a numerical age of 45 might become "Age: 40-50 (Quantile 50-60%)".

Encoding: A pretrained text encoder (e5-small-v2) transforms these strings into embeddings. Crucially, some layers of this encoder are unfrozen, allowing them to be fine-tuned for the specific task.

Fusion: Embeddings for numerical features are combined with their semantic string representations.

Interaction: A Transformer encoder processes the embeddings of all features and target tokens, allowing them to interact and influence each other.

Prediction: A shared prediction head uses the final representations of the target tokens to output probabilities for classification or a value for regression.

TabSTAR is pretrained on 350 tabular datasets and then fine-tuned for specific downstream tasks. The authors claim that this approach achieves state-of-the-art performance on classification tasks for medium-to-large datasets containing text, outperforming Gradient Boosting Decision Trees (GBDTs) and other tabular foundation models.

**Questions:**

Novelty: Can you clarify the primary novelty of your verbalization and encoding method compared to prior work like TP-BERTa, which also tokenizes numerical and categorical features for a transformer?

Baselines: Why were other foundation models for tabular data, such as XTab, not included in the comparison? Their inclusion seems critical for validating the claim of state-of-the-art performance in the TFM space.

Model Capabilities: Since the model relies on an embedding model rather than a full LLM, what are the theoretical limits of its reasoning ability? Do you believe the performance gains come solely from adaptive embeddings, or is there evidence of more complex semantic understanding?

Regression Performance: The paper notes that performance lags GBDTs on regression tasks. Beyond scaling the pretraining data, are there specific architectural weaknesses that you believe cause this gap? For instance, is the verbalization strategy inherently less effective for continuous targets?

Target-Awareness in Regression: For regression tasks, a single target token representing the target's name is used. How does this provide meaningful "awareness" compared to the classification case, where all possible target values are provided as input?

**Ethical Concerns:**

["NO or VERY MINOR ethics concerns only"]

**Limitations:**

The paper rightly acknowledges some of its limitations in the discussion section. Based on the work presented, the key limitations are:

Weak Regression Performance: The model is not competitive with established GBDT models on regression tasks, indicating that the current architecture is not well-suited for predicting continuous values.

Dependence on an Embedding Model: The model's capabilities are constrained by the pretrained text encoder it uses. It does not possess the broad reasoning and knowledge integration abilities of a true LLM.

Incomplete Benchmarking: The lack of comparison against a fuller suite of contemporary tabular foundation models makes the SOTA claims difficult to fully validate.

Marginal Gains in Classification: The performance improvements in classification, while present, are not overwhelmingly large, and the overlapping confidence intervals with the next-best model (TabPFN-v2) suggest the advantage may not be statistically robust across all datasets.

**Paper Formatting Concerns:**

The paper is well-structured and clearly formatted. There are no significant formatting issues to report.

**Quality:**

2

**Strengths And Weaknesses:**

Strengths

Focus on Textual Features: The model is explicitly designed to handle tabular data with rich text, a common and important type of real-world data that has been historically underrepresented in tabular learning benchmarks.

Novel Target-Aware Architecture: The introduction of target-aware tokens is an interesting architectural choice. It allows the model to learn representations that are specifically tailored to the prediction task and enables an architecture free of dataset-specific parameters, which is a key advantage for a general-purpose foundation model.

Systematic Ablation Studies: The paper provides a good analysis of its key components, showing that unfreezing the text encoder and using detailed numerical verbalization both significantly improve performance. The results also demonstrate that performance scales with the number of pretraining datasets, supporting the foundation model approach.

Emphasis on Reproducibility: The authors provide details on their experimental setup and plan to release pretrained models and code, which is a valuable contribution to the research community.


Weaknesses
Limited Novelty: The core method of converting tabular data into text ("verbalization") and processing it with a transformer-based model has been explored in prior work (e.g., TP-BERTa, TransTab). The paper does not sufficiently distinguish its approach to establish clear novelty over these existing methods.

Incomplete Baselines: For a paper claiming to be a new Tabular Foundation Model (TFM), the experimental comparison is missing several important baselines from the same category, such as XTab. The comparison set is primarily GBDTs and TabPFN-v2, which doesn't fully represent the current TFM landscape.

Misleading Framing around "LLMs": The model does not use a Large Language Model (LLM) in the conventional sense (i.e., a model with billions of parameters and emergent reasoning abilities). It uses a much smaller text embedding model (e5-small-v2). This limits its ability to perform complex reasoning or leverage the vast world knowledge associated with true LLMs. The improvements likely stem from better-tuned embeddings rather than higher-level reasoning.

Unconvincing Performance Gains: The reported improvements are not consistently strong. TabSTAR is significantly outperformed by GBDTs in regression tasks. Even in its strong suit, classification, the performance gains over TabPFN-v2 are modest, with overlapping confidence intervals in the summary charts, suggesting the difference may not be statistically significant in all cases

---

> ### Author Rebuttal · Authors · 2025-07-30
>
> Dear reviewer 1N8m,
>
> We thank you for taking the time and effort, and for the constructive feedback. Some of your suggestions are key to improve the paper. While constructive, we believe we offer more strengths than weaknesses, and we hope to engage in a discussion to convince you so.
>
> # Novelty
>
> > Limited Novelty: ... The paper does not sufficiently distinguish its approach to establish clear novelty ... Can you clarify the primary novelty of your verbalization and encoding method compared to prior work like TP-BERTa?
>
> While we rely on some the foundations of TP-BERT, our work provides several strong novelties:
>
> 1. We introduce target-aware tokens, a novel component which adds semantic knowledge, enables full parameter sharing and unlocks pretrain scaling. On the contrary, Tp-BERTa specifically adds parameter for every prediction head, a very undesired property when scaling the pretraining corpus to massive scale.
> 2. We finetune word embeddings, as opposed to their shallow infra-feature layer which can be seen as a mere pooling that fuses numerical tokens with word embeddings. Note that most of their transformer is equivalent to our interaction component, and that their "text encoding" component is very shallow. This seems to be a key aspect of TabSTAR's success (section 6), and we believe Tp-BERTa's architecture is suboptimal.
> 3. We suggest a novel mixed verbalization for numerical features, while Tp-BERTa uses special tokens free of semantics which don't exploit parametric knowledge of the LM.
> 4. We encode features separately, while Tp-BERTa has a limited context window, being irrelevant for free text (see Appendix E1).
>
> We hope that these arguments are convincing and we would love to keep discussing. We would also like to mention that both reviewer Cb5W and reviewer qMRw agree with the high novelty of our work.
>
> # Baselines
>
> > doesn't fully represent the current TFM landscape... The lack of comparison against a fuller suite...
>
> We expand the baseline section to include TabICL, the second-best TFM, as well as RealMLP and LightGBM. We observe that TabSTAR significantly beats these non-semantic approaches. For brevity, we only report the normalized metric in the Classification-10K condition, and leave regression and unlimited results for the camera-ready version:
>
> TabSTAR (0.82)
> TabPFN-v2 (0.76)
> CatBoost-Tuned (0.75)
> XGBoost-Tuned (0.73)
> CatBoost (0.7)
> RealMLP-Tuned (0.63)
> TabICL (0.61)
> RealMLP (0.59)
> LGBM-Tuned (0.54)
> LGBM (0.51)
> XGB (0.40)
> CARTE (0.39)
> RF (0.16)
>
> The added models underperform, as they are bounded to frozen, static embeddings. We believe it's highly unlikely that any non-semantic missing baseline could do something fundamentally better. We hope you'll consider our efforts extensive and would love to address any concerns regarding the extent of the evaluation.
>
> > Why were other foundation models for tabular data, such as XTab, not included in the comparison?
>
> XTab was deprecated from the AutoGluon package. To our understanding, the model presented in the paper isn't competitive, as written in the paper: *"XTab outperforms its counterpart FTT in all scenarios thanks to cross-table pretraining, whereas CatBoost is the overall best model."* Furthermore, it was not included in the newly released TabArena leaderboard, although its first author is one of XTab's authors and AutoGluon core developers.
>
> We also tried to include TabDPT, the other foundation model present in TabArena, but its results are surprisingly poor. We contacted their authors to verify the results, and we are collaborating on that matter. We would like to emphasize that we have verified with the authors of ALL the deep models in our paper (TabPFN-v2, CARTE, TabICL, RealMLP) that the baselines were run properly.
>
> # Misleading Framing around "LLMs"
>
> > The model does not use a Large Language Model (LLM)... This limits its ability to perform complex reasoning or leverage the vast world knowledge associated with true LLMs. The improvements likely stem from better-tuned embeddings rather than higher-level reasoning... It does not possess the broad reasoning and knowledge integration abilities of a true LLM.
>
> We appreciate this spot-on observation and analysis, and generally agree with your intuition. We see TabSTAR as both a model and a framework: with enough data and compute, we could replace e5-small-v2 with, for example, Qwen/Qwen3-Embedding-0.6B, making the model ~20x times bigger, and equipping it with much stronger semantic understanding and reasoning abilities. This is possible thanks to our novel target-aware tokens and no dataset-specific parameters, as well as the constant improvement with the increase of the pretraining corpus. We believe that the fact TabSTAR achieves such performance even with limited data and model size is a strength, not a weakness, and we envision future work exploring these areas with industry-level budget pretraining.
>
> Nevertheless, we acknowledge that the paper should be very clear and separate between the current capabilities of the released model, and the potential of the framework. It would be highly beneficial if you could pinpoint the sentences you found confusing, wrong or misleading, to ensure they are rephrased accordingly towards the camera-ready version.
>
> # Unconvincing Performance Gains
>
> > the performance gains over TabPFN-v2 are modest ... suggesting the difference may not be statistically significant in all cases...  improvements in classification, while present, are not overwhelmingly large
>
> TabPFN-v2-API is an extremely strong, closed-source model, powered by a commercial company, Prior Labs, and trained over 130M datasets. This very week Amazon released Mitra, publicly released but without the data. We believe that demanding consistent, significant, cross-dataset improvements vs emerging foundation models backed by industry-level budgets trained on much more data, will hinder progress if it becomes a condition for publishing. We believe we should focus on the potential of TabSTAR, which shows remarkable achievements in low-budget regimes.
>
> In addition, TabPFN-v2 is limited to 10K examples and thus can't compete in realistic dataset sizes, and thus TabSTAR is the definite SOTA for these very realistic use-cases. We limited the comparison to 10K examples exactly to show that we are slightly better than TabPFN-v2 on average even for small-medium datasets, but this is an artificial limitation, derived by their constraints. We see this as a crucial point: foundation models should also focus on bigger datasets, and TabSTAR shows that it scales well.
>
> > may not be statistically robust across all datasets.
>
> While TabSTAR doesn't consistently beats TabPFN-v2 across all datasets, Table 18 shows the per dataset comparison with its respective confidence intervals. For some datasets (C10, C13) the gain is of 2-3%. We would like to refer again to TabArena, written by strong leading researchers: "*We designate TabArena-Lite to be used in academic studies and find any novel model that outperforms other models on at least one dataset, even if it is not among the best on average, a valuable publication*". We believe that TabSTAR more than fully complies with this expectation.
>
> As a last point, we want to highlight that our work doesn't contribute only a SOTA model, but also important insights about how virtually all other algorithms miss the potential of Semantic Target-Aware Representations. This is especially relevant as we see new foundation models ignoring semantics whatsoever, and believe that our work will inspire competing models to benefit from our ideas and set new SOTA performance on the important, neglected task of tabular datasets with text features.
>
> # Regression Performance
>
> > The model is not competitive with established GBDT models on regression tasks
>
> We agree that our regression performance is a point for improvement, and would like to pinpoint that TabPFN-v1 and TabICL were released as classification-only models, and that Tp-BERTa trained separate models for regression and classification. Nevertheless, since TabSTAR consistently improved with more regression datasets and still beats several baselines, we proudly decided to publish these results, and open-sourced our full code, as we want to inspire future research to extend our work and tackle this challenge.
>
> > Beyond scaling the pretraining data, are there specific architectural weaknesses that you believe cause this gap? ... indicating that the current architecture is not well-suited for predicting continuous values.
>
> We discuss in the paper several potential reasons:
>
> 1. Lack of data: we use ~100 regression datasets, this is probably not enough to generalize and transfer learn this complex task. Scaling the training set with millions of synthetic regression tasks (e.g. TabICL's public data) could provide a huge impact.
>
> 2. Better numerical modeling: we used a naive single transformation for numerical features, which potentially limits TabSTAR's numerical capabilities. We believe this could be challenged, similarly to how "On embeddings for numerical features in tabular deep learning" expanded FT-Transformer.
>
> 3. Regression-as-classification: discretizing the target value and using binned (target) tokens, similarly to CHRONOS and other TS FMs.
>
> > Target-Awareness in Regression: For regression tasks, a single target token representing the target's name is used. How does this provide meaningful "awareness" compared to the classification case, where all possible target values are provided as input?
>
> During pretraining, the model sees different verbalizations per regression datasets, e.g. "Numerical Target Feature: house price" and "Numerical Target Feature: dog age". This verbalization could help transfer learning similar distributions, and potentially steer the LM to provide potentially useful prior knowledge emerging from its pretraining corpus. This would be boosted when plugging a stronger LLM.

---

> > ### Comment · Reviewer_1N8m · 2025-08-08
> >
> > Thanks for your responses.
> >
> > I keep my original score due to concerns in incremental novelty and unconvincing performance gains.
> >
> > In my opinion, the listed novelty points are not substantial.
> >
> > 'foundation models backed by industry-level budgets trained on much more data, will hinder progress if it becomes a condition for publishing" - on this point, there could be studies on showcasing scaling laws. Even if the budget does not allow full scaling, extrapolations into the large compute regime would be useful.

---

> ### Comment · Area_Chair_72DM · 2025-08-04
> **Author-Reviewer Discussion Ends Aug. 6**
>
> Dear Reviewer,
>
> Thank you for contributing to peer review at NeurIPS. Please respond to the rebuttal before Aug. 6 and complete the Mandatory Acknowledgment.
>
> Thank you,
>
> AC

---

> ### Author Response · Authors · 2025-08-08
>
> Thank you for responding.
>
> > there could be studies on showcasing scaling laws. Even if the budget does not allow full scaling, extrapolations into the large compute regime would be useful.
>
> We would like to highlight that we conducted such analysis in Section 6, and show that TabSTAR significantly improves with scale. We believe that extrapolating from a few hundreds into the hundreds of millions is virtually impossible, and stating an exact number in the paper would have probably been wrong. Nevertheless, we have conducted such calculations and we estimate that with ~10K datasets (x25 from now) we will not only push the SOTA barriers in classification, but also achieve near SOTA performance in regression.
>
> This is possible due to TabSTAR’s novel introduction of target-aware tokens, which disable the need for per-dataset heads. This contrasts with other works you mentioned like Tp-BERTa or XTab, which could never be pretrained in scale, as they use in their architectures dataset specific parameters. At that scale, the model will have more dataset-specific parameters than actual shared parameters!
>
> We hope this makes even clearer why we believe the performance of TabSTAR's framework is so strong, and why we believe our approach is novel and provides a substantial improvement above Tp-BERTa.

---

> > ### Comment · Reviewer_1N8m · 2025-08-08
> >
> > The main results are conducted with 350 pretraining datasets and the ablations show that with 256 datasets, the overall results are not that different?
> >
> > Please refer to scaling laws studies in the literature that clearly show how the performance evolves with the number of samples/tokens/dataset sizes. Table 3 is not sufficient in my opinion.
> >
> > If this model and training recipe are not showing large improvements in average across all datasets, I suggest focusing on the scenarios where you can bring significant improvements and considering paper repositioning around that.

---

> ### Author Response · Authors · 2025-08-08
>
> Thanks for engaging in the discussion :)
>
> Re scaling: So the question goes down to this - what will happen if you pre-pretrain TabSTAR's architecture with the 130M datasets used by TabPFN? We have no answer for that, but I hope you agree it will be quite surprising if this doesn't help. In addition, with such variety and richness of regression datasets, it's much more plausible that regression-as-classification will work, as they do in CHRONOS and in TabPFN. This approach is much harder to explore with 100 datasets.
> (Obviously, the TabPFN data is non-semantic, but we believe there are different ways we can exploit that information)
>
> At the end of the day we have an architecture which some novel components that make scaling possible, and introduce semantics and flexibility to tune embeddings in a way that non other approach does. We believe that this is the path towards solving tabular datasets with unstructured features, e.g. text but also images in a future. And we believe that this problem is much more important than the focus it gets today, and its performance is limited with all other approaches.

---

> > ### Comment · Reviewer_1N8m · 2025-08-08
> >
> > I think all these comments require substantially amount of additional results and paper rewriting beyond this rebuttal time.
> >
> > On the point of text-tabular multimodality, the paper writing needs to be modified to better highlight such results and studies.

---

> ### Author Response · Authors · 2025-08-08
>
> We agree that pre-pretraining over such corpus is out of the scope of this paper. We believe it's a worthy future work attempt, as well as many other exciting research directions that we believe our paper will inspire.
>
> Our idea was to propose an architecture that introduces a few novelties, unlock scale both for pretraining (no dataset specific parameters) as well as downstream task (ICL is also a limitation, as you can see from our "unlimited" condition in the main results: PFNs currently aren't a solution for bit enough datasets, and while TabICL improved efficiency, its performance on the benchmark on that condition is underwhelming).
>
> We believe that the paper is quite clear regarding the potential of scaling and semantic knowledge, but we agree that we could have elaborated on the discussion that we believe that our architecture can be well suited for multimodal tabular learning, as opposed to other alternatives. We will amend the paper and highlight this for the camera-ready version. We thank you for pointing this out, as we believe this emphasizes even more the value of our work.

---

### Official Review · Reviewer_Cb5W · 2025-07-03

**Clarity:** 4
**Significance:** 3
**Originality:** 3
**Rating:** 5
**Confidence:** 4

**Summary:**

The paper presents TabSTAR, a novel tabular foundation model that incorporates semantically target-aware tokens into an unfrozen pretrained text encoder, enabling end-to-end fine-tuning of textual and numerical features for tabular prediction tasks. The authors curate a large pretraining corpus of 350 tabular datasets (with both classification and regression tasks), demonstrate strong empirical performance on 50 benchmarks containing free-text or high-cardinality features, and show that TabSTAR scales with more pretraining data.

**Questions:**

- Could you please clarify the no. of evaluated trials per model and dataset? Also, were the results of the predictions of tree-based models of the five folds averaged (bagging)?
- Can you provide quantitative evidence isolating the benefit of a) including column names versus solely the values; b) including target tokens instead of just unfreezing the text embeddings using the values?
- Does the fusion layer provide any gain over simple concatenation? I am unsure whether this part of the modelis important.
- Figure 4: Looking at the right plot, it seems like the normalization was done solely on the TabStar versions, without the other models. Although the difference to the 0-unfrozen setup is quite large, the true impact on the final downstream performance cannot be assessed using this plot. The same seems to be the case for Tables 3 and 4. How would the plot look like if the other models would be included in the normalization?
- Table 3: The gains from pretraining seem surprisingly high. WHat are your thought on why TabStar is much worse than conventional neural networks in the 'classic' training regime?

**Ethical Concerns:**

["NO or VERY MINOR ethics concerns only"]

**Final Justification:**

Given the strong submission and rebuttal, I keep my positive score.

**Limitations:**

yes

**Paper Formatting Concerns:**

None.

**Quality:**

3

**Strengths And Weaknesses:**

### Strengths:
- Clear motivation & positioning. The paper convincingly identifies the limitations of static, target-agnostic text embeddings in existing tabular ML and foundation models, particularly when handling free-text or high-cardinality categorical features.
- Extensive related work section covering major approaches for tabular data.
- Well-written and easy-to-follow.
- Well-curated pretraining dataset collection and extensive efforts in avoiding contamination.
- Thoughtful choice of the evaluated baseline models. In particular, the exclusion of models such as TP-BERTa or Tabula, are very reasonable given the severe leakage concerns and unreasonable training times.
- Extensive appendix with all required information to reproduce the results.
- Insightful analyses. Ablations on the number of unfrozen encoder layers, the effect of pretraining dataset count, and different numerical verbalizations are very helpful.

- High novelty, as there is no comparable approach that uses "target-aware tokens" in tabular tasks.
- Empirical results are substantial, covering various datasets and providing clear comparative analyses against robust baseline methods.


### Weaknesses:
- Evaluation of baselines using a 4h time budget without reporting the no. of successful trials per model. The average computation times per 10k features in the Appendix are a reference, but without knowing how many trials were truly evaluated it is impossible to tell whether the baselines are well-tuned or not. As they are currently presented, the conclusions are only valid under the 4h time budget. A practitioner would invest much more time to solve a single task. Therefore, the general claim 'TabSTAR achieves state-of-the-art performance' cannot be substantiated. Therefore, the claim in the abstract should either be changed to reflect that the claim only holds with a 4h time budget, or more information on the no. of successful trials per model is required, to verify that the tree-based models were truly tuned close to their full potential.
- No classic deep learning methods (i.e. TabM or RealMLP) included.
- Despite being highlighted, computational cost and efficiency are not deeply analyzed. Given that gradient boosting decision trees (GBDTs) are highly performant with lower computational overhead, this factor should be more explicitly addressed. How does TabStar compare to tree-based models in terms of memory efficiency and inference time?
- Training on some of the benchmark datasets and evaluating five different models adds additional complexity to the evaluation, and, at the same time requires an additional ablation. How is the performance of a single TabStar model without using the additional training data from the downstream task benchmark?
- While the ablations are insightful, the core innovation that is also present in the full title have not been studied in isolation. Is it really beneficial to use target-aware tokens, or is it simply enough to retrain (unfreeze) the layers without providing semantic target information?

---

> ### Author Rebuttal · Authors · 2025-07-30
>
> Dear reviewer Cb5W,
>
> We appreciate that you recognize the strength of the results, the novelty of our work and the clarity of our paper. We carefully considered your questions and potential weaknesses and would like to address them thematically.
>
> # Baselines
>
> > No classic deep learning methods (i.e. TabM or RealMLP) included.
>
> We added RealMLP, LightGBM and TabICL. For brevity, we only report the normalized metric in the Classification-10K condition, but a similar behavior occurs in the unlimited one. For regression, the non-tuned based algorithms were below TabSTAR. The full results will appear in the camera-ready version.
>
> TabSTAR (0.82)
> TabPFN-v2 (0.76)
> CatBoost-Tuned (0.75)
> XGBoost-Tuned (0.73)
> CatBoost (0.7)
> RealMLP-Tuned (0.63)
> TabICL (0.61)
> RealMLP (0.59)
> LGBM-Tuned (0.54)
> LGBM (0.51)
> XGB (0.40)
> CARTE (0.39)
> RF (0.16)
>
> The added models underperform, as they are bounded to frozen, static embeddings. We believe it's highly unlikely that any non-semantic missing baseline could do something fundamentally better. We would like to emphasize that part of our contribution is not only SOTA performance, but also identifying a gap that hopefully will inspire competing methods to include a semantic approach, as this seems crucial for tabular datasets with text.
>
> # GBDTs Tuning
>
> > Could you please clarify the no. of evaluated trials per model and dataset?
>
> We apologize for forgetting to report it in the paper and will add to the camera-ready paper the full details. Here is the average number of successful trials for per classification dataset for CatBoost at the 10K condition, sorted as in Table 11:
> [27.1, 18.1, 18, 105.8, 122.1, 18.1, 17.3, 101.1, 169, 73.7, 58.3, 9.9, 85.8, 8.6].
>
> We tuned the trees with each trial running on one of the 8 cores, letting it complete at least a single trial and then refitted the best model. For two of the datasets, it took much more than the 4hs budget.
>
> > were the results of the predictions of tree-based models of the five folds averaged (bagging)?
>
> No, we refitted the model on the best configuration on the whole dataset, following TabPFN-v2 and CARTE.
>
> > the claim in the abstract should either be changed to reflect that the claim only holds with a 4h time budget
>
> Point taken, we'll refine the statement.
>
> > A practitioner would invest much more time to solve a single task
>
> Agreed, but note that TabSTAR’s downstream performance could also improve, as we train a single fold with no hyperparameter tuning.
>
> >… verify that the tree-based models were truly tuned close to their full potential…
>
> We understand your concern and we were obsessed with doing things right. We followed leading papers and found contradictions, ambiguity and no public code (until the concurrent release of TabArena). However, since for some datasets the gap is significant even vs TabPFN-v2-API, we believe our results are strong, transparent and replicable (we made the evaluation code public as well). In addition, TabSTAR is not just a model, but a pretraining framework which should improve with further scaling as we suggest in section 6. We thank you for the comments and would love to keep addressing any concerns or questions you might have on this matter during the discussion period.
>
> # Computational Costs
>
> > computational cost and efficiency are not deeply analyzed.
>
> Since we were driven by peak performance, and since TabSTAR could be optimized further with inference efficiency in mind, this was not a main focus. Nevertheless, this request came from multiple reviewers so we took it very seriously. We changed our code to meticulously log CPU/GPU memory and running times for both training and inference time, and we plan to rerun TabSTAR as well as a subset of the baselines. This analysis will complement the results section in the main paper.
>
> > Given that gradient boosting decision trees (GBDTs) are highly performant with lower computational overhead
>
> While true in general, we found that for tabular data with text features that require embedding models, GBDTs must use a GPU to hold this claim. Serving embeddings models with CPU-only makes both training and inference orders of magnitude slower. This claim is very important and non-trivial.
>
> > How does TabStar compare to tree-based models in terms of memory efficiency and inference time?
>
> Due to the rebuttal time constraints, we share a partial analysis of TabSTAR vs the best open-source representative for other categories: CatBoost, RealMLP and TabICL. As mentioned above, all baselines used a GPU to be competitive. While TabSTAR is GPU memory-hungry during training (3x more than others), it is memory efficient with comparative memory and latency to GBDTs. On the contrary, while TabICL has virtually no training time, its inference is prohibitively slower (~10x). As the community rapidly adapts ICL as the leading method for doing Foundation Models, the upside of no training time comes with the cost of GPU-based low-latency/high-memory inference. TabSTAR, on the contrary, is inference-time efficient.
>
> # Semantics Analysis
>
> > Can you provide quantitative evidence isolating the benefit of a) including column names versus solely the values; b) including target tokens instead of just unfreezing the text embeddings using the values?
>
> Great question, we agree that the benefit of including target tokens could have been analyzed in the paper more deeply, although the benefit of including column names worked for many architectures (TransTab, CM2, TpBERTa, CARTE). We conducted an experiment that will be added to the camera-ready paper, and got the following results:
>
> TabSTAR (0.8404)
> No Target Values (0.8387)
> No Column Names (0.8354)
> "Anonymized": Neither (0.8339)
>
> While the results are on the right track, the effect is moderate; it's worth noting that many pretraining datasets have no rich semantics, so they lower the average effect. We see TabSTAR's ability to thrive even in the absence of semantic information as a strength and a robust property.
>
> > Is it really beneficial to use target-aware tokens, or is it simply enough to retrain (unfreeze) the layers without providing semantic target information?
>
> It's worth emphasizing that TabSTAR is an architecture destined to scale, and that target-aware tokens should shine with more data, stronger LMs and less tuning. We believe that the further we go that road, the more semantic impact and world knowledge we'll witness.
>
> # Fusion Block
>
> > Does the fusion layer provide any gain over simple concatenation?
>
> In Appendix A3 we report the gain of the fusion layer over simpler approaches. The exact (unreported) results:
>
> Fusion (0.8351)
> Multiplication (0.8280)
> Concatenation (0.8241)
>
> The concatenation approach causes losing 1% on average. While multiplication is a non-parametric simpler alternative, we believe that the fusion component will scale better and lends itself to multimodality (e.g., adding image features, or richer numerical representations). We show additional experiments in our response to reviewer mFUa, under the header “Fusion vs Interaction”.
>
> # TabSTAR Eval Complexity
>
> > Training on some of the benchmark datasets and evaluating five different models adds additional complexity to the evaluation
>
> When training on real datasets, evaluating downstream tasks is a concern due to contamination risks. To mitigate it we released several TabSTAR models, so every pretraining dataset can be matched no a non-contaminated variant, and plan to support this mapping efforts for follow-up research evaluating TabSTAR. We believe that this step is necessary in the era of tabular foundation models, and hope to inspire practitioners, since success in benchmark is de-facto the way models are judged in academic settings.
>
> > How is the performance of a single TabStar model without using the additional training data from the downstream task benchmark?
>
> It should probably be slightly below TabPFN-v2, but stronger than trees. Growing from 350 to 390 datasets, especially as they are textual, was important to consolidate ourselves as SOTA. We see this as a technical detail, and with a larger pretraining corpus this practice is definitely not mandatory for evaluation, but important when releasing models.
>
> # Normalized Plots
>
> > Figure 4: Looking at the right plot ... How would the plot look like if the other models would be included in the normalization?
>
> We avoided this comparison as its done on a subset of the datasets, and thus the conclusions could be sensitive when comparing to other models. Nevertheless, we report the ranked variants from best to worse in Classification 10K and plan to include this information in Appendix G, with the appropriate disclaimers:
>
> TabSTAR: Scale 256 / Unfreeze-6 / Numerical-Full
> Numerical-Range
> TabPFN-v2
> Scale-64
> Unfreeze-3
> Numerical-None
> CatB-Tuned
> Unfreeze-9
> XGB-Tuned
> CatB
> Unfreeze-1
> Scale-16
> Scale-0
> RealMLP-Tuned
> TabICL
> RealMLP
> LGBM-Tuned
> LGBM
> CARTE
> XGB
> RF
> Unfreeze-0
>
> While we believe that strong conclusions shouldn’t be drawn from this limited comparison, it’s quite clear that both the pre-training scale and the unfreezing of layers are key to TabSTAR’s success.
>
> # Pretraining gains
>
> > The gains from pretraining seem surprisingly high. WHat are your thought on why TabStar is much worse than conventional neural networks in the 'classic' training regime?
>
> TabSTAR’s architecture is designed for transfer-learning without dataset-specific parameters, with components that aim to generalize, which can be a limited decision without pretraining. For example, the numerical encoder maps every (normalized) numerical value to a vector in the same way. Had we allowed for feature-specific mapping (like FT-Transformer) we could probably improve on the non-pretraining scenario, but this won't allow pretraining at scale. Similarly, the Fusion block might be an overkill without pretraining, but shines with enough data.

---

> > ### Comment · Reviewer_Cb5W · 2025-08-06
> >
> > Thank you for your detailed rebuttal and for taking the time to address my concerns.
> >
> > However, my concerns regarding the appropriate tuning of the baselines remain: 1) The choice of refitting over bagging is a limitation as bagging usually shows higher performance. 2) Even in the medium data regime with 10K samples, CatBoost had less than 20 successful trials on many datasets due to the restrictive time budgets. In addition, likely due to the choice of refitting, the no. of estimators was part of the search space and no early stopping was conducted. This leads to potentially underfitted models for the trials where a low no. of estimator was used. Due to these choices, I believe that the tree-based models were not evaluated to their full potential. For future work, I would recommend to tune at least one baseline to its full potential with less restrictive time budgets. Otherwise, it remains unclear whether the method truly has the potential to outperform baselines.
> >
> > Nevertheless, given the otherwise strong submission and rebuttal, I will keep my positive score.

---

> > > ### Author Response · Authors · 2025-08-06
> > >
> > > We thank you for your feedback and your strong assessment of our work.
> > >
> > > We replicated the evaluation protocol of TabPFN-v2 and CARTE, two leading foundation models coming from distinguished research labs, thinking that this was the best practice. In both works the number of iterations was part of the hyperparameter tuning search space (see Extended Data Table 5 in the TabPFN-v2 paper and Table 4 in CARTE).
> > >
> > > Nevertheless, we accept your comment, and will take this in consideration for future work as you suggested.

---

### Note · Authors · 2025-08-11

In this work we present TabSTAR, a Foundation Tabular Model with Semantically Target-Aware Representations. We believe that TabSTAR has the potential to become a very influential paper, due to its novelty, impressive result, and core contributions:

1. TabSTAR achieves SOTA performance on classification with text features, a fundamental problem with high importance.
2. TabSTAR’s architecture lends itself to pretraining at scale, with an architecture with no dataset-specific parameters thanks to the novel target-aware tokens.
3. TabSTAR’s architecture presents an inference-time efficient, scalable and highly-performant alternative to ICL Foundation Models, the leading approach nowadays. We also leverage the semantic parametric knowledge of language models to enable transfer learning, which is a factor missing in the priors of PFNs (Prior Fitted Networks).

We thank the reviewers for their time, effort, and constructive feedback. We replied to all concerns and questions during both the rebuttal and the discussion phase, and we believe that we defended the paper well. A few weaknesses were shared by more than one reviewer during the rebuttal phase:

1. Baselines: We added 3 more leading models which were significantly dominated by TabSTAR, as they are limited to frozen representations.
2. Missing analysis of latency and memory consumption: The results show that TabSTAR is efficient during inference time due to its non-ICL nature, and since GPUs are necessary when dense embeddings are used.
3. Regression performance: This limitation is acknowledged, and we believe both the paper and the rebuttal show encouraging results on how future generations will improve further, to match our remarkable classification performance.

To summarize, we humbly believe TabSTAR’s novelty and strong results in the very important, under-explored problem of tabular data with text features, are a strong contribution to the scientific community and will impact this important line of research.

---

### Decision · Program_Chairs · 2025-09-17

**Decision:**

Accept (poster)

**Comment:**

This paper presents a new architecture for a tabular foundation model that uses target aware tokens and Semantic Target-Aware Representations. It verbalizes all features into readable strings, encodes them with a pre-trained text embedding model, processes them with transformers, and finally uses a prediction head fine-tuned for a given dataset on the representations. Compared to existing tabular foundation models and non-foundational architectures, TabSTAR performs well on classification tasks (but poorly on regression).

The reviewers initially raised many different issues, including: limited novelty; debate on what counts as a foundation model (i.e. should fine-tuning be a requirement or allowed at all); the poor results on regression; scalability especially with number of classes and features; incomplete baselining; and missing evaluations on cost and efficiency.

Many of these issues were resolved in the discussions. Novelty was more clearly described in terms of target aware tokens and Semantic Target-Aware Representations. Scalability was discussed in comparison to other transformer-based TFMs. The authors presented additional baseline methods, both foundational and not (RealMLP, LightGBM, TabICL), although several relevant models could still be added (e.g. TabDPT a foundation model that shows strong regression performance on TabArena, TabM a non-foundation model with strong classification performance on TabArena).

One of the biggest outstanding issues is poor regression performance, which was mostly left for future improvements. However, a new foundation model that works well for classification can still be interesting on its own, as many tabular papers focus entirely on classification.

Hence, I am recommending to accept this submission. The authors must complete the baselining of their approach on the methods discussed with reviewers (atleast RealMLP, LightGBM, TabICL, TabDPT, TabM), and include the other improvements that resolved reviewers’ issues. Authors -- please make a comment when your camera-ready version is up with the completed baselining for I and the reviewers to check.